# Overlapping Clustering Models, and One (class) SVM to Bind Them All

**Xueyu Mao, Purnamrita Sarkar, Deepayan Chakrabarti**
The University of Texas at Austin
xmao@cs.utexas.edu, purna.sarkar@austin.utexas.edu, deepay@utexas.edu

## Abstract

People belong to multiple communities, words belong to multiple topics, and books cover multiple genres; overlapping clusters are commonplace. Many existing overlapping clustering methods model each person (or word, or book) as a non-negative weighted combination of "exemplars" who belong solely to one community, with some small noise. Geometrically, each person is a point on a cone whose corners are these exemplars. This basic form encompasses the widely used Mixed Membership Stochastic Blockmodel of networks [1] and its degree-corrected variants [16], as well as topic models such as LDA [9]. We show that a simple one-class SVM yields provably consistent parameter inference for all such models, and scales to large datasets. Experimental results on several simulated and real datasets show our algorithm (called SVM-cone) is both accurate and scalable.

## 1 Introduction

Clustering has many real-world applications: market segmentation, product recommendation, document clustering, finding protein complexes in gene networks, among others. The simplest form of a clustering model assumes that every record or entity belongs to exactly one cluster. More general forms allow for overlapping clusters, where each entity may belong to different clusters or communities to different degrees. For example, George Orwell's *1984* belongs to both the dystopian fiction and political fiction genres, and *Pink Floyd's* music is both progressive and psychedelic. In this paper, we show that many existing overlapping clustering models can be written in a general form, whose parameters can then be inferred using a one-class SVM.

In many clustering problems, overlapping or otherwise, we have access to a data matrix $\hat{\mathbf{Z}} \in \mathbb{R}^{n \times m}$, which is a noisy version of an ideal matrix $\mathbf{Z}$, i.e. $\hat{\mathbf{Z}} = \mathbf{Z} + \mathbf{R}$ where the norm of the rows of $\mathbf{R}$ is small. Also, $\mathbf{Z} = \mathbf{G}\mathbf{Z}_P$, where $\mathbf{Z}_P$ are ideal "exemplars" of the various communities, and $\mathbf{G} \in \mathbb{R}_{\geq 0}^{n \times K}$ gives the community memberships of each entity. We will now give some examples.

Consider the Stochastic Blockmodel (SBM) [13] for networks. In this model, each node belongs to one of $K$ communities, and the probability $\mathbf{P}_{ij}$ of an edge between nodes $i$ and $j$ is a function of their respective communities. Recent results [21] show that the eigenvectors $\hat{\mathbf{V}}$ of the adjacency matrix concentrate row-wise around the eigenvectors $\mathbf{V}$ of $\mathbf{P}$. The matrix $\mathbf{V}$ is also blockwise constant, mapping all nodes in one cluster to one point [27]. Hence, $\hat{\mathbf{V}} = \mathbf{G}\mathbf{V}_P + \mathbf{R}$, where $\mathbf{G} \in \{0, 1\}^{n \times K}$ is a binary membership matrix where each row sums to one. The Mixed Membership Stochastic Blockmodel (MMSB) [1] relaxes this by allowing the entries of $\mathbf{G}$ to be in $[0, 1]$. Since the rows of $\mathbf{G}$ sum to one, the ideal matrix $\mathbf{Z}$ arranges points in a simplex. The corners of this simplex represent the "pure" nodes, i.e. nodes belonging to exactly one community. Most algorithms first find the corners, and then estimate model parameters via regression [20, 21, 16, 23, 28]. Other notable methods include tensor based approaches [14, 2], Bayesian inference [12], etc. Related models and inference methods for overlapping networks have been presented in [24, 17, 26, 19], etc.

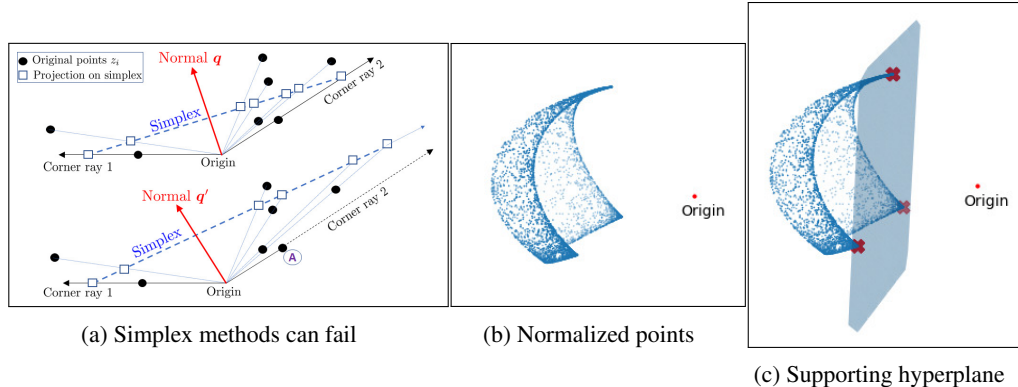

(a) Simplex methods can fail     (b) Normalized points

(c) Supporting hyperplane

Figure 1: (a) Simplex-based corner-finding methods require points on a simplex, with uniformly small errors. Projecting points to a simplex with normal vector $q$ works well, but a very similar $q'$ does not. Some points (such as Ⓐ) get projected to far-off points, amplifying errors in their positions. (b) Instead, we normalize points to the unit sphere, and (c) find corners from the support vectors of a one-class SVM.

The MMSB model does not allow for degree heterogeneity, which can be achieved via the Degree-corrected Mixed Membership Stochastic Blockmodel (DCMMSB) [16]. In DCMMSB, each node has an extra degree parameter, with a high parameter value leading to more edges for that node. Now, $\mathbf{G}$ is non-negative, but its rows do not sum to one. Thus the points lie inside a cone, and the pure nodes lie on the corner rays of this cone. Other network models also give rise to such cones [32, 17].

Existing algorithms for degree corrected overlapping models use a range of different techniques. OCCAM [32] uses a $k$-medians step on the regularized eigenvectors of the adjacency matrix to get the corners. While the algorithm is computationally efficient, a key assumption is that the $k$-medians loss function attains its minimum at the locations of the pure nodes and there is a curvature around this minimum. This condition is typically hard to check. In [16], the authors show an interesting result that the second to $K$ eigenvectors, element-wise divided by the first eigenvector entries form a simplex. The authors provide an algorithm for finding this simplex with $K$ corners in $K-1$ dimensions. The algorithm requires a combinatorial search step, which is prohibitive for large $K$.

Topic models [9] are another example of overlapping clustering models. Here the documents can be generated from a mixture of topics, which are the analog of communities in networks. The normalized word co-occurrence matrix forms a simplex structure, with the corners representing anchor words, i.e. words that belong to exactly one topic. While there are many existing inference methods, the ones that provide consistency guarantees are typically based on analyzing tensors or finding corners in simplexes [4, 18, 11, 3, 15, 7, 6, 8].

In this paper, we provide an overarching framework which incorporates all the above problems, from Mixed membership models (with or without degree correction) to topic models. As discussed before, in all the above models, the ideal data matrix lies inside a cone (a simplex is a special type of a cone). The goal is to infer $\mathbf{G}$, which depends on finding the correct corner rays.

Let us illustrate why seemingly obvious methods fail to obtain the corner rays. The simplest idea would be to generate a random plane (the "simplex"), and project points to the intersection of the line joining these points to this random plane as shown in Figure 1a. Corners of this simplex correspond to the corner rays. However, extending the idea to the sample or empirical cone is difficult, because if the simplex is not good, some points can get projected to arbitrarily far points, which will amplify their error. As Figure 1a shows, the set of good simplexes may be quite limited, and finding a good simplex is difficult.

We will illustrate our idea with the "ideal cone." First, we row-normalize the ideal data matrix to have unit $\ell_2$ norm (similar to Ng et al. [22], Qin and Rohe [25]). This projects all points inside the cone to the surface of the sphere, with the points on the corner rays being projected to the corners (Figure 1b).

Then, we show a rather fascinating result, namely, for all the above models, the corners can be obtained via the support vectors of a one class SVM [29], where all normalized points are in the positive class, and the origin is in the negative class (Figure 1c). Observe that a hyperplane through

the corners separates all the points from the origin. We also show that if the row-wise error of $\mathbf{R}$ is small, the SVM approach can be used to infer $\hat{\mathbf{G}}$ from empirical cones. Finally, we show that since the row-wise error of $\mathbf{R}$ is indeed small for different degree-corrected overlapping network models and topic models, we can use our algorithm to infer the parameters consistently. We provide error bounds for parameter estimates at the *per-node* and *per-word* level, in contrast to typical bounds for the entire parameter matrix. We conclude with experimental results on simulated and real datasets.

## 2 Proposed work

Consider a population matrix $\mathbf{P}$ of the form $\mathbf{P} = \rho\mathbf{\Gamma}\mathbf{\Theta}\mathbf{B}\mathbf{\Theta}^T\mathbf{\Gamma}$, with $\mathbf{\Gamma} \in \mathbb{R}_{>0}^{n \times n}$ being a positive diagonal matrix, $\mathbf{\Theta} \in \mathbb{R}_{\geq 0}^{n \times K}$ a community-membership matrix, and $\mathbf{B} \in \mathbb{R}_{\geq 0}^{K \times K}$ a cross-community connection matrix. We will make the following assumptions which are common in the literature:

**Assumption 2.1.** (a) *Pure nodes:* Each community has at least one "pure" node, which belongs solely to that community. (b) *Non-zero rows:* No row of $\mathbf{\Theta}$ is identically 0. (c) $\mathbf{B}$ is full rank.

The form of the population matrix $\mathbf{P}$, alongside Assumption 2.1, induces a conic structure on the rows of the eigenvectors of $\mathbf{P}$.

**Lemma 2.1.** *Let there be $K$ communities (rank$(\mathbf{P}) = K$), and let $\mathcal{I}$ be indices of $K$ pure nodes, one from each community. Let $\mathbf{P} = \mathbf{V}\mathbf{E}\mathbf{V}^T$ be the top-$K$ eigen-decomposition of $\mathbf{P}$, where columns of $\mathbf{V} \in \mathbb{R}^{n \times K}$ are the $K$ principal eigenvectors and $\mathbf{E} \in \mathbb{R}^{K \times K}$ is a diagonal matrix of the $K$ principal eigenvalues. Then, $\mathbf{V} = \mathbf{\Gamma}\mathbf{\Theta}\mathbf{\Gamma}_P^{-1}\mathbf{V}_P$, where $\mathbf{V}_P = \mathbf{V}(\mathcal{I}, :)$ is full rank and $\mathbf{\Gamma}_P = \mathbf{\Gamma}(\mathcal{I}, \mathcal{I})$.*

Since $(\mathbf{\Gamma}\mathbf{\Theta}\mathbf{\Gamma}_P^{-1})_{ij} \geq 0$ for all $(i, j)$, the rows of $\mathbf{V}$ fall within a cone with corners $\mathbf{V}_P$. This suggests the following idealized problem:

**Problem 1** (Ideal cone problem). *We are given a matrix $\mathbf{Z} \in \mathbb{R}^{n \times m}$ such that $\mathbf{Z} = \mathbf{M}\mathbf{Y}_P$, where $\mathbf{M} \in \mathbb{R}_{\geq 0}^{n \times K}$, no row of $\mathbf{M}$ is 0, and $\mathbf{Y}_P \in \mathbb{R}^{K \times m}$ corresponds to $K$ (unknown) rows of $\mathbf{Z}$, each scaled to unit $\ell_2$ norm. Infer $\mathbf{M}$ from $\mathbf{Z}$.*

The rows of $\mathbf{Y}_P$ are unit vectors representing the corner rays of the cone. Each row of $\mathbf{Z}$ is constructed from a non-negative weighted combination of these unit vectors, with the weights being given by the corresponding rows of $\mathbf{M}$. Rows of $\mathbf{Z}$ that lie on some corner correspond to rows of $\mathbf{M}$ that have zero in all but one component. Observe that $\mathbf{M}$ is invariant to the choice of $K$ corner rows of $\mathbf{Z}$ used to construct $\mathbf{Y}_P$.

Now consider solving the ideal cone problem with the eigenvector matrix, i.e., $\mathbf{Z} = \mathbf{V}$. From Lemma 2.1, the corner rows correspond to the pure nodes. Choosing one such row from each corner gives us a set of pure node indices $\mathcal{I}$. Hence, $\mathbf{M} = \mathbf{\Gamma}\mathbf{\Theta}\mathbf{\Gamma}_P^{-1}\mathbf{N}_P^{-1}$, where $\mathbf{N}$ is a diagonal matrix with $\mathbf{N}_{ii} = 1/\|\mathbf{e}_i^T\mathbf{Z}\|$ and $\mathbf{N}_P = \mathbf{N}(\mathcal{I}, :)$. We also have the identity $\rho\mathbf{\Gamma}_P\mathbf{B}\mathbf{\Gamma}_P = \mathbf{V}_P\mathbf{E}\mathbf{V}_P^T$. Coupled with model-specific identifiability conditions (details are provided in the supplementary material), these can be used to infer $\mathbf{\Theta}$ and $\rho\mathbf{B}$ ($\mathbf{\Gamma}$ are typically considered nuisance parameters).

In practice, we only have an observation matrix $\mathbf{A}$ that is stochastically generated from the population matrix $\mathbf{P}$. Hence, we must actually solve:

**Problem 2** (Empirical cone problem). *We are given a matrix $\hat{\mathbf{Z}} \in \mathbb{R}^{n \times m}$ such that $\max_{i \in [n]} \|\mathbf{e}_i^T(\mathbf{Y} - \hat{\mathbf{Y}})\|_2 \leq \epsilon$, where $\mathbf{Y} = \mathbf{N}\mathbf{Z}$ is the row-normalized version of $\mathbf{Z}$, and $\hat{\mathbf{Y}}$ is constructed similarly from $\hat{\mathbf{Z}}$. Again, $\mathbf{Z} = \mathbf{M}\mathbf{Y}_P$, where $\mathbf{M} \geq 0$, no row of $\mathbf{M}$ is 0, and $\mathbf{Y}_P = \mathbf{Y}(\mathcal{I}, :)$ corresponds to $K$ (unknown) rows of $\mathbf{Y}$ with indices $\mathcal{I}$. Infer $\mathbf{M}$ from $\hat{\mathbf{Z}}$.*

We will first present the solution to the ideal cone problem. We will then show that the same algorithm with some post-processing solves the empirical cone problem up to $O(\epsilon)$ error. Finally, we apply our algorithm to infer parameters for a variety of models, and present error bounds for each.

**Notation:** We shall refer to the $i^{th}$ row of $\mathbf{Z}$ as $\mathbf{z}_i^T$ *expressed using a column vector*, i.e., $\mathbf{z}_i = \mathbf{Z}^T\mathbf{e}_i$. The same pattern will be used for rows $\mathbf{m}_i^T$ of $\mathbf{M}$, and other matrices as well.

### 2.1 The Ideal Cone Problem

Observe that given the corner indices $\mathcal{I}$ (i.e., given $\mathbf{Y}_P$), finding $\mathbf{M}$ such that $\mathbf{Z} = \mathbf{M}\mathbf{Y}_P$ is a simple regression problem. Thus, the only difficulty is in finding the corner indices.

Our key insight is that under certain conditions, the ideal cone problem can be solved easily by a **one-class SVM applied to the rows of Y**. Figure 1 plots the normalized rows $\mathbf{y}_i$ of $\mathbf{Y}$ for an example cone. Observe that a hyperplane through the corners separates all the points from the origin. This suggests that the normalized corners are the support vectors found by a one-class SVM:

$$\text{maximize} \quad b \quad \text{s.t.} \quad \mathbf{w}^T\mathbf{y}_i \geq b \text{ (for } i = 1, \dots, n) \text{ and } \|\mathbf{w}\|_2 \leq 1. \tag{1}$$

We show next that this intuition is correct. Define the following condition.

**Condition 1.** *The matrix $\mathbf{Y}_P$ satisfies $(\mathbf{Y}_P\mathbf{Y}_P^T)^{-1}\mathbf{1} > 0$.*

**Theorem 2.2.** *Each support vector selected by the one-class SVM (Eq. 1) is a corner of $\mathbf{Z}$. Also, if Condition 1 holds, there is at least one support vector for every corner.*

Thus, under Condition 1, we can get all the corners from the support vectors, and then find $\mathbf{M}$ via regression of $\mathbf{Z}$ on $\mathbf{Y}_P$. Condition 1 is always satisfied for our problem setting, as shown next.

**Theorem 2.3.** *Let $\mathbf{P}$ be a population matrix satisfying Assumption 2.1. Let $\mathbf{Z} = \mathbf{V}$, where $\mathbf{V}$ is the rank-$K$ eigenvector matrix. Let $\mathbf{Y} = \mathbf{NZ}$ as defined above. Then, Condition 1 is true.*

Thus, the ideal cone problem is easily solved by a one-class SVM. Next, we show that the same method suffices for the empirical cone problem too.

## 2.2 The Empirical Cone Problem

Now, instead of the normalized eigenvector rows $\mathbf{Y}$, we are given the empirical matrix $\hat{\mathbf{Y}}$ with rows $\hat{\mathbf{z}}_i^T/\|\hat{\mathbf{z}}_i\|$, where $\max_i \|\mathbf{e}_i^T(\hat{\mathbf{Y}} - \mathbf{Y})\| \leq \epsilon$. Once again, we focus on finding the corner indices, using which $\mathbf{M}$ can be inferred by regression. We will show that running a one-class SVM on the rows of $\hat{\mathbf{Y}}$ yields "near-corners," after some post-processing. We will need a stronger form of Condition 1:

**Condition 2.** *The matrix $\mathbf{Y}_P$ satisfies $(\mathbf{Y}_P\mathbf{Y}_P^T)^{-1}\mathbf{1} \geq \eta\mathbf{1}$ for some constant $\eta > 0$.*

It is easy to show that the solution $(\mathbf{w}, b)$ of the population SVM under Condition 1 is given by

$$\mathbf{w} = b^{-1} \cdot \mathbf{Y}_P^T\boldsymbol{\beta} \qquad b = \left(\mathbf{1}^T(\mathbf{Y}_P\mathbf{Y}_P^T)^{-1}\mathbf{1}\right)^{-1/2} \qquad \boldsymbol{\beta} = \frac{(\mathbf{Y}_P\mathbf{Y}_P^T)^{-1}\mathbf{1}}{\mathbf{1}^T(\mathbf{Y}_P\mathbf{Y}_P^T)^{-1}\mathbf{1}}. \tag{2}$$

Thus, Condition 1 implies that $\mathbf{w}$ is a convex combination of the corners, while Condition 2 additionally requires a minimum contribution from each corner.

**Lemma 2.4** (SVM solution is nearly ideal). *Let $(\hat{\mathbf{w}}, \hat{b})$ be the solution for the one-class SVM (Eq. 1) applied to the rows of $\hat{\mathbf{Y}}$. Under Condition 2, we have $|\hat{b} - b| \leq \epsilon$ and $\|\hat{\mathbf{w}} - \mathbf{w}\| \leq \zeta\epsilon$, for $\zeta = \frac{4}{\eta b^2\sqrt{\lambda_K(\mathbf{Y}_P\mathbf{Y}_P^T)}} \leq \frac{4K}{\eta(\lambda_K(\mathbf{Y}_P\mathbf{Y}_P^T))^{1.5}}$.*

Unlike the ideal cone scenario, the rows $\hat{\mathbf{Y}}_P$ corresponding to the corners need not be support vectors for the empirical cone. However, they are not far off.

**Lemma 2.5** (Corners are nearly support vectors). *The corners of the population cone are close to the supporting hyperplane: $\hat{b}\mathbf{1} \leq \hat{\mathbf{Y}}_P\hat{\mathbf{w}} \leq \hat{b}\mathbf{1} + (\zeta + 2)\epsilon\mathbf{1}$.*

This suggests that we should consider all points that are up to $(\zeta + 2)\epsilon$ away from the supporting hyperplane when searching for corners. The next Lemma shows that each such point is a "near-corner."

Recall that each row $\hat{\mathbf{y}}_i^T$ is a noisy version of a population row $\mathbf{y}_i^T = \mathbf{m}_i^T\mathbf{Y}_P/\|\mathbf{m}_i^T\mathbf{Y}_P\|$, which can be rewritten as a scaled convex combination of the normalized corners: $\mathbf{y}_i^T = r_i\boldsymbol{\phi}_i^T\mathbf{Y}_P$, where $\boldsymbol{\phi}_i^T\mathbf{1} = 1$. Specifically, $r_i = \frac{\mathbf{m}_i^T\mathbf{1}}{\|\mathbf{m}_i^T\mathbf{Y}_P\|}$ and $\boldsymbol{\phi}_i = \frac{\mathbf{m}_i}{\mathbf{m}_i^T\mathbf{1}}$. For a corner, $r_i = 1$ and $\boldsymbol{\phi}_i = \mathbf{e}_j$ for some $j$. We now show that every point $i$ that is close to the supporting hyperplane is nearly a corner of the ideal cone.

**Lemma 2.6** (Points close to support vectors are near-corners). *If $\hat{\mathbf{w}}^T\hat{\mathbf{y}}_i \leq \hat{b} + (\zeta + 2)\epsilon$ for some point $i \in [n]$, then $1 \leq r_i \leq 1 + \frac{(\zeta+4)\epsilon}{b-\epsilon}$ and $\phi_{ij} \geq 1 - \frac{2\zeta\epsilon}{b\lambda_K(\mathbf{Y}_P\mathbf{Y}_P^T)}$ for some $j \in [K]$.*

Consider the set of points $S_c = \{i \mid \hat{\mathbf{w}}^T \hat{\mathbf{y}}_i \leq \hat{b} + (\zeta + 2)\epsilon\}$ that are close to the supporting hyperplane. Lemmas 2.5 and 2.6 show that $S_c$ contains all corners, and possibly other points that are all near-corners. This suggests that we can cluster the vectors $\{\hat{\mathbf{y}}_i \mid i \in S\}$ into $K$ clusters, each corresponding to one corner and possibly extra near-corners close to that corner. Randomly selecting one point from each cluster gives us the set of inferred corners.

**Lemma 2.7** (Each corner has its own cluster). *There exist exactly $K$ clusters in $S_c$, as long as* $\epsilon \leq c_\epsilon \frac{\eta(\lambda_K(\mathbf{Y}_P \mathbf{Y}_P^T))^3}{K^{1.5}\sqrt{\kappa(\mathbf{Y}_P \mathbf{Y}_P^T)}}$, *for some global constant $c_\epsilon$.*

Let $C$ be the indices of the near-corners picked by this clustering step. Since $\mathbf{Z} = \mathbf{M}\mathbf{Y}_P$, this suggests $\mathbf{M}$ can be obtained via regression: $\mathbf{M} \approx \hat{\mathbf{Z}}\hat{\mathbf{Y}}_C^T(\hat{\mathbf{Y}}_C\hat{\mathbf{Y}}_C^T)^{-1}\mathbf{\Pi}$, where $\hat{\mathbf{Y}}_C := \hat{\mathbf{Y}}(C,:)$ and $\mathbf{\Pi}$ is a permutation matrix that matches the ordering of ideal corners and the empirical near-corners.

**Theorem 2.8.** *If Condition 2 and the condition on $\epsilon$ in Lemma 2.7 holds, then for any $i \in [n]$,* $\|\mathbf{e}_i^T(\mathbf{M} - \hat{\mathbf{Z}}\hat{\mathbf{Y}}_C^T(\hat{\mathbf{Y}}_C\hat{\mathbf{Y}}_C^T)^{-1}\mathbf{\Pi})\| \leq \frac{c_M \kappa(\mathbf{Y}_P \mathbf{Y}_P^T)\|\mathbf{e}_i^T \mathbf{Z}\|K\zeta}{(\lambda_K(\mathbf{Y}_P \mathbf{Y}_P^T))^{2.5}}\epsilon$, *where $c_M$ is a global constant, and $\kappa(.)$ is the ratio of the largest and smallest nonzero singular values of a matrix.*

Algorithm 1 shows all the steps of our method (SVM-cone). The algorithm requires an estimate of $\delta := (\zeta + 2)\epsilon$, and returns the inferred $\mathbf{M}$ and near-corners $C$. When the row-wise error bound $\epsilon$ is unknown, we can start with $\delta = 0$ and incrementally increase it until $K$ distinct clusters are found.

---

**Algorithm 1** SVM-cone

---

**Input:** $\hat{\mathbf{Z}} \in \mathbb{R}^{n \times m}$, number of corners $K$, estimated distance of corners from hyperplane $\delta$
**Output:** Estimated conic combination matrix $\hat{\mathbf{M}}$ and near-corner set $C$
 1: Normalize rows of $\hat{\mathbf{Z}}$ by $\ell_2$ norm to get $\hat{\mathbf{Y}}$ with rows $\hat{\mathbf{y}}_i^T$
 2: Run one-class SVM on $\hat{\mathbf{y}}_i$ to get the normal $\hat{\mathbf{w}}$ and distance $\hat{b}$ of the supporting hyperplane
 3: Cluster points $\{\hat{\mathbf{y}}_i \mid \hat{\mathbf{w}}^T \hat{\mathbf{y}}_i \leq \hat{b} + \delta\}$ that are close to the hyperplane into $K$ clusters
 4: Pick one point from each cluster to get near-corner set $C$
 5: $\hat{\mathbf{M}} = \hat{\mathbf{Z}}\hat{\mathbf{Y}}_C^T(\hat{\mathbf{Y}}_C\hat{\mathbf{Y}}_C^T)^{-1}$

---

# 3 Applications

Many network models and topic models have population matrices of the form $\mathbf{P} = \rho\mathbf{\Gamma}\mathbf{\Theta}\mathbf{B}\mathbf{\Theta}^T\mathbf{\Gamma}$. We have already shown that in such cases, the eigenvector matrix $\mathbf{V}$ forms an ideal cone (Lemma 2.1), and that Condition 1 holds. It is easy to see that the same holds for $\mathbf{V}\mathbf{V}^T$ as well. This suggests that SVM-cone can be applied to the matrix $\hat{\mathbf{V}}\hat{\mathbf{V}}^T$, where $\hat{\mathbf{V}}$ is the empirical top-$K$ eigenvector matrix. We shall show that this yields *per-node error bounds* in estimating community memberships and *per-word error bounds* for word-topic distributions.

## 3.1 Network models

Define a "DCMMSB-type" model as a model with population matrix $\mathbf{P} = \rho\mathbf{\Gamma}\mathbf{\Theta}\mathbf{B}\mathbf{\Theta}^T\mathbf{\Gamma}$ and an empirical adjacency matrix $\mathbf{A}$ with $\mathbf{A}_{ji} = \mathbf{A}_{ij} \sim \text{Bernoulli}(\mathbf{P}_{ij})$ for all $i > j$. Assume that rows of $\mathbf{\Theta}$ have unit $\ell_p$ norm, for $p = 1$ (DCMMSB) or $p = 2$ (OCCAM [32]). Let $\mathbf{v}_i = \mathbf{V}^T\mathbf{e}_i$, $\hat{\mathbf{v}}_i = \hat{\mathbf{V}}^T\mathbf{e}_i$, $\mathbf{y}_i = \mathbf{V}\mathbf{v}_i/\|\mathbf{V}\mathbf{v}_i\|$, and $\hat{\mathbf{y}}_i = \hat{\mathbf{V}}\hat{\mathbf{v}}_i/\|\hat{\mathbf{V}}\hat{\mathbf{v}}_i\|$. Denote $\gamma_{\max} = \max_i \mathbf{\Gamma}_{ii}$ and $\gamma_{\min} = \min_i \mathbf{\Gamma}_{ii}$.

**Theorem 3.1** (Small row-wise error in Network Models). *Consider a DCMMSB-type model with $\boldsymbol{\theta}_i \sim \text{Dirichlet}(\boldsymbol{\alpha})$, and $\alpha_0 := \boldsymbol{\alpha}^T\mathbf{1}$. If $\nu := \frac{\alpha_0}{\alpha_{\min}} \leq \frac{\min(\sqrt{\frac{n}{27\log n}}, \frac{\gamma_{\min}^2}{\gamma_{\max}^2}n\rho)}{2(1+\alpha_0)}$,* $\frac{\lambda^*(\mathbf{B})}{\nu} \geq \frac{8(1+\alpha_0)(\log n)^\xi}{\gamma_{\min}^2\sqrt{n\rho}}$ *for some constant $\xi > 1$, $\kappa(\mathbf{\Theta}^T\mathbf{\Gamma}^2\mathbf{\Theta}) = \Theta(1)$, and $\alpha_0 = O(1)$, then*

$$\epsilon = \max_i \|\mathbf{y}_i - \hat{\mathbf{y}}_i\| = \tilde{O}\left(\frac{\gamma_{\max}\min\{K^2, (\kappa(\mathbf{P}))^2\}K^{0.5}\nu(1+\alpha_0)}{\gamma_{\min}^3\lambda^*(\mathbf{B})\sqrt{n\rho}}\right)$$

*with probability at least $1 - O(Kn^{-2})$. Here $\lambda^*(\mathbf{B})$ is the smallest singular value of $\mathbf{B}$.*

Similar results for the non-Dirichlet case follow easily as long as $n\rho = \Omega((\log n)^{2\xi})$, $\lambda_K(\mathbf{P}) = \Omega(\sqrt{n\rho}(\log n)^{\xi})$, and $\max_i \|\mathbf{V}(:,i)\| = O(\sqrt{\rho})$ with high probability. This shows that the rows of $\hat{\mathbf{V}}\hat{\mathbf{V}}^T$ are close to those of $\mathbf{V}\mathbf{V}^T$, and the latter forms an ideal cone satisfying Condition 1. Hence, the conic combination for each node can be recovered by Algorithm 1 applied to $\hat{\mathbf{V}}\hat{\mathbf{V}}^T$. In fact, we can run the algorithm on $\hat{\mathbf{V}}$ itself; the output depends only on the SVM dual variables $\boldsymbol{\beta}$ (Eq. 2), which are the same whether the input is $\hat{\mathbf{V}}$ or $\hat{\mathbf{V}}\hat{\mathbf{V}}^T$. The output is the same conic combination matrix $\hat{\mathbf{M}}$ and the same set $C$ of nearly-pure nodes.

For **identifiability** of $\boldsymbol{\Theta}$, we need another condition. We will assume that $\sum \boldsymbol{\Gamma}_{ii} = n$ and all diagonal entries of $\mathbf{B}$ are equal (details are provided in the supplementary material). The next theorem shows that SVM-cone can be used to consistently infer the parameters of DCMMSB as well as OCCAM [32].

**Theorem 3.2** (Consistent inference of community memberships for each node). *Consider DCMMSB-type models where the conditions of Theorem 3.1 are satisfied and $\kappa(\boldsymbol{\Theta}^T\boldsymbol{\Gamma}^2\boldsymbol{\Theta}) = \Theta(1)$. Let $\hat{\mathbf{D}}$ be a diagonal matrix with entries $\hat{\mathbf{D}}_{ii} = \sqrt{\mathbf{e}_i^T \hat{\mathbf{Y}}_C \hat{\mathbf{V}} \hat{\mathbf{E}} \hat{\mathbf{V}}^T \hat{\mathbf{Y}}_C^T \mathbf{e}_i}$. Let $\hat{\boldsymbol{\Theta}} = \hat{\mathbf{F}}^{-1}\hat{\mathbf{M}}\hat{\mathbf{D}}$, where $\hat{\mathbf{F}}$ is a diagonal matrix with entries $\hat{\mathbf{F}}_{ii} = \|\mathbf{e}_i^T \hat{\mathbf{M}}\hat{\mathbf{D}}\|_1$ (for DCMMSB) and $\hat{\mathbf{F}}_{ii} = \|\mathbf{e}_i^T \hat{\mathbf{M}}\hat{\mathbf{D}}\|_2$ (for OCCAM [32]). Then there exists a permutation matrix $\boldsymbol{\Pi}$ such that*

$$\|\mathbf{e}_i^T(\boldsymbol{\Theta} - \hat{\boldsymbol{\Theta}}\boldsymbol{\Pi})\| = \tilde{O}\left( \frac{\gamma_{\max}K^{2.5}\min\{K^2, (\kappa(\mathbf{P}))^2\}n^{3/2}}{\gamma_{\min}\eta\lambda^*(\mathbf{B})\lambda_K^2(\boldsymbol{\Theta}^T\boldsymbol{\Gamma}^2\boldsymbol{\Theta})\sqrt{\rho}} \right)$$

*with probability at least $1 - O(Kn^{-2})$.*

*Remark* 3.1. The error bound is small when the clusters are well separated (large $\lambda^*(\mathbf{B})$), the network is dense (large $\rho$), there are few blocks (small $K$), and the membership vectors $\boldsymbol{\Theta}$ are drawn from a balanced Dirichlet distribution (small $\nu$, and hence small $\kappa(\mathbf{P})$), which leads to balanced block sizes.

*Remark* 3.2. For DCMMSB-type models, $\eta \geq \frac{\gamma_{\min}^2 \min_i(\mathbf{e}_i^T\boldsymbol{\Theta}^T\mathbf{1})}{\lambda_1(\boldsymbol{\Theta}^T\boldsymbol{\Gamma}^2\boldsymbol{\Theta})}$. Also, under the conditions of Theorem 3.1, $\eta \geq \frac{\gamma_{\min}^2}{3\nu\gamma_{\max}^2}$ with high probability. Proofs are in the supplementary material.

Observe that these are **per-node error bounds**, as against a simpler bound on $\|\boldsymbol{\Theta} - \hat{\boldsymbol{\Theta}}\|$. Clearly, the same results extend to the special case of the Mixed Membership Stochastic Blockmodel [1] and the Stochastic Blockmodel [13] as well (the assumption of equal diagonal entries of $\mathbf{B}$ is no longer needed, since $\boldsymbol{\Gamma}_{ii} = 1$ is enough for parameter identifiability [21]).

## 3.2 Topic Models

Let $\mathbf{T} \in \mathbb{R}_{\geq 0}^{V \times K}$ be a matrix of the word to topic probabilities with unit column sum, and let $\mathbf{H} \in \mathbb{R}_{\geq 0}^{K \times D}$ be the topic to document matrix. Then $\boldsymbol{\mathcal{A}} := \mathbf{TH}$ is the probability matrix for words appearing in documents. The actual counts of words in documents are assumed to be generated iid as $\mathbf{A}_{ij} \sim \text{Binomial}(N, \boldsymbol{\mathcal{A}}_{ij})$ for $i \in [V], j \in [D]$.

The word co-occurrence probability matrix is given by $\boldsymbol{\mathcal{A}}\boldsymbol{\mathcal{A}}^T/D = \mathbf{T}(\mathbf{HH}^T/D)\mathbf{T}^T$. Setting $\boldsymbol{\Gamma}_{ii} = \|\mathbf{T}(i,:)\|_1$, $\boldsymbol{\Theta} = \boldsymbol{\Gamma}^{-1}\mathbf{T}$, and $\mathbf{B} = \mathbf{HH}^T/D$, we find that $\boldsymbol{\mathcal{A}}\boldsymbol{\mathcal{A}}^T/D = \boldsymbol{\Gamma}\boldsymbol{\Theta}\mathbf{B}\boldsymbol{\Theta}^T\boldsymbol{\Gamma}$ with $\boldsymbol{\Theta}\mathbf{1} = \mathbf{1}$. This clearly matches the form of $\mathbf{P}$ in the DCMMSB model. Hence, its eigenvector matrix has the desired conic structure with weight matrix $\mathbf{M} = \mathbf{T}\boldsymbol{\Gamma}_P^{-1}\mathbf{N}_P^{-1}$, with the "pure nodes" being *anchor words* that only occur in a single topic. We now show that the row-wise error between the empirical and population eigenvector matrices decays with increasing number of documents $D$ and number of words in a document $N$.

**Assumption 3.1.** Let $g_{ik} = \mathbf{e}_i^T\boldsymbol{\mathcal{A}}\boldsymbol{\mathcal{A}}^T e_k$. We assume that when it is not zero, it goes to infinity, in particular, $g_{ik} \geq N\log\max(V, D)$, which gives $D/N \to \infty$. We also assume that $\lambda_i(\mathbf{HH}^T) = \Theta(D)$, for $i \in [K]$, and $\kappa(\mathbf{TT}^T) = \Theta(1)$.

These assumptions are similar to ones made in other theoretical literature on topic models [18].

We will construct a matrix $\mathbf{A}_1\mathbf{A}_2^T$, where $\mathbf{A}_1$ and $\mathbf{A}_2$ are obtained by dividing the words in each document uniformly randomly in two equal parts. This ensures that $\mathbb{E}[\mathbf{A}_1\mathbf{A}_2^T] = \boldsymbol{\mathcal{A}}\boldsymbol{\mathcal{A}}^T$, which in turn helps establishing concentration of empirical singular vectors as shown in the following lemma. For simplicity denote $N_1 = N/2$.

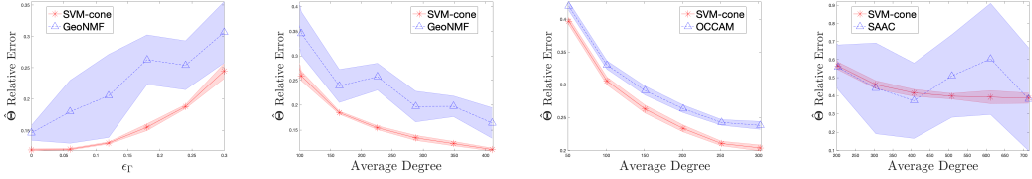

(a) Varying degree hetero-geneity for DCMMSB

(b) Varying sparsity for DCMMSB

(c) Varying sparsity for OCCAM model

(d) Varying sparsity for SBMO [17]

Figure 2: Relative error in estimation of community memberships: Plots (a) and (b) compare SVM-cone against the closest baseline (GeoNMF) on the degree-corrected MMSB model. We then compare against (c) OCCAM and (d) SAAC on networks drawn from their generative models.

**Lemma 3.3** (Small row-wise error in Topic Models). *Let $\hat{\mathbf{V}}$ denote the matrix of the top-$K$ singular vectors of $\mathbf{U} = \mathbf{A}_1\mathbf{A}_2^T/N_1^2$, and let the population counterpart of this be $\mathbf{V}$. Let $\mathbf{v}_i = \mathbf{V}^T\mathbf{e}_i$, $\hat{\mathbf{v}}_i = \hat{\mathbf{V}}^T\mathbf{e}_i$, $\mathbf{y}_i = \mathbf{V}\mathbf{v}_i/\|\mathbf{V}\mathbf{v}_i\|$, and $\hat{\mathbf{y}}_i = \hat{\mathbf{V}}\hat{\mathbf{v}}_i/\|\hat{\mathbf{V}}\hat{\mathbf{v}}_i\|$. Under Assumption 3.1, we have:*

$$\epsilon = \max_i \|\mathbf{y}_i - \hat{\mathbf{y}}_i\| = \frac{1}{\min_j \|\mathbf{e}_i^T\mathbf{T}\|_1\sqrt{\lambda_K(\mathbf{T}^T\mathbf{T})}}O_P\left(\sqrt{\frac{K\log\max(V,D)}{DN}}\right)$$

*with probability at least $1 - O(1/D^2)$.*

Thus, Algorithm 1 run on $\hat{\mathbf{V}}\hat{\mathbf{V}}^T$ (or equivalently, just $\hat{\mathbf{V}}$) can be used to find the conic combination weights $\hat{\mathbf{M}} \approx \mathbf{M}$. Since $\mathbf{M}$ being the product of $\mathbf{T}$ with a diagonal matrix where $\mathbf{T}$ has unit column sum, we can extract $\hat{\mathbf{T}} = \hat{\mathbf{M}}\hat{\mathbf{D}}^{-1}$, where $\hat{\mathbf{D}}$ is a diagonal matrix with $\hat{\mathbf{D}}_{ii} = \|\hat{\mathbf{M}}(:,i)\|_1$.

**Theorem 3.4** (Consistent inference of word-topic probabilities for each word). *Under Assumption 3.1, there exists a permutation matrix $\mathbf{\Pi}$ such that with probability at least $1 - O(1/D^2)$,*

$$\frac{\|\mathbf{e}_i^T(\hat{\mathbf{T}} - \mathbf{T}\mathbf{\Pi}^T)\|}{\|\mathbf{e}_i^T\mathbf{T}\|} = O_P\left(\frac{K^4\max_j\|\mathbf{e}_j^T\mathbf{T}\|_1}{\eta(\min_j\|\mathbf{e}_j^T\mathbf{T}\|_1)^2}\sqrt{\frac{\log\max(V,D)}{DN}}\right).$$

*Remark* 3.3. We have $\eta \geq \min_i \|\mathbf{e}_i^T\mathbf{T}\|_1/\lambda_1(\mathbf{T}^T\mathbf{T}) \geq \min_i \|\mathbf{e}_i^T\mathbf{T}\|_1/K$ (supplementary material).

## 4 Experiments

We ran experiments on simulated and real-world datasets to verify the accuracy and scalability of SVM-cone. We compared SVM-cone against several competing baselines. For network models, **GeoNMF** detects the corners of a simplex formed by the MMSB model by constructing the graph Laplacian and picking nodes that have large norms in the Laplacian [20]. It assumes balanced communities (i.e., the rows of $\mathbf{\Theta}$ are drawn from a Dirichlet with identical community weights). **SVI** uses stochastic variational inference for MMSB [12]. **BSNMF** [24] presents a Bayesian approach to Symmetric Nonnegative Matrix Factorization; it can be applied to do inference for MMSB models with $\mathbf{B} = c\mathbf{I}$ where $c \in [0,1]$. **OCCAM** works on a variant of MMSB where each row of $\mathbf{\Theta}$ has unit $\ell_2$ norm, and the model allows for degree heterogeneity [32]. **SAAC** [17] uses alternating optimization on a version of the stochastic blockmodel where each node can be a member of multiple communities, but the membership weight is binary. For topic models, **RecoverL2** [5] uses a combinatorial algorithm to pick anchor words from the word co-occurrence matrix and then recovers the word-topic vectors by optimizing a quadratic loss function. **TSVD** [7] uses a thresholded SVD based procedure to recover the topics. **GDM** [31] is a geometric algorithm that involves a weighted clustering procedure augmented with geometric corrections. We could not obtain the code for [18].

### 4.1 Networks with overlapping communities

In this section, we present experiments on simulated and large real networks.

#### 4.1.1 Simulations

We test the recovery of population parameters $(\mathbf{\Theta}, \mathbf{B})$ given adjacency matrices $\mathbf{A}$ generated from the corresponding population matrices $\mathbf{P}$ ($\mathbf{\Gamma}$ are nuisance parameters). We generate networks with

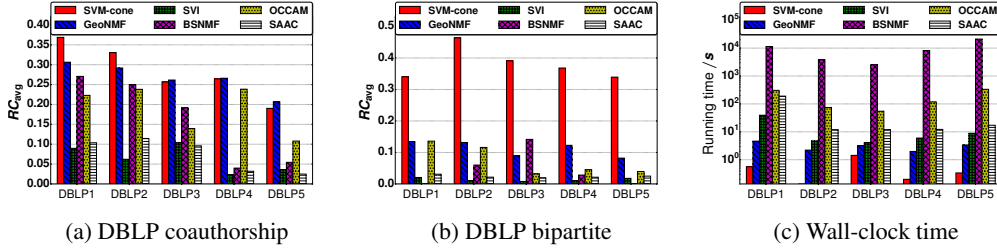

| (a) DBLP coauthorship | (b) DBLP bipartite | (c) Wall-clock time |

Figure 3: Accuracy of estimated community memberships for (a) the DBLP coauthorship network and (b) the biparite author-paper DBLP network. (c) The wall-clock time of the competing methods on the DBLP coauthorship network respectively.

$n = 5000$ nodes and $K = 3$ communities. The rows of $\Theta$ are drawn from Dirichlet($\alpha$) for DCMMSB and OCCAM; for DCMMSB, $\alpha = (1/3, 1/3, 1/3)$; for OCCAM, $\alpha = (1/6, 1/6, 1/6)$ and the rows are normalized to have unit $\ell_2$ norm. We set $\mathbf{B}_{ii} = 1$ and $\mathbf{B}_{ij} = 0.1$ for all $i \neq j$. The default degree parameters for DCMMSB are as follows: for all nodes $i$ that are predominantly in the $j$-th community ($\theta_{ij} > 0.5$), we set $\Gamma_{ii}$ to 0.3, 0.5, and 0.7 for the 3 respective communities; all other nodes have $\Gamma_{ii} = 1$. For OCCAM, we draw degree parameters from a $\mathrm{Beta}(1, 3)$ distribution.

**Varying degree parameters $\Gamma$:** We set the degree parameters for predominant nodes in the 3 communities as $0.5 + \epsilon_\Gamma$, $0.5$, and $0.5 + \epsilon_\Gamma$ respectively. Figure 2a shows SVM-cone outperforms GeoNMF consistently for all choices of $\epsilon_\Gamma$.

**Varying network sparsity $\rho$:** Figure 2b shows the relative error in estimating $\Theta$ as a function of the network sparsity $\rho$. Increasing $\rho$ increases the average degree of nodes in the network without affecting the skew induced by their degree parameters $\Gamma$. As expected, all methods tend to improve with increasing degree. Our method dominates GeoNMF over the entire range of average degrees. Figures 2c and 2d show results for networks generated under the models used by OCCAM and SAAC respectively. SVM-cone is comparable or better than these methods even on their generative models. The smaller error bars on SVM-cone show that it is more stable than SAAC.

### 4.1.2 Real-world experiments

We tested SVM-cone on large network datasets and word-document datasets. For networks, we used the 5 DBLP coauthorship networks[1] (used in [20], where each ground truth community corresponds to a group of conferences on the same topic. We also use bipartite author-paper variants for these 5 networks. Following [20], we evaluate results by the rank correlation between the predicted vector for community $i$ against the true vector, averaged over all communities: $RC_{avg}(\hat{\Theta}, \Theta) = \frac{1}{K} \max_\sigma \sum_{i=1}^K RC(\hat{\Theta}(:, i), \Theta(:, \sigma(i)))$, where $\sigma$ is a permutation over the $K$ communities. We have $-1 \leq RC_{avg}(\hat{\Theta}, \Theta) \leq 1$, with higher numbers implying a better match between $\hat{\Theta}$ and $\Theta$. We do not use metrics like NMI [30] or ExNVI [32] that require binary overlapping membership vectors to avoid thresholding issues on real-valued membership vectors.

We find that SVM-cone outperforms competing baselines on 2 of the 5 DBLP coauthorship datasets, and is similar on the remaining three (Figure 3a). The closest competitor is GeoNMF [20], which assumes that all nodes have the same degree parameter, and the community sizes are balanced. Both assumptions are reasonable for the dataset, since the number of coauthors (the degree) does not vary significantly among authors, and the communities are formed from conferences where no one conference dominates the others. The differences between SVM-cone and the competition is starker on the bipartite dataset (Figure 3b). There is severe degree heterogeneity: an author can be connected to many papers, while each paper only has a few authors at best. Our method is able to accommodate such differences between the nodes, and hence yields much better accuracy than others.

Finally, Figure 3c shows the wall-clock time for running the various methods on DBLP coauthorship networks (wall-clock time on DBLP bipartite author-paper networks is included in the supplementary material). Our method is among the fastest. This is expected; the only computationally intensive step is the one-class SVM and top-$K$ eigen-decomposition (or SVD), for which off-the-shelf efficient and scalable implementations already exist [10].

### 4.2 Topic Models

We generate semi-synthetic data following [5] and [7] using **NIPS**[1], **New York Times**[1] (NYT), **PubMed**[1], and **20NewsGroup**[2] (20NG). Dataset statistics are included in the supplementary material. We use Matlab R2018a built-in Gibbs Sampling function for learning topic models to learn the word by topic matrix, which should retain the characteristics of real data distributions. Then we draw the topic-document matrix from Dirichlet with symmetric hyper-parameter 0.01. We set $K = 40$ for the first 3 datasets and $K = 20$ for 20NG. The word counts matrix is sampled with $N = 1000, 300, 100, 200$ respectively, which matches the mean document length of the real datasets. We evaluate the performance of different algorithms using $\ell_1$ reconstruction error $\frac{1}{K} \sum_{i,j} |\mathbf{T}(i,j) - \hat{\mathbf{T}}(i, \pi(j))|$, where $\pi(.)$ is a permutation function that matches the topics. Table 1 shows the $\ell_1$ reconstruction error and wall-clock running time of different algorithms with datasets generated from different number of documents. Each setting is repeated 5 times, and we report the mean and standard deviation of the results. SVM-cone is much faster than the other methods. Its accuracy is comparable to RecoverL2, and significantly better than TSVD and GDM. The supplementary material also shows the top-10 words of 5 topics learned from SVM-cone for each dataset.

Table 1: $\ell_1$ reconstruction error and wall-clock time on semi-synthetic datasets

| Corpus | Documents | | RecoverL2 | TSVD | GDM | SVM-cone |
|---|---|---|---|---|---|---|
| NIPS | 20000 | $\ell_1$ Error | **0.059** (**± 0.000**) | 0.237 (± 0.017) | 0.081 (± 0.057) | 0.071 (± 0.004) |
| | | Time/$s$ | 100.11 (± 8.81) | 18.54 (± 2.04) | 119.66 (± 4.41) | **5.33 (± 0.39)** |
| | 40000 | $\ell_1$ Error | **0.043 (± 0.000)** | 0.250 (± 0.045) | 0.061 (± 0.038) | 0.051 (± 0.002) |
| | | Time/$s$ | 143.34 (± 0.53) | 21.97 (± 1.49) | 220.92 (± 3.10) | **9.07 (± 0.00)** |
| | 60000 | $\ell_1$ Error | **0.036 (± 0.000)** | 0.269 (± 0.064) | 0.059 (± 0.038) | 0.041 (± 0.002) |
| | | Time/$s$ | 247.34 (± 20.84) | 35.77 (± 3.28) | 406.87 (± 36.57) | **17.63 (± 5.29)** |
| NYT | 20000 | $\ell_1$ Error | **0.125 (± 0.000)** | 0.207 (± 0.025) | 0.223(± 0.008) | 0.131 (± 0.003) |
| | | Time/$s$ | 78.15 (± 7.14) | 25.11 (± 6.39) | 193.43 (± 12.02) | **4.51 (± 0.70)** |
| | 40000 | $\ell_1$ Error | **0.103 (± 0.000)** | 0.197 (± 0.045) | 0.216 (± 0.010) | 0.106 (± 0.001) |
| | | Time/$s$ | 140.84 (± 15.50) | 50.18 (± 13.14) | 394.16 (± 30.42) | **8.04 (± 1.15)** |
| | 60000 | $\ell_1$ Error | **0.095 (± 0.000)** | 0.166 (± 0.028) | 0.210 (± 0.010) | 0.096 (± 0.002) |
| | | Time/$s$ | 184.69 (± 20.65) | 42.96 (± 7.95) | 595.54 (± 91.57) | **11.82 (± 1.91)** |
| PubMed | 20000 | $\ell_1$ Error | **0.163 (± 0.000)** | 0.239 (± 0.032) | 0.277 (± 0.051) | 0.181 (± 0.002) |
| | | Time/$s$ | 54.32 (± 5.94) | 15.75 (± 2.34) | 205.95 (± 11.27) | **2.06 (± 0.46)** |
| | 40000 | $\ell_1$ Error | **0.122 (± 0.000)** | 0.255 (± 0.018) | 0.251 (± 0.041) | 0.138 (± 0.001) |
| | | Time/$s$ | 78.99 (± 9.99) | 26.44 (± 4.49) | 459.17 (± 30.71) | **3.73 (± 0.37)** |
| | 60000 | $\ell_1$ Error | **0.098 (± 0.000)** | 0.275 (± 0.041) | 0.269 (± 0.052) | 0.114 (± 0.001) |
| | | Time/$s$ | 98.19 (± 15.06) | 24.57 (± 4.59) | 649.97 (± 26.48) | **5.44 (± 0.38)** |
| 20NG | 20000 | $\ell_1$ Error | 0.100 (± 0.000) | 0.111 (± 0.051) | 0.137 (± 0.001) | **0.090 (± 0.003)** |
| | | Time/$s$ | 40.74 (± 0.64) | 7.51 (± 0.42) | 102.86 (± 4.05) | **1.85 (± 0.26)** |
| | 40000 | $\ell_1$ Error | 0.074 (± 0.000) | 0.081 (± 0.043) | 0.131 (± 0.072) | **0.064 (± 0.001)** |
| | | Time/$s$ | 94.42 (± 9.92) | 16.04 (± 2.28) | 273.51 (± 16.45) | **4.33 (± 0.71)** |
| | 60000 | $\ell_1$ Error | 0.058 (± 0.000) | 0.133 (± 0.045) | 0.096 (± 0.063) | **0.052 (± 0.002)** |
| | | Time/$s$ | 142.34 (± 20.31) | 23.36 (± 5.85) | 388.47 (± 43.22) | **5.89 (± 0.67)** |

## 5 Conclusions

We showed that many distinct models for overlapping clustering can be placed under one general framework, where the data matrix is a noisy version of an ideal matrix and each row is a non-negative weighted sum of "exemplars." In other words, the connection probabilities of one node to others in a network is a non-negative combination of the connection probabilities of $K$ "pure" nodes to others in the network. Each pure node is an examplar of a single community, and we require one pure node from each of the $K$ communities. This geometrically corresponds to a cone, with the pure nodes being its corners. This subsumes Mixed-Membership Stochastic Blockmodels and their degree-corrected variants, as well as commonly used topic models. We showed that a one-class SVM applied to the normalized rows of the data matrix can find both the corners and the weight matrix. We proved the consistency of our SVM-cone algorithm, and used it to develop consistent parameter inference methods for several widely used network and topic models. Experiments on simulated and large real-world datasets show both the accuracy and the scalability of SVM-cone.

[1]`https://archive.ics.uci.edu/ml/datasets/Bag+of+Words`
[2]`http://qwone.com/~jason/20Newsgroups/`

## Acknowledgments

X.M. and P.S. were partially supported by NSF grant DMS 1713082. D.C. was partially supported by a Facebook Faculty Research Award.

## Footnotes

[1] http://www.cs.utexas.edu/~xmao/coauthorship

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
