[Supplementary Material]

# Supplementary Material for "Overlapping Clustering Models, and One (class) SVM to Bind Them All"

**Xueyu Mao, Purnamrita Sarkar, Deepayan Chakrabarti**
The University of Texas at Austin
xmao@cs.utexas.edu, purna.sarkar@austin.utexas.edu, deepay@utexas.edu

In this document, we will use $I$ to denote the set of corner indices.

## A   Geometric structure of normalized points from a cone

**Lemma A.1.** *Let* $\mathbf{y}_i = \mathbf{z}_i/\|\mathbf{z}_i\|$*, then* $\mathbf{y}_i^T = r_i \boldsymbol{\phi}_i^T \mathbf{Y}_P$ *for* $r_i = \frac{\boldsymbol{m}_i^T \mathbf{1}}{\|\mathbf{m}_i^T \mathbf{Y}_P\|} \geq 1$*, and* $\boldsymbol{\phi}_i = (\phi_{i1}, \phi_{i2}, \cdots, \phi_{iK})^T$*,* $\phi_{ij} = \frac{m_{ij}}{\sum_j m_{ij}}$*.*

*Proof.* $\mathbf{y}_i^T = \frac{\mathbf{z}_i^T}{\|\mathbf{z}_i\|} = \frac{\mathbf{m}_i^T \mathbf{Y}_P}{\|\mathbf{m}_i^T \mathbf{Y}_P\|} = \frac{\boldsymbol{m}_i^T \mathbf{1}}{\|\mathbf{m}_i^T \mathbf{Y}_P\|} \frac{\mathbf{m}_i^T}{\boldsymbol{m}_i^T \mathbf{1}} \mathbf{Y}_P = r_i \boldsymbol{\phi}_i^T \mathbf{Y}_P$. Clearly $\|\mathbf{m}_i^T \mathbf{Y}_P\| = \|\sum_j m_{ij} \mathbf{y}_{I(j)}\| \leq \sum_j m_{ij} \|\mathbf{y}_{I(j)}\| = \sum_j m_{ij} = \boldsymbol{m}_i^T \mathbf{1}$, so $r_i \geq 1$. $\qquad\square$

*Proof of Lemma 2.1.* Since $\mathrm{rank}(\mathbf{P}) = K$, we have $\mathbf{V}\mathbf{E}\mathbf{V}^T = \mathbf{P} = \rho \boldsymbol{\Gamma}\boldsymbol{\Theta}\mathbf{B}\boldsymbol{\Theta}^T\boldsymbol{\Gamma}$. W.L.O.G, let $\boldsymbol{\Theta}(I,:) = \mathbf{I}$, then $\mathbf{V}_P \mathbf{E}\mathbf{V}^T = \rho \boldsymbol{\Gamma}_P \mathbf{B}\boldsymbol{\Theta}^T\boldsymbol{\Gamma}$. Now $\mathbf{V}\mathbf{E} = \mathbf{P}\mathbf{V} = \rho \boldsymbol{\Gamma}\boldsymbol{\Theta}\mathbf{B}\boldsymbol{\Theta}^T\boldsymbol{\Gamma}\mathbf{V} = \boldsymbol{\Gamma}\boldsymbol{\Theta}(\rho\mathbf{B}\boldsymbol{\Theta}^T\boldsymbol{\Gamma})\mathbf{V} = \boldsymbol{\Gamma}\boldsymbol{\Theta}(\boldsymbol{\Gamma}_P^{-1}\mathbf{V}_P\mathbf{E}\mathbf{V}^T)\mathbf{V} = \boldsymbol{\Gamma}\boldsymbol{\Theta}\boldsymbol{\Gamma}_P^{-1}\mathbf{V}_P\mathbf{E}$, right multiplying $\mathbf{E}^{-1}$ gives $\mathbf{V} = \boldsymbol{\Gamma}\boldsymbol{\Theta}\boldsymbol{\Gamma}_P^{-1}\mathbf{V}_P$. Also consider that $\mathbf{V}_P \mathbf{E}\mathbf{V}_P^T = \rho \boldsymbol{\Gamma}_P \mathbf{B}\boldsymbol{\Gamma}_P$, $\mathbf{V}_P$ is full rank. $\qquad\square$

## B   Identifiability of DCMMSB-type Models

**Lemma B.1.** *For DCMMSB-type models such that* $f(\boldsymbol{\theta}_i) = 1, \forall i \in [n]$ *for some degree 1 homogeneous function $f$ (e.g., $f(\boldsymbol{\theta}) = \|\boldsymbol{\theta}\|_p$), the sufficient conditions for $(\boldsymbol{\Theta}, \mathbf{B}, \boldsymbol{\Gamma})$ to be identifiable up to a permutation of the communities are (a) there is at least one pure node in each community, (b) $\sum_i \gamma_i = n$, (c) $\mathbf{B}$ has unit diagonal.*

*Proof.* From Lema 2.1 we have $\mathbf{V} = \boldsymbol{\Gamma}\boldsymbol{\Theta}\boldsymbol{\Gamma}_P^{-1}\mathbf{V}_P$ and $\mathbf{V}_P$ is full rank. Suppose two set of parameters $\{\boldsymbol{\Gamma}^{(1)}, \boldsymbol{\Theta}^{(1)}, \mathbf{B}^{(1)}\}$ and $\{\boldsymbol{\Gamma}^{(2)}, \boldsymbol{\Theta}^{(2)}, \mathbf{B}^{(2)}\}$ yield the same $\mathbf{P}$ (W.L.O.G., we abort $\rho$ in $\mathbf{B}$) and each has a pure node set $P_1$ and $P_2$ and W.L.O.G., assume the permutation of the communities is fixed, i.e., $\boldsymbol{\Theta}_{P_1}^{(1)} = \boldsymbol{\Theta}_{P_2}^{(2)} = \mathbf{I}$. Then,

$$\boldsymbol{\Gamma}^{(1)}\boldsymbol{\Theta}^{(1)}(\boldsymbol{\Gamma}_{P_1}^{(1)})^{-1}\mathbf{V}_{P_1} = \mathbf{V} = \boldsymbol{\Gamma}^{(2)}\boldsymbol{\Theta}^{(2)}(\boldsymbol{\Gamma}_{P_2}^{(2)})^{-1}\mathbf{V}_{P_2}. \tag{3}$$

Taking indices $P_1$ and $P_2$ respectively on $\mathbf{V}$, we have,

$$\mathbf{V}_{P_1} = \boldsymbol{\Gamma}_{P_1}^{(2)}\boldsymbol{\Theta}_{P_1}^{(2)}(\boldsymbol{\Gamma}_{P_2}^{(2)})^{-1}\mathbf{V}_{P_2} \quad\text{and}\quad \mathbf{V}_{P_2} = \boldsymbol{\Gamma}_{P_2}^{(1)}\boldsymbol{\Theta}_{P_2}^{(1)}(\boldsymbol{\Gamma}_{P_1}^{(1)})^{-1}\mathbf{V}_{P_1}. \tag{4}$$

Then,

$$\begin{aligned}
\mathbf{V}_{P_1} &= \boldsymbol{\Gamma}_{P_1}^{(2)}\boldsymbol{\Theta}_{P_1}^{(2)}(\boldsymbol{\Gamma}_{P_2}^{(2)})^{-1}\boldsymbol{\Gamma}_{P_2}^{(1)}\boldsymbol{\Theta}_{P_2}^{(1)}(\boldsymbol{\Gamma}_{P_1}^{(1)})^{-1}\mathbf{V}_{P_1} \\
\implies \quad \mathbf{I} &= \boldsymbol{\Gamma}_{P_1}^{(2)}\boldsymbol{\Theta}_{P_1}^{(2)}(\boldsymbol{\Gamma}_{P_2}^{(2)})^{-1}\boldsymbol{\Gamma}_{P_2}^{(1)}\boldsymbol{\Theta}_{P_2}^{(2)}(\boldsymbol{\Gamma}_{P_1}^{(1)})^{-1}, \quad\text{as } \mathbf{V}_{P_1} \text{ is full rank.}
\end{aligned} \tag{5}$$

As $\mathbf{\Gamma}_{P_1}^{(2)}\mathbf{\Theta}_{P_1}^{(2)}(\mathbf{\Gamma}_{P_2}^{(2)})^{-1}$ and $\mathbf{\Gamma}_{P_2}^{(1)}\mathbf{\Theta}_{P_2}^{(1)}(\mathbf{\Gamma}_{P_1}^{(1)})^{-1}$ are all nonnegative, using Lemma 1.1 of [6], they are both generalized permutation matrices. Also since $\mathbf{\Gamma}_{P_1}^{(2)}$, $(\mathbf{\Gamma}_{P_2}^{(2)})^{-1}$ are diagonal matrix, $\mathbf{\Theta}_{P_1}^{(2)}$ must be a permutation matrix as $f(\boldsymbol{\theta}_i^{(2)}) = 1, \forall i \in [n]$, and $f$ is homogeneous with degree 1. So nodes in $P_1$ are also pure nodes in $\mathbf{\Theta}^{(2)}$. With same arguments, nodes in $P_2$ are also pure nodes in $\mathbf{\Theta}^{(1)}$. So the pure nodes match up.

Now since $\mathbf{V}_P\mathbf{E}\mathbf{V}_P^T = \mathbf{\Gamma}_P\mathbf{B}\mathbf{\Gamma}_P$, we have $\mathbf{\Gamma}_{P_1}^{(1)}\mathbf{B}^{(1)}\mathbf{\Gamma}_{P_1}^{(1)} = \mathbf{V}_{P_1}\mathbf{E}\mathbf{V}_{P_1} = \mathbf{\Gamma}_{P_1}^{(2)}\mathbf{B}^{(2)}\mathbf{\Gamma}_{P_1}^{(2)}$. As $\mathbf{B}^{(1)}$ and $\mathbf{B}^{(2)}$ both have unit diagonal, we must have $\mathbf{\Gamma}_{P_1}^{(1)} = c\mathbf{\Gamma}_{P_1}^{(2)}$ for $c = \sqrt{\mathbf{B}_{11}^{(2)}/\mathbf{B}_{11}^{(1)}}$. Now substituting $P_2$ with $P_1$ in Eq. (3), and using $\mathbf{V}_{P_1}$ has full rank, we have,

$$\mathbf{\Gamma}^{(2)}\mathbf{\Theta}^{(2)}(\mathbf{\Gamma}_{P_1}^{(2)})^{-1} = \mathbf{\Gamma}^{(1)}\mathbf{\Theta}^{(1)}(\mathbf{\Gamma}_{P_1}^{(1)})^{-1} = \mathbf{\Gamma}^{(1)}\mathbf{\Theta}^{(1)}(\mathbf{\Gamma}_{P_1}^{(2)})^{-1}/c,$$

which gives $\mathbf{\Gamma}^{(1)}\mathbf{\Theta}^{(1)} = c\mathbf{\Gamma}^{(2)}\mathbf{\Theta}^{(2)}$, applying $f(\cdot)$ to rows' transpose on both side, since $f(\boldsymbol{\theta}_i^{(1)}) = f(\boldsymbol{\theta}_i^{(2)}) = 1, \forall i \in [n]$, and $f$ is homogeneous with degree 1, we have $\mathbf{\Gamma}^{(1)} = c\mathbf{\Gamma}^{(2)}$. Now as $\mathbf{1}_n^T\mathbf{\Gamma}^{(1)}\mathbf{1}_n = \mathbf{1}_n^T\mathbf{\Gamma}^{(2)}\mathbf{1}_n = n$ from condition (b), we must have $c = 1$, then $\mathbf{\Gamma}^{(1)} = \mathbf{\Gamma}^{(2)}$, and this immediately gives $\mathbf{\Theta}^{(1)} = \mathbf{\Theta}^{(2)}$. Finally, $\mathbf{\Gamma}_{P_1}^{(1)}\mathbf{B}^{(1)}\mathbf{\Gamma}_{P_1}^{(1)} = \mathbf{\Gamma}_{P_1}^{(2)}\mathbf{B}^{(2)}\mathbf{\Gamma}_{P_1}^{(2)} = \mathbf{\Gamma}_{P_1}^{(1)}\mathbf{B}^{(2)}\mathbf{\Gamma}_{P_1}^{(1)}$, and this gives $\mathbf{B}^{(1)} = \mathbf{B}^{(2)}$. $\qquad\square$

## C  Algorithms

In this section we provide the detailed algorithms for parameter estimations of DCMMSB, OCCAM (Algorithm A) and Topic Models (Algorithm B). These algorithms both reply on the one class SVM (Algorithm 1) for finding the corner rays and then use those for parameter estimation, the details of which vary from model to model. Note for Algorithm A, step 7 is to normalize rows of $\mathbf{\Theta}$ by $\ell_1$ norm, if we normalize by $\ell_2$ norm, then it can be used for estimation of OCCAM.

---

**Algorithm A** SVM-cone-DCMMSB

---

**Input:** Adjacency matrix $\mathbf{A} \in \mathbb{R}^{n \times n}$, number of communities $K$
**Output:** Estimated degree parameters $\hat{\mathbf{\Gamma}}$, community memberships $\hat{\mathbf{\Theta}}$, and community interaction matrix $\hat{\mathbf{B}}$
 1: Get top-$K$ eigen-decomposition of $\mathbf{A}$ as $\hat{\mathbf{V}}\hat{\mathbf{E}}\hat{\mathbf{V}}^T$
 2: Normalize rows of $\hat{\mathbf{V}}$ by $\ell_2$ norm
 3: Use SVM-cone to get pure node set $C$ and estimated $\hat{\mathbf{M}}$
 4: $\hat{\mathbf{V}}_C = \hat{\mathbf{V}}(C,:)$, get $\hat{\mathbf{N}}_C$ from row norms of $\hat{\mathbf{V}}_C$
 5: $\hat{\mathbf{D}} = \sqrt{\text{diag}(\hat{\mathbf{N}}_C\hat{\mathbf{V}}_C\hat{\mathbf{E}}\hat{\mathbf{V}}_C^T\hat{\mathbf{N}}_C)}$
 6: $\hat{\mathbf{F}} = \text{diag}(\hat{\mathbf{M}}\hat{\mathbf{D}}\mathbf{1}_K)$
 7: $\hat{\mathbf{\Theta}} = \hat{\mathbf{F}}^{-1}\hat{\mathbf{M}}\hat{\mathbf{D}}$
 8: $\hat{\mathbf{\Gamma}} = n\hat{\mathbf{F}}/(\mathbf{1}_n^T\hat{\mathbf{F}}\mathbf{1})$, $\hat{\mathbf{\Gamma}}_C = \hat{\mathbf{\Gamma}}(C,C)$
 9: $\mathbf{B} = \hat{\mathbf{\Gamma}}_C^{-1}\hat{\mathbf{V}}_C\hat{\mathbf{E}}\hat{\mathbf{V}}_C\hat{\mathbf{\Gamma}}_C^{-1}$
10: $\mathbf{B} = \mathbf{B}/\max_{i,j}\mathbf{B}_{ij}$

---

---

**Algorithm B** SVM-cone-topic

---

**Input:** Word-document count matrix $\mathbf{A} \in \mathbb{R}^{V \times D}$, number of topics $K$
**Output:** Estimated word-topic matrix $\hat{\mathbf{T}}$
 1: Randomly splitting the words in each document to two halves to get $\mathbf{A}_1$ and $\mathbf{A}_2$
 2: Normalize columns of $\mathbf{A}_1$ and $\mathbf{A}_2$ by $\ell_1$ norm to get $\hat{\mathbf{A}}_1$ and $\hat{\mathbf{A}}_2$
 3: Get top-$K$ SVD of $\mathbf{U} = \hat{\mathbf{A}}_1\hat{\mathbf{A}}_2^T$ as $\hat{\mathbf{V}}\hat{\mathbf{E}}\hat{\mathbf{V}}^T$
 4: Normalize rows of $\hat{\mathbf{V}}$ by $\ell_2$ norm
 5: Use SVM-cone to get pure node set $C$ and estimated $\hat{\mathbf{M}}$
 6: Normalizing columns of $\hat{\mathbf{M}}$ by $\ell_1$ norm to get $\hat{\mathbf{T}}$

---

# D Corner finding with One-class SVM with population inputs

**Lemma D.1.** *If* $\mathrm{Proj}_{\mathrm{Conv}(\mathbf{Y}_P^T)}(\mathbf{0})$ *is an interior point in* $\mathrm{Conv}(\mathbf{Y}_P^T)$, *then One-class SVM can find all the $K$ corners with $m_{ij} = 1$ as support vectors given $\mathbf{y}_i$, $i \in [n]$ as inputs. And a sufficient condition for this to hold is* $(\mathbf{Y}_P \mathbf{Y}_P^T)^{-1} \mathbf{1} > \mathbf{0}$.

*Proof.* The primal problem of One-class SVM in [7] is

$$\min \quad \frac{1}{2}\|\mathbf{w}\|^2 - b \qquad s.t. \quad \mathbf{w}^T \mathbf{y}_i \geq b, \ i \in [n].$$

First of all note that $b \geq 0$ because if $b < 0$, we can always make $b = 0$ to satisfy the condition and decrees the value of the object function. From Lemma A.1, we have $\mathbf{y}_i^T = r_i \boldsymbol{\phi}_i^T \mathbf{Y}_P$. As $r_i \geq 1$, if there exists $(\mathbf{w}, b)$ that $\mathbf{w}^T \mathbf{y}_i \geq b$, $i \in I$, we have $\mathbf{w}^T \mathbf{y}_i = r_i \boldsymbol{\phi}_i^T \mathbf{Y}_P \mathbf{w} = r_i \sum_j \phi_{ij} \mathbf{w}^T \mathbf{y}_{I(j)} \geq r_i b \geq b$, $i \in [n]$. So we can reduce the problem to using points $i \in I$ as inputs. Furthermore, we consider an equivalent primal problem and its dual:

$$\text{Primal}: \quad \max \quad b \qquad\qquad \text{Dual}: \quad \min \quad \frac{1}{2}\sum_{i,j} \beta_i \beta_j \mathbf{y}_i^T \mathbf{y}_j \qquad (6)$$

$$s.t. \quad \|\mathbf{w}\| \leq 1, \ \mathbf{w}^T \mathbf{y}_i \geq b, \ i \in I \qquad\qquad s.t. \quad \sum_i \beta_i = 1, \ \beta_i \geq 0, \ i \in I$$

The dual problem is basically to find a point in $\mathrm{Conv}(\mathbf{Y}_P^T)$ that has the minimum norm (closest to origin). Now denote the optimal function value for the dual problem as $L_{\mathbf{Y}_P}$ and for any subset $\mathcal{S} \subset I$, let $L_{\mathbf{Y}_{P(\mathcal{S},:)}}$ be the optimal value when we want to find a point in $\mathrm{Conv}(\mathbf{Y}_{P(\mathcal{S},:)}^T)$ that has the minimum norm.

Let $\mathbf{N} \in \mathbb{R}^{n \times n}$ be a diagonal matrix such that $\mathbf{N}_{ii} = 1/\|\mathbf{z}_i\|$, then $\mathbf{Y}_P = \mathbf{N}_P \mathbf{Z}_P$ is also full rank. If for $\boldsymbol{\beta}^* = \arg\min_{\boldsymbol{\beta}} L_{\mathbf{Y}_P}(\boldsymbol{\beta})$, each coordinate is strictly larger than 0, it is easy to see that $L_{\mathbf{Y}_P} > L_{\mathbf{Y}_{P(\mathcal{S},:)}}$ since $\mathbf{Y}_P$ is full rank. So a sufficient condition for One-class SVM to find all $K$ corners of $L_{\mathbf{Y}_P}$ is $\boldsymbol{\beta}^* > \mathbf{0}$, which means the closet point to origin in $\mathrm{Conv}(\mathbf{Y}_P^T)$ is an interior point (also the projection of origin to $\mathrm{Conv}(\mathbf{Y}_P^T)$ ). Now we will show a sufficient condition for this.

Suppose the $\boldsymbol{\beta}^* > \mathbf{0}$. First let us find a hyperplane $(\mathbf{w}, d)$ that is through columns of $\mathbf{Y}_P^T$ with $d < 0$ (since $\mathbf{Y}_P$ is full rank, we must have $d \neq 0$). We have $\mathbf{Y}_P \mathbf{w} = d\mathbf{1}$. Since the distance from origin to hyperplane $(\mathbf{w}, d)$ is $\frac{|d|}{\|\mathbf{w}\|}$, $\mathrm{Proj}_{\mathrm{Conv}(\mathbf{Y}_P^T)}(\mathbf{0})$ is an interior point in $\mathrm{Conv}(\mathbf{Y}_P^T)$, we have

$$\mathbf{Y}_P^T \boldsymbol{\beta}^* = \mathrm{Proj}_{\mathrm{Conv}(\mathbf{Y}_P^T)}(\mathbf{0}) = \frac{d}{\|\mathbf{w}\|}\frac{\mathbf{w}}{\|\mathbf{w}\|} \qquad (7)$$

Then,

$$\mathbf{w}^T \mathbf{Y}_P^T \boldsymbol{\beta}^* = \frac{d\mathbf{w}^T \mathbf{w}}{\|\mathbf{w}\|^2} = d.$$

As $\mathbf{w}^T \mathbf{Y}_P^T = d\mathbf{1}^T$, we have $d\mathbf{1}^T \boldsymbol{\beta}^* = d$, so $\mathbf{1}^T \boldsymbol{\beta}^* = 1$. So the only condition left to be satisfied is that $\boldsymbol{\beta}^* > \mathbf{0}$, using Eq. (7),

$$\mathbf{Y}_P \mathbf{Y}_P^T \boldsymbol{\beta}^* = \frac{d\mathbf{Y}_P \mathbf{w}}{\|\mathbf{w}\|^2} = \frac{d(d\mathbf{1})}{\|\mathbf{w}\|^2},$$

so $\boldsymbol{\beta}^* = \frac{d^2}{\|\mathbf{w}\|^2}(\mathbf{Y}_P \mathbf{Y}_P^T)^{-1}\mathbf{1} > \mathbf{0}$ and all we require is:

$$(\mathbf{Y}_P \mathbf{Y}_P^T)^{-1}\mathbf{1} > \mathbf{0}.$$

$\square$

*Proof of Theorem 2.3.* Using Lemma 2.1, we have:

$$\mathbf{I} = \mathbf{V}^T \mathbf{V} = \mathbf{V}_P^T \boldsymbol{\Gamma}_P^{-1} \boldsymbol{\Theta}^T \boldsymbol{\Gamma}^2 \boldsymbol{\Theta} \boldsymbol{\Gamma}_P^{-1} \mathbf{V}_P \implies (\mathbf{V}_P \mathbf{V}_P^T)^{-1} = \boldsymbol{\Gamma}_P^{-1} \boldsymbol{\Theta}^T \boldsymbol{\Gamma}^2 \boldsymbol{\Theta} \boldsymbol{\Gamma}_P^{-1}. \qquad (8)$$

Since $\mathbf{Y}_P = \mathbf{N}_P \mathbf{V}_P$, we have:

$$(\mathbf{Y}_P \mathbf{Y}_P^T)^{-1} = \mathbf{N}_P^{-1} \mathbf{\Gamma}_P^{-1} \mathbf{\Theta}^T \mathbf{\Gamma}^2 \mathbf{\Theta} \mathbf{\Gamma}_P^{-1} \mathbf{N}_P^{-1}. \tag{9}$$

On the RHS of Eq. (9), as $\mathbf{N}_P^{-1}$, $\mathbf{\Gamma}_P^{-1}$ and $\mathbf{\Gamma}$ are all diagonal matrix with strictly positive diagonal elements, then diagonal of $(\mathbf{Y}_P \mathbf{Y}_P^T)^{-1}$ must be strictly positive, as the $i$-th element on its diagonal is proportional to $\|\mathbf{\Gamma}\mathbf{\Theta}(:,i)\|^2$, and since $\mathbf{\Theta}$ is nonnegative, we can easily get that $(\mathbf{Y}_P \mathbf{Y}_P^T)^{-1}\mathbf{1} > 0$. So for DCMMSB-type models, it is always true that the closet point in $\mathrm{Conv}(\mathbf{Y}_P^T)$ to origin is an interior point of $\mathrm{Conv}(\mathbf{Y}_P^T)$. $\qquad\square$

# E  Corner finding with One-class SVM with empirical inputs

**Lemma E.1.** *Let $\epsilon = \max_i \|\mathbf{y}_i - \hat{\mathbf{y}}_i\|$. Denote $(\mathbf{w}, b)$ and $(\hat{\mathbf{w}}, \hat{b})$ be the optimal solution for the primal problem of One-class SVM in (6) with population $(\mathbf{y}_1, \mathbf{y}_2, \cdots, \mathbf{y}_n)$ and empirical inputs $(\hat{\mathbf{y}}_1, \hat{\mathbf{y}}_2, \cdots, \hat{\mathbf{y}}_n)$ respectively, then $|\hat{b} - b| \le \epsilon$.*

*Proof.* First we have $\mathbf{w}^T \mathbf{y}_i \ge b$, $\forall i \in [n]$, and $\|\mathbf{w}^T(\hat{\mathbf{y}}_i - \mathbf{y}_i)\| \le \epsilon$. Then $\mathbf{w}^T \hat{\mathbf{y}}_i = \mathbf{w}^T \mathbf{y}_i + \mathbf{w}^T(\hat{\mathbf{y}}_i - \mathbf{y}_i) \ge b - \epsilon$. As $(\mathbf{w}, b - \epsilon)$ is a feasible solution of the primal problem with empirical inputs, by optimality of $\hat{b}$, we have $\hat{b} \ge b - \epsilon$. Similarly we can get $b \ge \hat{b} - \epsilon$, so $|\hat{b} - b| \le \epsilon$. $\qquad\square$

**Lemma E.2.** *Let $(\mathbf{w}, b)$, $(\hat{\mathbf{w}}, \hat{b})$ be the hyperplane of the optimal solution of One-class SVM with population and empirical inputs respectively, then $\|\hat{\mathbf{w}} - \mathbf{w}\| \le \zeta\epsilon$, for $\zeta = \frac{4}{\eta b^2 \sqrt{\lambda_K(\mathbf{Y}_P\mathbf{Y}_P^T)}} \le \frac{4K}{\eta(\lambda_K(\mathbf{Y}_P\mathbf{Y}_P^T))^{1.5}}$.*

*Proof.* Let $\beta_l$, $l \in I$ be the solution of the dual problem in Eq. (6) with population inputs, from the construction of this dual problem, we know $\mathbf{w} = \frac{\sum_{l \in I} \beta_l \mathbf{y}_l}{\|\sum_{l \in I} \beta_l \mathbf{y}_l\|}$, $\|\sum_{l \in I} \beta_l \mathbf{y}_l\| = b$, and $\boldsymbol{\beta} := (\beta_{I(1)}, \beta_{I(2)}, \cdots, \beta_{I(p)}) = b^2 (\mathbf{Y}_P\mathbf{Y}_P^T)^{-1}\mathbf{1}$, as shown in Lemma D.1. So $\mathbf{w} = \mathbf{Y}_P^T\boldsymbol{\beta}/b = b\mathbf{Y}_P^T(\mathbf{Y}_P\mathbf{Y}_P^T)^{-1}\mathbf{1}$. From the condition of the primal problem, $\hat{\mathbf{Y}}_P\hat{\mathbf{w}} \ge \hat{b}\mathbf{1}$, then we have $\mathbf{Y}_P\hat{\mathbf{w}} = \hat{\mathbf{Y}}_P\hat{\mathbf{w}} - (\hat{\mathbf{Y}}_P - \mathbf{Y}_P)\hat{\mathbf{w}} \ge (\hat{b} - \epsilon)\mathbf{1} \ge (b - 2\epsilon)\mathbf{1}$. Then there exists a vector $\mathbf{c} \ge \mathbf{0}$ such that $\mathbf{Y}_P\hat{\mathbf{w}} = (b - 2\epsilon)\mathbf{1} + \mathbf{c}$. Now let $\hat{\mathbf{w}} = \mathbf{Y}_P^T\boldsymbol{\varphi} + \hat{\mathbf{w}}_\perp$, where $\mathbf{Y}_P\hat{\mathbf{w}}_\perp = \mathbf{0}$. So $\mathbf{Y}_P\hat{\mathbf{w}} = \mathbf{Y}_P\mathbf{Y}_P^T\boldsymbol{\varphi} = (b - 2\epsilon)\mathbf{1} + \mathbf{c}$, which gives $\hat{\mathbf{w}} = \mathbf{Y}_P^T(\mathbf{Y}_P\mathbf{Y}_P^T)^{-1}((b - 2\epsilon)\mathbf{1} + \mathbf{c}) + \hat{\mathbf{w}}_\perp$. Since $\|\hat{\mathbf{w}}\| = 1$, we have

$$1 = \|\hat{\mathbf{w}}\|^2 = ((b - 2\epsilon)\mathbf{1} + \mathbf{c})^T(\mathbf{Y}_P\mathbf{Y}_P^T)^{-1}((b - 2\epsilon)\mathbf{1} + \mathbf{c}) + \|\hat{\mathbf{w}}_\perp\|^2$$
$$= b^2\mathbf{1}^T(\mathbf{Y}_P\mathbf{Y}_P^T)^{-1}\mathbf{1} + 2b\mathbf{1}^T(\mathbf{Y}_P\mathbf{Y}_P^T)^{-1}(\mathbf{c} - 2\epsilon\mathbf{1}) + (\mathbf{c} - 2\epsilon\mathbf{1})^T(\mathbf{Y}_P\mathbf{Y}_P^T)^{-1}(\mathbf{c} - 2\epsilon\mathbf{1}) + \|\hat{\mathbf{w}}_\perp\|^2.$$

Since $1 = \|\mathbf{w}\|^2 = b^2\mathbf{1}^T(\mathbf{Y}_P\mathbf{Y}_P^T)^{-1}\mathbf{1}$, we have

$$0 \le (\mathbf{c} - 2\epsilon\mathbf{1})^T(\mathbf{Y}_P\mathbf{Y}_P^T)^{-1}(\mathbf{c} - 2\epsilon\mathbf{1}) + \|\hat{\mathbf{w}}_\perp\|^2 = -2b\mathbf{1}^T(\mathbf{Y}_P\mathbf{Y}_P^T)^{-1}(\mathbf{c} - 2\epsilon\mathbf{1}) \tag{10}$$
$$= -2b\mathbf{1}^T(\mathbf{Y}_P\mathbf{Y}_P^T)^{-1}\mathbf{c} + 4b\epsilon\mathbf{1}^T(\mathbf{Y}_P\mathbf{Y}_P^T)^{-1}\mathbf{1},$$

which uses that $(\mathbf{Y}_P\mathbf{Y}_P^T)^{-1}$ is positive definite. This gives

$$2b\mathbf{1}^T(\mathbf{Y}_P\mathbf{Y}_P^T)^{-1}\mathbf{c} \le 4b\epsilon\mathbf{1}^T(\mathbf{Y}_P\mathbf{Y}_P^T)^{-1}\mathbf{1} = 4b\epsilon/b^2$$
$$\implies (\min_i \mathbf{1}^T(\mathbf{Y}_P\mathbf{Y}_P^T)^{-1}\mathbf{e}_i)\|\mathbf{c}\|_1 \le \mathbf{1}^T(\mathbf{Y}_P\mathbf{Y}_P^T)^{-1}\mathbf{c} \le 2\epsilon/b^2,$$

and by Condition 2 we know $(\min_i \mathbf{1}^T(\mathbf{Y}_P\mathbf{Y}_P^T)^{-1}\mathbf{e}_i) \ge \eta$, so $\|\mathbf{c}\| \le \|\mathbf{c}\|_1 \le 2\epsilon/(\eta b^2)$.

Let $\hat{P}$ be the set of support vectors returned by empirical One-class SVM, and $\hat{\boldsymbol{\beta}}$ as the optimal solution for the dual problem, then $\hat{\mathbf{w}} = \hat{\mathbf{Y}}_{\hat{P}}\hat{\boldsymbol{\beta}}/\hat{b}$ and $\sum_{j \in \hat{P}} \hat{\beta}_j = 1$. Now we will give an upper bound on $\|\hat{\mathbf{w}}_\perp\|$. For any $\boldsymbol{v} \in \mathrm{span}(\mathbf{Y}_P)$, we have $\|\hat{\mathbf{w}}_\perp\| \le \|\hat{\mathbf{w}} - \boldsymbol{v}\|$. Now take $\boldsymbol{v} = \mathbf{Y}_{\hat{P}}^T\hat{\boldsymbol{\beta}}/\hat{b}$, since all rows of $\mathbf{Y}$ lie in the span of $\mathbf{Y}_P$, this choice of $\boldsymbol{v}$ also lies in the span of $\mathbf{Y}_P$. Thus,

$$\|\hat{\mathbf{w}}_\perp\| \le \|\hat{\mathbf{w}} - \boldsymbol{v}\| = \|\hat{\mathbf{Y}}_{\hat{P}}^T\hat{\boldsymbol{\beta}} - \mathbf{Y}_{\hat{P}}^T\hat{\boldsymbol{\beta}}\|/\hat{b} = \|\sum_{j \in \hat{P}} \hat{\beta}_j(\mathbf{y}_j - \hat{\mathbf{y}}_j)\|/\hat{b} \le \epsilon/(b - \epsilon).$$

Now, we have

$$\hat{\mathbf{w}} - \mathbf{w} = \mathbf{Y}_P^T(\mathbf{Y}_P\mathbf{Y}_P^T)^{-1}((b-2\epsilon)\mathbf{1} + \mathbf{c}) + \hat{\mathbf{w}}_\perp - b\mathbf{Y}_P^T(\mathbf{Y}_P\mathbf{Y}_P^T)^{-1}\mathbf{1} = \mathbf{Y}_P^T(\mathbf{Y}_P\mathbf{Y}_P^T)^{-1}(\mathbf{c} - 2\epsilon\mathbf{1}) + \hat{\mathbf{w}}_\perp,$$

$$\|\hat{\mathbf{w}} - \mathbf{w}\|^2 = (\mathbf{c} - 2\epsilon\mathbf{1})^T(\mathbf{Y}_P\mathbf{Y}_P^T)^{-1}(\mathbf{c} - 2\epsilon\mathbf{1}) + \|\hat{\mathbf{w}}_\perp\|^2 \le \mathbf{c}^T(\mathbf{Y}_P\mathbf{Y}_P^T)^{-1}\mathbf{c} + 4\epsilon^2/b^2 + \epsilon^2/(b-\epsilon)^2$$

$$\le \|\mathbf{c}\|^2 \lambda_1((\mathbf{Y}_P\mathbf{Y}_P^T)^{-1}) + 4\epsilon^2/b^2 + \epsilon^2/(b-\epsilon)^2 \le \left( \frac{4}{\eta^2 b^4 \lambda_K(\mathbf{Y}_P\mathbf{Y}_P^T)} + \frac{4}{b^2} + \frac{1}{(b-\epsilon)^2} \right)\epsilon^2,$$

where we use Eq. (10) to get that the cross terms are non-negative for the first inequality. First $\frac{4}{\eta^2 b^4 \lambda_K(\mathbf{Y}_P\mathbf{Y}_P^T)} + \frac{4}{b^2} + \frac{1}{(b-\epsilon)^2} < \frac{4}{\eta^2 b^4 \lambda_K(\mathbf{Y}_P\mathbf{Y}_P^T)} + \frac{8}{b^2} < \frac{12}{\eta^2 b^4 \lambda_K(\mathbf{Y}_P\mathbf{Y}_P^T)}$, using $\epsilon < b/2$, $\eta < 1$, $b \le 1$, and $\lambda_K(\mathbf{Y}_P\mathbf{Y}_P^T) < 1$. Then by taking $\zeta = \frac{4}{\eta b^2 \sqrt{\lambda_K(\mathbf{Y}_P\mathbf{Y}_P^T)}}$, we have $\|\hat{\mathbf{w}} - \mathbf{w}\| \le \zeta\epsilon$. Furthermore, $\zeta \le \frac{4K}{\eta(\lambda_K(\mathbf{Y}_P\mathbf{Y}_P^T))^{1.5}}$ by using

$$1/b^2 = \mathbf{1}^T(\mathbf{Y}_P\mathbf{Y}_P^T)^{-1}\mathbf{1} \le K\lambda_1((\mathbf{Y}_P\mathbf{Y}_P^T)^{-1}) = K/\lambda_K(\mathbf{Y}_P\mathbf{Y}_P^T).$$

$\square$

**Lemma E.3.** *Let $(\hat{\mathbf{w}}, \hat{b})$ be the hyperplane of the optimal solution of One-class SVM with empirical inputs, then $\hat{b}\mathbf{1} \le \hat{\mathbf{Y}}_P\hat{\mathbf{w}} \le \hat{b}\mathbf{1} + (\zeta+2)\epsilon\mathbf{1}$.*

*Proof.* Using Lemma E.2,

$$\hat{\mathbf{Y}}_P\hat{\mathbf{w}} = \mathbf{Y}_P\hat{\mathbf{w}} + (\hat{\mathbf{Y}}_P - \mathbf{Y}_P)\hat{\mathbf{w}} \le \mathbf{Y}_P\mathbf{w} + \mathbf{Y}_P(\hat{\mathbf{w}} - \mathbf{w}) + \epsilon\mathbf{1} \le b\mathbf{1} + (\zeta\epsilon + \epsilon)\mathbf{1} \le \hat{b}\mathbf{1} + (\zeta+2)\epsilon\mathbf{1}.$$

$\square$

**Lemma E.4.** *Let $(\mathbf{w}, b)$, $(\hat{\mathbf{w}}, \hat{b})$ be the hyperplane of the optimal solution of One-class SVM with population and empirical inputs respectively, and S be the set of nodes selected as support vectors in the optimal solution of the dual problem with empirical inputs. Then for $r_i$ defined in Lemma A.1, $r_i - 1 \le \frac{1}{b/(2\epsilon)-1}, \forall i \in S$. Furthermore, $\forall i \in [n]$, if $\hat{\mathbf{w}}^T\hat{\mathbf{y}}_i \le \hat{b} + (\zeta+2)\epsilon$, then $r_i - 1 \le \frac{(\zeta+4)\epsilon}{b-2\epsilon}$.*

*Proof.* First $\forall i \in S$, we have,

$$\hat{b} = \hat{\mathbf{w}}^T\hat{\mathbf{y}}_i = \hat{\mathbf{w}}^T\mathbf{y}_i + \hat{\mathbf{w}}^T(\hat{\mathbf{y}}_i - \mathbf{y}_i) = r_i\sum_j \phi_{ij}\hat{\mathbf{w}}^T\mathbf{y}_{I(j)} + \hat{\mathbf{w}}^T(\hat{\mathbf{y}}_i - \mathbf{y}_i)$$

$$= r_i\sum_j \phi_{ij}\hat{\mathbf{w}}^T\hat{\mathbf{y}}_{I(j)} + r_i\sum_j \phi_{ij}\hat{\mathbf{w}}^T(\mathbf{y}_{I(j)} - \hat{\mathbf{y}}_{I(j)}) + \hat{\mathbf{w}}^T(\hat{\mathbf{y}}_i - \mathbf{y}_i)$$

$$\ge r_i\hat{b} - r_i\epsilon - \epsilon.$$

This gives

$$r_i \le \frac{\hat{b}+\epsilon}{\hat{b}-\epsilon} \implies r_i - 1 \le \frac{2\epsilon}{\hat{b}-\epsilon} \le \frac{2\epsilon}{b-\epsilon-\epsilon} = \frac{1}{b/(2\epsilon)-1},$$

where the last step uses $b \ge \hat{b} - \epsilon$ from Lemma E.1. Similarly, for $i \in [n]$ such that $\hat{\mathbf{w}}^T\hat{\mathbf{y}}_i \le \hat{b} + (\zeta+2)\epsilon$, we have $\hat{b} + (\zeta+2)\epsilon \ge r_i\hat{b} - r_i\epsilon - \epsilon$ and this gives $r_i - 1 \le \frac{(\zeta+4)\epsilon}{b-2\epsilon}$. $\square$

**Lemma E.5.** *For S defined in Lemma E.4, $\forall i \in S$, $\exists j \in [K]$ such that for $\phi_{ij}$ defined in Lemma A.1, $\phi_{ij} \ge 1 - \epsilon_1$, for $\epsilon_1 = \frac{2\epsilon}{b\lambda_K(\mathbf{Y}_P\mathbf{Y}_P^T)}$. Furthermore, $\forall i \in [n]$, if $\hat{\mathbf{w}}^T\hat{\mathbf{y}}_i \le \hat{b} + (\zeta+2)\epsilon$, then $\exists j \in [K]$, $\phi_{ij} \ge 1 - \epsilon_2$, for $\epsilon_2 = \frac{(\zeta+4)\epsilon}{(b+(\zeta+2)\epsilon)\lambda_K(\mathbf{Y}_P\mathbf{Y}_P^T)} < \frac{2\zeta\epsilon}{b\lambda_K(\mathbf{Y}_P\mathbf{Y}_P^T)}$.*

*Proof.* By Lemma E.4 we have $r_i \le 1 + \frac{1}{b/(2\epsilon)-1} = \frac{1}{1-2\epsilon/b}$. As $\mathbf{y}_i = r_i\phi_i^T\mathbf{Y}_P$, we have $1 = \|\mathbf{y}_i\| = r_i\|\phi_i^T\mathbf{Y}_P\|$, so $\|\phi_i^T\mathbf{Y}_P\| \ge 1 - 2\epsilon/b$. Let $\mathbf{y}_{-k} = \sum_{j\ne k}\frac{\phi_{ij}}{1-\phi_{ik}}\mathbf{y}_{I(j)}, \forall k \in [K]$. then $\phi_i^T\mathbf{Y}_P = \phi_{ik}\mathbf{y}_{I(k)} + (1-\phi_{ik})\mathbf{y}_{-k}$. It is easy to see that $\|\mathbf{y}_{-k}\| \le 1$, then

$$\|\phi_i^T\mathbf{Y}_P\|^2 \le \phi_{ik}^2 + (1-\phi_{ik})^2 + 2\phi_{ik}(1-\phi_{ik})\mathbf{y}_{I(k)}^T\mathbf{y}_{-k},$$

$$\mathbf{y}_{I(k)}^T\mathbf{y}_{-k} = \sum_{j\ne k}\frac{\phi_{ij}}{1-\phi_{ik}}\mathbf{y}_{I(k)}^T\mathbf{y}_{I(j)} \le \max_{j\ne k}\mathbf{y}_{I(k)}^T\mathbf{y}_{I(j)} \le \max_{i\ne l}\mathbf{y}_{I(i)}^T\mathbf{y}_{I(l)}.$$

Using $2\mathbf{x}_1^T\mathbf{x}_2 = \|\mathbf{x}_1\|^2 + \|\mathbf{x}_2\|^2 - \|\mathbf{x}_1 - \mathbf{x}_2\|^2$ for any same length vectors $\mathbf{x}_1$ and $\mathbf{x}_2$, and

$$\|\mathbf{y}_{I(i)} - \mathbf{y}_{I(l)}\|^2 = \|(\mathbf{e}_i - \mathbf{e}_l)^T\mathbf{Y}_P\|^2 = (\mathbf{e}_l - \mathbf{e}_j)^T\mathbf{Y}_P\mathbf{Y}_P^T(\mathbf{e}_i - \mathbf{e}_l)$$
$$\geq 2\min_{\|\mathbf{x}\|=1}\mathbf{x}^T\mathbf{Y}_P\mathbf{Y}_P^T\mathbf{x} = 2\lambda_K(\mathbf{Y}_P\mathbf{Y}_P^T),$$

we have $\max_{i\neq l}\mathbf{y}_{I(i)}^T\mathbf{y}_{I(l)} \leq 1 - \lambda_K(\mathbf{Y}_P\mathbf{Y}_P^T)$. Then,

$$(1 - 2\epsilon/b)^2 \leq \|\boldsymbol{\phi}_i^T\mathbf{Y}_P\|^2 \leq \phi_{ik}^2 + (1 - \phi_{ik})^2 + 2\phi_{ik}(1 - \phi_{ik})(1 - \lambda_K(\mathbf{Y}_P\mathbf{Y}_P^T))$$
$$= 1 - 2\phi_{ik}(1 - \phi_{ik})\lambda_K(\mathbf{Y}_P\mathbf{Y}_P^T),$$

which gives $\phi_{ik}(1 - \phi_{ik}) \leq \frac{2\epsilon}{b\lambda_K(\mathbf{Y}_P\mathbf{Y}_P^T)} := \epsilon_1, \forall k \in [K]$. Since $\sum_k \phi_{ik} = 1$, we must have $\exists j \in [K], \phi_{ij} \geq 1 - \epsilon_1$. Similarly, for $i \in [n]$ such that $\hat{\mathbf{w}}^T\hat{\mathbf{y}}_i \leq \hat{b} + (\zeta + 2)\epsilon$, we have $r_i - 1 \leq \frac{(\zeta+4)\epsilon}{b-2\epsilon}$ from Lemma E.4, then $\boldsymbol{\phi}_i^T\mathbf{Y}_P = \frac{1}{r_i} \geq 1 - \frac{(\zeta+4)\epsilon}{b+(\zeta+2)\epsilon}$, and this gives that $\phi_{ik}(1 - \phi_{ik}) \leq \frac{(\zeta+4)\epsilon}{(b+(\zeta+2)\epsilon)\lambda_K(\mathbf{Y}_P\mathbf{Y}_P^T)} := \epsilon_2 < \frac{2\zeta\epsilon}{b\lambda_K(\mathbf{Y}_P\mathbf{Y}_P^T)}$, using $\zeta \geq 4$ and $(\zeta + 2)\epsilon \geq 0$. Also since $\sum_k \phi_{ik} = 1$, we must have $\exists j \in [K], \phi_{ij} \geq 1 - \epsilon_2$. $\qquad\square$

*Remark* E.1. Lemma E.5 shows that for One-class SVM with empirical inputs, the support vectors selected are all nearly corner points. Lemma E.3 shows that each corner point is closed to the hyperplane $(\hat{\mathbf{w}}, \hat{b})$ selected by One-class SVM by $(\zeta + 2)\epsilon$, and then Lemma E.5 shows that points close to hyperplane $(\hat{\mathbf{w}}, \hat{b})$ by $(\zeta + 2)\epsilon$ are all nearly corner points. So choosing points that are $(\zeta + 2)\epsilon$ close to $(\hat{\mathbf{w}}, \hat{b})$ will guarantee us all the $K$ corner points and some nearly corner points.

**Lemma E.6.** *Let* $S_c = \{i : \hat{\mathbf{w}}^T\hat{\mathbf{y}}_i \leq \hat{b} + (\zeta + 2)\epsilon\}$, *then* $\forall i, j \in S_c$, *for* $\epsilon_3 = \epsilon + \frac{(\zeta+4)\epsilon}{b-2\epsilon}$, *we have*

$$\|\boldsymbol{\phi}_i - \boldsymbol{\phi}_j\|\sqrt{\lambda_K(\mathbf{Y}_P\mathbf{Y}_P^T)} - 2\epsilon_3 \leq \|\hat{\mathbf{y}}_i - \hat{\mathbf{y}}_j\| \leq \|\boldsymbol{\phi}_i - \boldsymbol{\phi}_j\|\sqrt{\lambda_1(\mathbf{Y}_P\mathbf{Y}_P^T)} + 2\epsilon_3.$$

*Proof.* First we have, $\|\hat{\mathbf{y}}_i - \boldsymbol{\phi}_i^T\mathbf{Y}_P\| = \|\hat{\mathbf{y}}_i - r_i\boldsymbol{\phi}_i^T\mathbf{Y}_P + (r_i - 1)\boldsymbol{\phi}_i^T\mathbf{Y}_P\| \leq \epsilon + \frac{(\zeta+4)\epsilon}{b-2\epsilon} := \epsilon_3$, where last step is by Lemma E.4. This gives $\|(\hat{\mathbf{y}}_i - \hat{\mathbf{y}}_j) - (\boldsymbol{\phi}_i\mathbf{Y}_P - \boldsymbol{\phi}_j\mathbf{Y}_P)\| \leq 2\epsilon_3$, then we have $\|\boldsymbol{\phi}_i^T\mathbf{Y}_P - \boldsymbol{\phi}_j^T\mathbf{Y}_P\| - 2\epsilon_3 \leq \|\hat{\mathbf{y}}_i - \hat{\mathbf{y}}_j\| \leq \|\boldsymbol{\phi}_i^T\mathbf{Y}_P - \boldsymbol{\phi}_j^T\mathbf{Y}_P\| + 2\epsilon_3$. Combing with

$$\|\boldsymbol{\phi}_i - \boldsymbol{\phi}_j\|\sqrt{\lambda_K(\mathbf{Y}_P\mathbf{Y}_P^T)} \leq \|\boldsymbol{\phi}_i^T\mathbf{Y}_P - \boldsymbol{\phi}_j^T\mathbf{Y}_P\| \leq \|\boldsymbol{\phi}_i - \boldsymbol{\phi}_j\|\sqrt{\lambda_1(\mathbf{Y}_P\mathbf{Y}_P^T)},$$

we have the result. $\qquad\square$

**Lemma E.7.** *Let* $S_c = \{i : \hat{\mathbf{w}}^T\hat{\mathbf{y}}_i \leq \hat{b} + (\zeta + 2)\epsilon)\}$, *then there exists exact $K$ clusters in $S_c$, given* $\epsilon \leq c_\epsilon \frac{\eta(\lambda_K(\mathbf{Y}_P\mathbf{Y}_P^T))^3}{K^{1.5}\sqrt{\kappa(\mathbf{Y}_P\mathbf{Y}_P^T)}}$, *for some constant $c_\epsilon$.*

*Proof.* First because $I \in S_c$ from Lemma E.3, there exists at least $K$ clusters in $S_c$. By Lemma E.5, $\forall i \in S_c, \exists k_i \in [K], \phi_{ik_i} \geq 1 - \epsilon_2$. If $k_i = k_j$, by Lemma E.6,

$$\|\hat{\mathbf{y}}_i - \hat{\mathbf{y}}_j\| \leq \|\boldsymbol{\phi}_i - \boldsymbol{\phi}_j\|\sqrt{\lambda_1(\mathbf{Y}_P\mathbf{Y}_P^T)} + 2\epsilon_3 \leq \sqrt{3}\epsilon_2\sqrt{\lambda_1(\mathbf{Y}_P\mathbf{Y}_P^T)} + 2\epsilon_3.$$

This means if $j$ is a corner point, $i$ will be close to it, and will be in the same cluster as long as there is enough separation between different clusters. Now we will prove this is true. Similarly, if $k_i \neq k_j$,

$$\|\hat{\mathbf{y}}_i - \hat{\mathbf{y}}_j\| \geq \|\boldsymbol{\phi}_i - \boldsymbol{\phi}_j\|\sqrt{\lambda_K(\mathbf{Y}_P\mathbf{Y}_P^T)} - 2\epsilon_3 \geq \sqrt{2}(1 - 2\epsilon_2)\sqrt{\lambda_K(\mathbf{Y}_P\mathbf{Y}_P^T)} - 2\epsilon_3.$$

In order to have enough separation between $p$ clusters, we need

$$\sqrt{2}(1 - 2\epsilon_2)\sqrt{\lambda_K(\mathbf{Y}_P\mathbf{Y}_P^T)} - 2\epsilon_3 = \sqrt{2}\sqrt{\lambda_K(\mathbf{Y}_P\mathbf{Y}_P^T)} - 2\sqrt{2}\epsilon_2\sqrt{\lambda_K(\mathbf{Y}_P\mathbf{Y}_P^T)} - 2\epsilon_3$$
$$> c'(\sqrt{3}\epsilon_2\sqrt{\lambda_1(\mathbf{Y}_P\mathbf{Y}_P^T)} + 2\epsilon_3),$$

for some constant $c' > 2$. This is equivalent to show

$$\sqrt{2} > (2\sqrt{2} + \sqrt{3}c'\sqrt{\kappa(\mathbf{Y}_P\mathbf{Y}_P^T)})\epsilon_2 + \frac{2 + 2c'}{\sqrt{\lambda_K(\mathbf{Y}_P\mathbf{Y}_P^T)}}\epsilon_3.$$

As

$$(2\sqrt{2} + \sqrt{3}c'\sqrt{\kappa(\mathbf{Y}_P\mathbf{Y}_P^T)})\epsilon_2 + \frac{2 + 2c'}{\sqrt{\lambda_K(\mathbf{Y}_P\mathbf{Y}_P^T)}}\epsilon_3$$

$$\leq (2\sqrt{2} + \sqrt{3}c'\sqrt{\kappa(\mathbf{Y}_P\mathbf{Y}_P^T)})\frac{2\zeta\epsilon}{b\lambda_K(\mathbf{Y}_P\mathbf{Y}_P^T)} + \frac{2 + 2c'}{\sqrt{\lambda_K(\mathbf{Y}_P\mathbf{Y}_P^T)}}\left(\epsilon + \frac{(\zeta + 4)\epsilon}{b - 2\epsilon}\right)$$

$$\leq c_1\frac{\sqrt{\kappa(\mathbf{Y}_P\mathbf{Y}_P^T)}\zeta\epsilon}{b\lambda_K(\mathbf{Y}_P\mathbf{Y}_P^T)} + \frac{c_2}{\lambda_K(\mathbf{Y}_P\mathbf{Y}_P^T)}\frac{\zeta\epsilon}{b} \leq c_3\frac{\sqrt{\kappa(\mathbf{Y}_P\mathbf{Y}_P^T)}\epsilon}{\lambda_K(\mathbf{Y}_P\mathbf{Y}_P^T)}\frac{4K}{\eta(\lambda_K(\mathbf{Y}_P\mathbf{Y}_P^T))^{1.5}}\frac{\sqrt{K}}{\sqrt{\lambda_K(\mathbf{Y}_P\mathbf{Y}_P^T)}}$$

$$\leq c_4\frac{K^{1.5}\sqrt{\kappa(\mathbf{Y}_P\mathbf{Y}_P^T)}}{\eta(\lambda_K(\mathbf{Y}_P\mathbf{Y}_P^T))^3}\epsilon,$$

where $c_i$, $i \in [4]$ are some constants we do not specify and we use $1/b^2 \leq K/\lambda_K(\mathbf{Y}_P\mathbf{Y}_P^T)$ in the second last inequality. So a sufficient condition for separated clusters is $c_4\frac{K^{1.5}\sqrt{\kappa(\mathbf{Y}_P\mathbf{Y}_P^T)}}{\eta(\lambda_K(\mathbf{Y}_P\mathbf{Y}_P^T))^3}\epsilon < \sqrt{2}$, which is

$$\epsilon \leq c_\epsilon\frac{\eta(\lambda_K(\mathbf{Y}_P\mathbf{Y}_P^T))^3}{K^{1.5}\sqrt{\kappa(\mathbf{Y}_P\mathbf{Y}_P^T)}},$$

for some constant $c_\epsilon$. $\square$

# F   Consistency of inferred parameters

**Lemma F.1.** *For set $C$ returned by Algorithm 1, there exits a permutation matrix $\mathbf{\Pi} \in \mathbb{R}^{K \times K}$ that $\|\hat{\mathbf{Y}}_C - \mathbf{\Pi}\mathbf{Y}_P\|_F \leq \epsilon_4$, for $\epsilon_4 = \frac{c_Y K\zeta}{(\lambda_K(\mathbf{Y}_P\mathbf{Y}_P^T))^{1.5}}\epsilon$ and $c_Y$ is some constant.*

*Proof.* By Lemma E.5, we know that $\forall i \in S_c$, $\exists j \in [K]$ such that $\phi_{ij} \geq 1 - \epsilon_2$. Then we have:

$$\|\hat{\mathbf{y}}_i - \mathbf{y}_{I(j)}\| \leq \|\hat{\mathbf{y}}_i - \mathbf{y}_i\| + \|\mathbf{y}_i - \mathbf{y}_{I(j)}\| \leq \epsilon + \|r_i\sum_l \phi_{il}\mathbf{y}_{I(l)} - r_i\mathbf{y}_{I(j)}\| + \|(r_i - 1)\mathbf{y}_{I(j)}\|$$

$$\leq \epsilon + r_i((1 - \phi_{ij}) + \|\sum_{l \neq j}\phi_{il}\mathbf{y}_{I(l)}\|) + (r_i - 1)$$

$$\leq \epsilon + \left(1 + \frac{(\zeta + 4)\epsilon}{b - 2\epsilon}\right)(2\epsilon_2) + \frac{(\zeta + 4)\epsilon}{b - 2\epsilon} \qquad \text{(by Lemma E.4)}$$

$$\leq \left(1 + \frac{4\zeta}{b}\right)\epsilon + 4\epsilon_2 < \frac{c_Y\zeta}{b\lambda_K(\mathbf{Y}_P\mathbf{Y}_P^T)}\epsilon \leq \frac{c_Y\sqrt{K}\zeta}{(\lambda_K(\mathbf{Y}_P\mathbf{Y}_P^T))^{1.5}}\epsilon,$$

where we use $\epsilon \leq b/(4\zeta)$ and $\zeta \geq 4$. And $c_Y$ is a constant. Then $\|\hat{\mathbf{Y}}_C - \mathbf{\Pi}\mathbf{Y}_P\|_F \leq \frac{c_Y K\zeta}{(\lambda_K(\mathbf{Y}_P\mathbf{Y}_P^T))^{1.5}}\epsilon$. $\square$

**Lemma F.2.** *Let $\max_i \|\mathbf{e}_i^T(\mathbf{Z} - \hat{\mathbf{Z}})\| = \epsilon_0$, then $\|\mathbf{y}_i - \hat{\mathbf{y}}_i\| \leq \frac{2\epsilon_0}{\|\mathbf{z}_i\|}$.*

*Proof.* First note that by definition $\|\|\mathbf{z}_i\| - \|\hat{\mathbf{z}}_i\|\| \leq \epsilon_0$, then,

$$\|\mathbf{y}_i - \hat{\mathbf{y}}_i\| = \left\|\frac{\mathbf{z}_i}{\|\mathbf{z}_i\|} - \frac{\hat{\mathbf{z}}_i}{\|\hat{\mathbf{z}}_i\|}\right\| = \left\|\frac{\|\hat{\mathbf{z}}_i\|\mathbf{z}_i - \|\mathbf{z}_i\|\hat{\mathbf{z}}_i}{\|\mathbf{z}_i\|\|\hat{\mathbf{z}}_i\|}\right\| = \left\|\frac{\|\hat{\mathbf{z}}_i\|(\mathbf{z}_i - \hat{\mathbf{z}}_i) + (\|\hat{\mathbf{z}}_i\| - \|\mathbf{z}_i\|)\hat{\mathbf{z}}_i}{\|\mathbf{z}_i\|\|\hat{\mathbf{z}}_i\|}\right\|$$

$$\leq \left\|\frac{\|\hat{\mathbf{z}}_i\|(\mathbf{z}_i - \hat{\mathbf{z}}_i)}{\|\mathbf{z}_i\|\|\hat{\mathbf{z}}_i\|}\right\| + \left\|\frac{(\|\hat{\mathbf{z}}_i\| - \|\mathbf{z}_i\|)\hat{\mathbf{z}}_i}{\|\mathbf{z}_i\|\|\hat{\mathbf{z}}_i\|}\right\| \leq \left\|\frac{\mathbf{z}_i - \hat{\mathbf{z}}_i}{\|\mathbf{z}_i\|}\right\| + \left\|\frac{\|\hat{\mathbf{z}}_i\| - \|\mathbf{z}_i\|}{\|\mathbf{z}_i\|}\right\| \leq \frac{2\epsilon_0}{\|\mathbf{z}_i\|}.$$

$\square$

*Proof of Theorem 2.8.* First let us get some important intermediate bounds. Using Weyl's inequality,

$$|\sigma_i(\hat{\mathbf{Y}}_C) - \sigma_i(\mathbf{Y}_P)| \leq \|\hat{\mathbf{Y}}_C - \mathbf{\Pi}\mathbf{Y}_P\| \leq \epsilon_4$$

$$|\lambda_i(\hat{\mathbf{Y}}_C\hat{\mathbf{Y}}_C^T) - \lambda_i(\mathbf{Y}_P\mathbf{Y}_P^T)| = |\sigma_i^2(\hat{\mathbf{Y}}_C) - \sigma_i^2(\mathbf{Y}_P)| \leq (\sigma_i(\hat{\mathbf{Y}}_C) + \sigma_i(\mathbf{Y}_P))\epsilon_4$$
$$\leq (2\sigma_i(\mathbf{Y}_P) + \epsilon_4)\epsilon_4.$$

Secondly,

$$\|(\hat{\mathbf{Y}}_C\hat{\mathbf{Y}}_C^T)^{-1}\| = \frac{1}{\lambda_K(\hat{\mathbf{Y}}_C\hat{\mathbf{Y}}_C^T)} \leq \frac{1}{\lambda_K(\mathbf{Y}_P\mathbf{Y}_P^T) - (2\sigma_K(\mathbf{Y}_P) + \epsilon_4)\epsilon_4} \leq \frac{2}{\lambda_K(\mathbf{Y}_P\mathbf{Y}_P^T)},$$

where we use $(2\sigma_K(\mathbf{Y}_P) + \epsilon_4)\epsilon_4 < \lambda_K(\mathbf{Y}_P\mathbf{Y}_P^T)/2$. Then,

$$\|\mathbf{\Pi}(\mathbf{Y}_P\mathbf{Y}_P^T)^{-1} - (\hat{\mathbf{Y}}_C\hat{\mathbf{Y}}_C^T)^{-1}\mathbf{\Pi}\| = \|(\mathbf{\Pi}\mathbf{Y}_P(\mathbf{\Pi}\mathbf{Y}_P)^T)^{-1} - (\hat{\mathbf{Y}}_C\hat{\mathbf{Y}}_C^T)^{-1}\|$$
$$= \|(\mathbf{\Pi}\mathbf{Y}_P(\mathbf{\Pi}\mathbf{Y}_P)^T)^{-1}(\mathbf{\Pi}\mathbf{Y}_P(\mathbf{\Pi}\mathbf{Y}_P)^T - \hat{\mathbf{Y}}_C\hat{\mathbf{Y}}_C^T)(\hat{\mathbf{Y}}_C\hat{\mathbf{Y}}_C^T)^{-1}\|$$
$$\leq \|(\mathbf{Y}_P\mathbf{Y}_P^T)^{-1}\|\|\mathbf{\Pi}\mathbf{Y}_P(\mathbf{\Pi}\mathbf{Y}_P)^T - \hat{\mathbf{Y}}_C\hat{\mathbf{Y}}_C^T\|\|(\hat{\mathbf{Y}}_C\hat{\mathbf{Y}}_C^T)^{-1}\|$$
$$\leq 2\|(\mathbf{Y}_P\mathbf{Y}_P^T)^{-1}\|^2(\|\mathbf{\Pi}\mathbf{Y}_P - \hat{\mathbf{Y}}_C\|\|(\mathbf{\Pi}\mathbf{Y}_P)^T\| + \|\hat{\mathbf{Y}}_C\|\|(\mathbf{\Pi}\mathbf{Y}_P)^T - \hat{\mathbf{Y}}_C^T\|)$$
$$\leq 2\|(\mathbf{Y}_P\mathbf{Y}_P^T)^{-1}\|^2((\|\mathbf{Y}_P\| + \|\hat{\mathbf{Y}}_C\|)\|\hat{\mathbf{Y}}_C - \mathbf{\Pi}\mathbf{Y}_P\|)$$
$$\leq 2\|(\mathbf{Y}_P\mathbf{Y}_P^T)^{-1}\|^2(2\|\mathbf{Y}_P\|\epsilon_4 + \epsilon_4^2).$$

Note that $\mathbf{M} = \mathbf{Z}\mathbf{Y}_P^T(\mathbf{Y}_P\mathbf{Y}_P^T)^{-1}$. Let $\max_i \|\mathbf{e}_i^T(\mathbf{Z} - \hat{\mathbf{Z}})\| = \epsilon_0$, then,

$$\|\mathbf{e}_i^T(\mathbf{M} - \hat{\mathbf{Z}}\hat{\mathbf{Y}}_C^T(\hat{\mathbf{Y}}_C\hat{\mathbf{Y}}_C^T)^{-1}\mathbf{\Pi})\| = \|\mathbf{e}_i^T(\mathbf{Z}\mathbf{Y}_P^T(\mathbf{Y}_P\mathbf{Y}_P^T)^{-1} - \hat{\mathbf{Z}}\hat{\mathbf{Y}}_C^T(\hat{\mathbf{Y}}_C\hat{\mathbf{Y}}_C^T)^{-1}\mathbf{\Pi})\|$$
$$= \|\mathbf{e}_i^T((\mathbf{Z} - \hat{\mathbf{Z}})\mathbf{Y}_P^T(\mathbf{Y}_P\mathbf{Y}_P^T)^{-1})\| + \|\mathbf{e}_i^T(\hat{\mathbf{Z}}(\mathbf{Y}_P - \mathbf{\Pi}^T\hat{\mathbf{Y}}_C)^T(\mathbf{Y}_P\mathbf{Y}_P^T)^{-1})\|$$
$$+ \|\mathbf{e}_i^T(\hat{\mathbf{Z}}\hat{\mathbf{Y}}_C^T(\mathbf{\Pi}(\mathbf{Y}_P\mathbf{Y}_P^T)^{-1} - (\hat{\mathbf{Y}}_C\hat{\mathbf{Y}}_C^T)^{-1}\mathbf{\Pi}))\|$$
$$\leq \|\mathbf{e}_i^T(\mathbf{Z} - \hat{\mathbf{Z}})\|\|\mathbf{Y}_P\|\|(\mathbf{Y}_P\mathbf{Y}_P^T)^{-1}\| + \|\mathbf{e}_i^T\hat{\mathbf{Z}}\|\|\hat{\mathbf{Y}}_C - \mathbf{\Pi}\mathbf{Y}_P\|\|(\mathbf{Y}_P\mathbf{Y}_P^T)^{-1}\|$$
$$+ \|\mathbf{e}_i^T\hat{\mathbf{Z}}\|\|\hat{\mathbf{Y}}_C\|\|\mathbf{\Pi}(\mathbf{Y}_P\mathbf{Y}_P^T)^{-1} - (\hat{\mathbf{Y}}_C\hat{\mathbf{Y}}_C^T)^{-1}\mathbf{\Pi}\|$$
$$\leq (\|\mathbf{e}_i^T(\mathbf{Z} - \hat{\mathbf{Z}})\|\|\mathbf{Y}_P\| + \|\mathbf{e}_i^T\hat{\mathbf{Z}}\|\|\hat{\mathbf{Y}}_C - \mathbf{\Pi}\mathbf{Y}_P\|)\|(\mathbf{Y}_P\mathbf{Y}_P^T)^{-1}\|$$
$$+ 2\|\mathbf{e}_i^T\hat{\mathbf{Z}}\|\|\hat{\mathbf{Y}}_C\|\|(\mathbf{Y}_P\mathbf{Y}_P^T)^{-1}\|^2(2\|\mathbf{Y}_P\|\epsilon_4 + \epsilon_4^2)$$
$$\leq \|(\mathbf{Y}_P\mathbf{Y}_P^T)^{-1}\|(\|\mathbf{Y}_P\|\epsilon_0 + 13\|\mathbf{Y}_P\|^2\|\mathbf{e}_i^T\mathbf{Z}\|\|(\mathbf{Y}_P\mathbf{Y}_P^T)^{-1}\|\epsilon_4)$$
$$\leq \frac{\|\mathbf{Y}_P\|\epsilon_0 + 13\kappa(\mathbf{Y}_P\mathbf{Y}_P^T)\|\mathbf{e}_i^T\mathbf{Z}\|\frac{c_Y K\zeta}{(\lambda_K(\mathbf{Y}_P\mathbf{Y}_P^T))^{1.5}}\epsilon}{\lambda_K(\mathbf{Y}_P\mathbf{Y}_P^T)} \leq \frac{c_M\kappa(\mathbf{Y}_P\mathbf{Y}_P^T)\|\mathbf{e}_i^T\mathbf{Z}\|K\zeta}{(\lambda_K(\mathbf{Y}_P\mathbf{Y}_P^T))^{2.5}}\epsilon := \epsilon_{M,i}$$

where we uses $\epsilon_4 \leq \|\mathbf{Y}_P\|/2$, $\epsilon_0 < \|\mathbf{e}_i^T\mathbf{Z}\|\epsilon/2$ for relaxations. $\square$

# G  Equivalence of using $\hat{\mathbf{V}}$ and $\hat{\mathbf{V}}\hat{\mathbf{V}}^T$ as input of Algorithm 1

**Lemma G.1.** *For DCMMSB-type models, let* $\mathbf{u}_i = \mathbf{U}^T\mathbf{e}_i = \mathbf{v}_i/\|\mathbf{v}_i\|$, $\mathbf{y}_i = \mathbf{Y}^T\mathbf{e}_i = \mathbf{V}\mathbf{v}_i/\|\mathbf{V}\mathbf{v}_i\|$, $\hat{\mathbf{u}}_i = \hat{\mathbf{U}}^T\mathbf{e}_i = \hat{\mathbf{v}}_i/\|\hat{\mathbf{v}}_i\|$, $\hat{\mathbf{y}}_i = \hat{\mathbf{Y}}^T\mathbf{e}_i = \hat{\mathbf{V}}\hat{\mathbf{v}}_i/\|\hat{\mathbf{V}}\hat{\mathbf{v}}_i\|$ *where* $\mathbf{V} = (\mathbf{v}_1, \mathbf{v}_2, \cdots, \mathbf{v}_n)^T$ *and* $\hat{\mathbf{V}} = (\hat{\mathbf{v}}_1, \hat{\mathbf{v}}_2, \cdots, \hat{\mathbf{v}}_n)^T$ *are population and empirical eigenvectors respectively. One-class SVM using rows of* $\mathbf{U}$ *(or* $\hat{\mathbf{U}}$*) and rows of* $\mathbf{Y}$ *(or* $\hat{\mathbf{Y}}$*) will return the same solution* $\boldsymbol{\beta}$.

*Proof.* Since $\mathbf{y}_i = \mathbf{V}\mathbf{v}_i/\|\mathbf{V}\mathbf{v}_i\| = \mathbf{V}\mathbf{v}_i/\|\mathbf{v}_i\| = \mathbf{V}\mathbf{u}_i$, and $\hat{\mathbf{y}}_i = \hat{\mathbf{V}}\hat{\mathbf{v}}_i/\|\hat{\mathbf{V}}\hat{\mathbf{v}}_i\| = \hat{\mathbf{V}}\hat{\mathbf{v}}_i/\|\hat{\mathbf{v}}_i\| = \hat{\mathbf{V}}\hat{\mathbf{u}}_i$, we have $\mathbf{y}_i^T\mathbf{y}_j = \mathbf{u}_i^T\mathbf{V}^T\mathbf{V}\mathbf{u}_j = \mathbf{u}_i^T\mathbf{u}_j$ and $\hat{\mathbf{y}}_i^T\hat{\mathbf{y}}_j = \hat{\mathbf{u}}_i^T\hat{\mathbf{V}}^T\hat{\mathbf{V}}\hat{\mathbf{u}}_j = \hat{\mathbf{u}}_i^T\hat{\mathbf{u}}_j$. It is easy to see that One-class SVM using rows of $\mathbf{U}$ (or $\hat{\mathbf{U}}$) and rows of $\mathbf{Y}$ (or $\hat{\mathbf{Y}}$) have the same objective function (Eq. 6) and thus will have the same solution of $\beta_i$, $i \in [n]$. $\square$

*Remark* G.1. By Lemmas G.1, D.1, and Theorem 2.3, One-class SVM with $\mathbf{y}_i = \mathbf{V}\mathbf{v}_i/\|\mathbf{V}\mathbf{v}_i\|$, $i \in [n]$ as inputs can find all the $K$ corners corresponding to the pure nodes as support vectors for DCMMSB-type models. Furthermore, as $\hat{\mathbf{Y}}_C = \hat{\mathbf{U}}_C\hat{\mathbf{V}}^T$,

$$\hat{\mathbf{M}} = \hat{\mathbf{Z}}\hat{\mathbf{Y}}_C^T(\hat{\mathbf{Y}}_C\hat{\mathbf{Y}}_C^T)^{-1} = \hat{\mathbf{V}}\hat{\mathbf{V}}^T\hat{\mathbf{V}}\hat{\mathbf{U}}_C^T(\hat{\mathbf{U}}_C\hat{\mathbf{V}}^T\hat{\mathbf{V}}\hat{\mathbf{U}}_C)^{-1} = \hat{\mathbf{V}}\hat{\mathbf{U}}_C^T(\hat{\mathbf{U}}_C\hat{\mathbf{U}}_C)^{-1},$$

which shows that outputs of Algorithm 1 using $\hat{\mathbf{V}}$ and $\hat{\mathbf{V}}\hat{\mathbf{V}}^T$ as input are same.

## H  DCMMSB-type models properties

**Lemma H.1.** *For DCMMSB-type models, if $\|\boldsymbol{\theta}_i\|_p = 1$, for $p = 1$ (DCMMSB) or $p = 2$ (OCCAM), then we have $\gamma_i/\sqrt{\lambda_1(\boldsymbol{\Theta}^T\boldsymbol{\Gamma}^2\boldsymbol{\Theta})} \leq \|\mathbf{v}_i\| \leq \gamma_i/\sqrt{\lambda_K(\boldsymbol{\Theta}^T\boldsymbol{\Gamma}^2\boldsymbol{\Theta})}$, and $\gamma_i/\sqrt{\lambda_1(\boldsymbol{\Theta}^T\boldsymbol{\Gamma}^2\boldsymbol{\Theta})} \leq \|\mathbf{v}_i\| \leq \gamma_i/\sqrt{\lambda_K(\boldsymbol{\Theta}^T\boldsymbol{\Gamma}^2\boldsymbol{\Theta})}$, $\forall i \in I$.*

*Proof.* Eq. (8) gives $((\boldsymbol{\Gamma}_P^{-1}\mathbf{V}_P)(\boldsymbol{\Gamma}_P^{-1}\mathbf{V}_P)^T)^{-1} = \boldsymbol{\Theta}^T\boldsymbol{\Gamma}^2\boldsymbol{\Theta}$, then,

$$\max_i \|\mathbf{e}_i^T(\boldsymbol{\Gamma}_P^{-1}\mathbf{V}_P)\|^2 = \max_i \mathbf{e}_i^T(\boldsymbol{\Gamma}_P^{-1}\mathbf{V}_P)(\boldsymbol{\Gamma}_P^{-1}\mathbf{V}_P)^T\mathbf{e}_i \leq \max_{\|\mathbf{x}\|=1} \mathbf{x}^T(\boldsymbol{\Gamma}_P^{-1}\mathbf{V}_P)(\boldsymbol{\Gamma}_P^{-1}\mathbf{V}_P)^T\mathbf{x}$$

$$= \lambda_1((\boldsymbol{\Gamma}_P^{-1}\mathbf{V}_P)(\boldsymbol{\Gamma}_P^{-1}\mathbf{V}_P)^T) = 1/\lambda_K(\boldsymbol{\Theta}^T\boldsymbol{\Gamma}^2\boldsymbol{\Theta})$$

$$\min_i \|\mathbf{e}_i^T(\boldsymbol{\Gamma}_P^{-1}\mathbf{V}_P)\|^2 = \min_i \mathbf{e}_i^T(\boldsymbol{\Gamma}_P^{-1}\mathbf{V}_P)(\boldsymbol{\Gamma}_P^{-1}\mathbf{V}_P)^T\mathbf{e}_i \geq \min_{\|\mathbf{x}\|=1} \mathbf{x}^T(\boldsymbol{\Gamma}_P^{-1}\mathbf{V}_P)(\boldsymbol{\Gamma}_P^{-1}\mathbf{V}_P)^T\mathbf{x}$$

$$= \lambda_K((\boldsymbol{\Gamma}_P^{-1}\mathbf{V}_P)(\boldsymbol{\Gamma}_P^{-1}\mathbf{V}_P)^T) = 1/\lambda_1(\boldsymbol{\Theta}^T\boldsymbol{\Gamma}^2\boldsymbol{\Theta}).$$

By Lemma 2.1, $\forall i \in [n]$, if $\|\boldsymbol{\theta}_i\|_p = 1$, for $p = 1$ or $2$,

$$\|\mathbf{v}_i\| = \gamma_i \boldsymbol{\theta}_i^T\boldsymbol{\Gamma}_P^{-1}\mathbf{V}_P \leq \gamma_i \max_i \|\boldsymbol{\theta}_i\|\|\boldsymbol{\Gamma}_P^{-1}\mathbf{V}_P\| \leq \gamma_i/\sqrt{\lambda_K(\boldsymbol{\Theta}^T\boldsymbol{\Gamma}^2\boldsymbol{\Theta})},$$

where we use $\|\boldsymbol{\theta}_i\| \leq \|\boldsymbol{\theta}_i\|_p = 1$ for $0 < p \leq 2$. Similarly,

$$\|\mathbf{v}_i\| \geq \gamma_i \min_i \|\boldsymbol{\theta}_i\|_1 \min_i \|\mathbf{e}_i(\boldsymbol{\Gamma}_P^{-1}\mathbf{V}_P)\| \geq \gamma_i/\sqrt{\lambda_1(\boldsymbol{\Theta}^T\boldsymbol{\Gamma}^2\boldsymbol{\Theta})}.$$

Note that if $\|\boldsymbol{\theta}_i\|_p = 1$, as $\|\boldsymbol{\theta}_i\|_2 \leq K^{1/2-1/p}\|\boldsymbol{\theta}_i\|_p = K^{1/2-1/p}$, for models with $p > 2$, we need to add a model specifically parameter $\psi = K^{1/2-1/p}$ to the upper bound of $\|\mathbf{v}_i\|$. For simplicity we omit this and only consider cases when $0 < p \leq 2$. $\qquad\square$

**Lemma H.2.** *For DCMMSB-type models whose eigenvectors has the form in Lemma 2.1, if using $\mathbf{Z} = \mathbf{V}\mathbf{V}^T$, $\mathbf{M} = \boldsymbol{\Gamma}\boldsymbol{\Theta}\boldsymbol{\Gamma}_P^{-1}\mathbf{N}_P^{-1}$, then:*

$$\lambda_1(\mathbf{Y}_P\mathbf{Y}_P^T) \leq \kappa(\boldsymbol{\Theta}^T\boldsymbol{\Gamma}^2\boldsymbol{\Theta}), \; \lambda_K(\mathbf{Y}_P\mathbf{Y}_P^T) \geq 1/\kappa(\boldsymbol{\Theta}^T\boldsymbol{\Gamma}^2\boldsymbol{\Theta}), \; and \; \kappa(\mathbf{Y}_P\mathbf{Y}_P^T) \leq (\kappa(\boldsymbol{\Theta}^T\boldsymbol{\Gamma}^2\boldsymbol{\Theta}))^2.$$

*Proof.* For DCMMSB-type models, we have $\mathbf{V} = \boldsymbol{\Gamma}\boldsymbol{\Theta}\boldsymbol{\Gamma}_P^{-1}\mathbf{V}_P$, and $(\mathbf{V}_P\mathbf{V}_P^T)^{-1} = \boldsymbol{\Gamma}_P^{-1}\boldsymbol{\Theta}^T\boldsymbol{\Gamma}^2\boldsymbol{\Theta}\boldsymbol{\Gamma}_P^{-1}$ by Lemma 2.1 and Theorem 2.3 (Eq. 8). Note that $\mathbf{Y}_P = \mathbf{N}_P\mathbf{Z}_P$, then we have

$$\lambda_1(\mathbf{Y}_P\mathbf{Y}_P^T) = \lambda_1(\mathbf{N}_P\mathbf{Z}_P\mathbf{Z}_P^T\mathbf{N}_P) = \lambda_1(\mathbf{N}_P\mathbf{V}_P\mathbf{V}_P^T\mathbf{N}_P) = \lambda_1(\mathbf{N}_P\boldsymbol{\Gamma}_P(\boldsymbol{\Theta}^T\boldsymbol{\Gamma}^2\boldsymbol{\Theta})^{-1}\boldsymbol{\Gamma}_P\mathbf{N}_P)$$

$$\leq (\lambda_1(\mathbf{N}_P\boldsymbol{\Gamma}_P))^2\lambda_1((\boldsymbol{\Theta}^T\boldsymbol{\Gamma}^2\boldsymbol{\Theta})^{-1}) \leq (\max_{i \in I} \gamma_i/\|\mathbf{v}_i\|)^2/\lambda_K(\boldsymbol{\Theta}^T\boldsymbol{\Gamma}^2\boldsymbol{\Theta})$$

$$\leq \lambda_1(\boldsymbol{\Theta}^T\boldsymbol{\Gamma}^2\boldsymbol{\Theta})/\lambda_K(\boldsymbol{\Theta}^T\boldsymbol{\Gamma}^2\boldsymbol{\Theta}) = \kappa(\boldsymbol{\Theta}^T\boldsymbol{\Gamma}^2\boldsymbol{\Theta}) \qquad \text{(by proof of Lemma H.1)}$$

Note that $\mathbf{N}_{ii} = 1/\|\mathbf{e}_i^T\mathbf{Z}\| = 1/\|\mathbf{e}_i^T\mathbf{V}\mathbf{V}^T\| = 1/\|\mathbf{e}_i^T\mathbf{V}\|$. Similarly, we have:

$$\lambda_K(\mathbf{Y}_P\mathbf{Y}_P^T) = \lambda_K(\mathbf{N}_P\boldsymbol{\Gamma}_P(\boldsymbol{\Theta}^T\boldsymbol{\Gamma}^2\boldsymbol{\Theta})^{-1}\boldsymbol{\Gamma}_P\mathbf{N}_P) \geq (\lambda_K(\mathbf{N}_P\boldsymbol{\Gamma}_P))^2\lambda_K((\boldsymbol{\Theta}^T\boldsymbol{\Gamma}^2\boldsymbol{\Theta})^{-1})$$

$$\geq (\min_{i \in I} \gamma_i/\|\mathbf{v}_i\|)^2/\lambda_1(\boldsymbol{\Theta}^T\boldsymbol{\Gamma}^2\boldsymbol{\Theta}) \geq \lambda_K(\boldsymbol{\Theta}^T\boldsymbol{\Gamma}^2\boldsymbol{\Theta})/\lambda_1(\boldsymbol{\Theta}^T\boldsymbol{\Gamma}^2\boldsymbol{\Theta})$$

$$= 1/\kappa(\boldsymbol{\Theta}^T\boldsymbol{\Gamma}^2\boldsymbol{\Theta}) \qquad \text{(by proof of Lemma H.1)}$$

And finally we have,

$$\kappa(\mathbf{Y}_P\mathbf{Y}_P^T) \leq (\kappa(\boldsymbol{\Theta}^T\boldsymbol{\Gamma}^2\boldsymbol{\Theta}))^2.$$

$\qquad\square$

**Lemma H.3.** *For DCMMSB-type models, let $\mathbf{v}_i = \mathbf{V}^T\mathbf{e}_i$, $\hat{\mathbf{v}}_i = \hat{\mathbf{V}}^T\mathbf{e}_i$, $\mathbf{z}_i = \mathbf{V}\mathbf{v}_i$, $\hat{\mathbf{z}}_i = \hat{\mathbf{V}}\hat{\mathbf{v}}_i$, $\mathbf{y}_i = \mathbf{V}\mathbf{v}_i/\|\mathbf{V}\mathbf{v}_i\|$, and $\hat{\mathbf{y}}_i = \hat{\mathbf{V}}\hat{\mathbf{v}}_i/\|\hat{\mathbf{V}}\hat{\mathbf{v}}_i\|$, $i \in [n]$. Also let $\epsilon_0 = \max_i \|\mathbf{z}_i - \hat{\mathbf{z}}_i\|$, then,*

$$\|\mathbf{y}_i - \hat{\mathbf{y}}_i\| \leq \frac{2\epsilon_0}{\|\mathbf{v}_i\|} \leq \frac{2\epsilon_0\sqrt{\lambda_1(\boldsymbol{\Theta}^T\boldsymbol{\Gamma}^2\boldsymbol{\Theta})}}{\gamma_i}.$$

*Proof.* From Lemma F.2, we have

$$\|\mathbf{y}_i - \hat{\mathbf{y}}_i\| \leq \frac{2\epsilon_0}{\|\mathbf{V}\mathbf{v}_i\|} = \frac{2\epsilon_0}{\|\mathbf{v}_i\|} \leq \frac{2\epsilon_0\sqrt{\lambda_1(\boldsymbol{\Theta}^T\boldsymbol{\Gamma}^2\boldsymbol{\Theta})}}{\gamma_i},$$

where the last step uses Lemma H.1. $\qquad\square$

**Lemma H.4.** *For DCMMSB-type models, $\lambda^*(\mathbf{P}) \geq \rho\lambda^*(\mathbf{B})\lambda_K(\boldsymbol{\Theta}^T\boldsymbol{\Gamma}^2\boldsymbol{\Theta})$.*

*Proof.* Let $\mathbf{X} = \mathbf{B}\boldsymbol{\Theta}^T\boldsymbol{\Gamma}^2\boldsymbol{\Theta}\mathbf{B}$, it easy to see that $\mathbf{X}$ is full rank and positive definite, then

$$\lambda^*(\mathbf{P}) = \rho\lambda^*(\boldsymbol{\Gamma}\boldsymbol{\Theta}\mathbf{B}\boldsymbol{\Theta}^T\boldsymbol{\Gamma}) = \rho\sqrt{\lambda_K(\boldsymbol{\Gamma}\boldsymbol{\Theta}\mathbf{B}\boldsymbol{\Theta}^T\boldsymbol{\Gamma}^2\boldsymbol{\Theta}\mathbf{B}\boldsymbol{\Theta}^T\boldsymbol{\Gamma})} = \rho\sqrt{\lambda_K(\boldsymbol{\Gamma}\boldsymbol{\Theta}\mathbf{X}\boldsymbol{\Theta}^T\boldsymbol{\Gamma})}$$

$$= \rho\sqrt{\lambda_K(\mathbf{X}^{1/2}\boldsymbol{\Theta}^T\boldsymbol{\Gamma}^2\boldsymbol{\Theta}\mathbf{X}^{1/2})} = \rho\sqrt{\lambda_K(\mathbf{X}\boldsymbol{\Theta}^T\boldsymbol{\Gamma}^2\boldsymbol{\Theta})} \geq \rho\sqrt{\lambda_K(\mathbf{X})\lambda_K(\boldsymbol{\Theta}^T\boldsymbol{\Gamma}^2\boldsymbol{\Theta})}$$

$$\geq \rho\sqrt{(\lambda_K(\mathbf{B}))^2(\lambda_K(\boldsymbol{\Theta}^T\boldsymbol{\Gamma}^2\boldsymbol{\Theta}))^2} = \rho\lambda^*(\mathbf{B})\lambda_K(\boldsymbol{\Theta}^T\boldsymbol{\Gamma}^2\boldsymbol{\Theta}),$$

where we use that $\mathbf{L}\mathbf{L}^T$ and $\mathbf{L}^T\mathbf{L}$ have the same leading $K$ eigenvalues for a matrix $\mathbf{L} \in \mathbb{R}^{n \times K}$ with rank $K < n$. $\qquad\square$

# I DCMMSB error bounds

**Lemma I.1.** *For DCMMSB-type models, if $\boldsymbol{\theta}_i \sim \text{Dirichlet}(\boldsymbol{\alpha})$, let $\alpha_0 = \mathbf{1}_K^T\boldsymbol{\alpha}$, $\alpha_{\max} = \max_i \alpha_i$, $\alpha_{\min} = \min \alpha_i$, $\nu = \alpha_0/\alpha_{\min}$, then*

$$P\left(\lambda_1(\boldsymbol{\Theta}^T\boldsymbol{\Gamma}^2\boldsymbol{\Theta}) \leq \frac{3\gamma_{\max}^2 n\left(\alpha_{\max} + \|\boldsymbol{\alpha}\|^2\right)}{2\alpha_0(1 + \alpha_0)}\right) \geq 1 - K\exp\left(-\frac{n}{36\nu^2(1 + \alpha_0)^2}\right)$$

$$P\left(\lambda_K(\boldsymbol{\Theta}^T\boldsymbol{\Gamma}^2\boldsymbol{\Theta}) \geq \frac{\gamma_{\min}^2 n}{2\nu(1 + \alpha_0)}\right) \geq 1 - K\exp\left(-\frac{n}{36\nu^2(1 + \alpha_0)^2}\right)$$

$$P\left(\kappa(\boldsymbol{\Theta}^T\boldsymbol{\Gamma}^2\boldsymbol{\Theta}) \leq 3\frac{\gamma_{\max}^2}{\gamma_{\min}^2}\frac{\alpha_{\max} + \|\boldsymbol{\alpha}\|^2}{\alpha_{\min}}\right) \geq 1 - 2K\exp\left(-\frac{n}{36\nu^2(1 + \alpha_0)^2}\right)$$

$$P\left(\lambda^*(\mathbf{P}) \geq \frac{\gamma_{\min}^2\lambda^*(\mathbf{B})}{2\nu(1 + \alpha_0)}\rho n\right) \geq 1 - K\exp\left(-\frac{n}{36\nu^2(1 + \alpha_0)^2}\right)$$

*where $\lambda^*(\mathbf{P})$ is the $K$-th singular value of $\mathbf{P}$.*

*Proof.* First note that

$$\lambda_1(\boldsymbol{\Theta}^T\boldsymbol{\Gamma}^2\boldsymbol{\Theta}) = \lambda_1(\boldsymbol{\Gamma}\boldsymbol{\Theta}\boldsymbol{\Theta}^T\boldsymbol{\Gamma}) \leq (\lambda_1(\boldsymbol{\Gamma}))^2\lambda_1(\boldsymbol{\Theta}\boldsymbol{\Theta}^T) = (\lambda_1(\boldsymbol{\Gamma}))^2\lambda_1(\boldsymbol{\Theta}^T\boldsymbol{\Theta}).$$

Here we use that $\mathbf{X}\mathbf{X}^T$ and $\mathbf{X}^T\mathbf{X}$ have the same leading $K$ eigenvalues for $\mathbf{X} \in \mathbb{R}^{n \times K}$ with rank $K < n$. Also, as $\boldsymbol{\Theta}^T(\boldsymbol{\Gamma}^2 - \gamma_{\min}^2\mathbf{I})\boldsymbol{\Theta}$ is positive semidefinite, we have

$$\lambda_K(\boldsymbol{\Theta}^T\boldsymbol{\Gamma}^2\boldsymbol{\Theta}) = \lambda_K(\boldsymbol{\Theta}^T(\boldsymbol{\Gamma}^2 - \gamma_{\min}^2\mathbf{I})\boldsymbol{\Theta} + \gamma_{\min}^2\boldsymbol{\Theta}^T\boldsymbol{\Theta}) \geq \lambda_K(\boldsymbol{\Theta}^T(\boldsymbol{\Gamma}^2 - \gamma_{\min}^2\mathbf{I})\boldsymbol{\Theta}) + \lambda_K(\gamma_{\min}^2\boldsymbol{\Theta}^T\boldsymbol{\Theta})$$

$$\geq \gamma_{\min}^2\lambda_K(\boldsymbol{\Theta}^T\boldsymbol{\Theta})$$

By Lemma A.2 of [5],

$$P\left(\lambda_1(\boldsymbol{\Theta}^T\boldsymbol{\Theta}) \leq \frac{3n\left(\alpha_{\max} + \|\boldsymbol{\alpha}\|^2\right)}{2\alpha_0(1 + \alpha_0)}\right) \geq 1 - K\exp\left(-\frac{n}{36\nu^2(1 + \alpha_0)^2}\right)$$

$$P\left(\lambda_K(\boldsymbol{\Theta}^T\boldsymbol{\Theta}) \geq \frac{n}{2\nu(1 + \alpha_0)}\right) \geq 1 - K\exp\left(-\frac{n}{36\nu^2(1 + \alpha_0)^2}\right)$$

$$P\left(\kappa(\boldsymbol{\Theta}^T\boldsymbol{\Theta}) \leq 3\frac{\alpha_{\max} + \|\boldsymbol{\alpha}\|^2}{\alpha_{\min}}\right) \geq 1 - 2K\exp\left(-\frac{n}{36\nu^2(1 + \alpha_0)^2}\right)$$

So $\kappa(\mathbf{\Theta}^T\mathbf{\Gamma}^2\mathbf{\Theta}) = \frac{\lambda_1(\mathbf{\Theta}^T\mathbf{\Gamma}^2\mathbf{\Theta})}{\lambda_K(\mathbf{\Theta}^T\mathbf{\Gamma}^2\mathbf{\Theta})} \leq \frac{\gamma_{\max}^2}{\gamma_{\min}^2}\kappa(\mathbf{\Theta}^T\mathbf{\Theta}) \leq 3\frac{\gamma_{\max}^2}{\gamma_{\min}^2}\frac{\alpha_{\max}+\|\boldsymbol{\alpha}\|^2}{\alpha_{\min}}$ with high probability. Using Lemma H.4, we have,

$$\lambda^*(\mathbf{P}) \geq \rho\lambda^*(\mathbf{B})\lambda_K(\mathbf{\Theta}^T\Gamma^2\mathbf{\Theta}) \geq \frac{\gamma_{\min}^2\lambda^*(\mathbf{B})}{2\nu(1+\alpha_0)}\rho n,$$

with probability at least $1 - K\exp\left(-\frac{n}{36\nu^2(1+\alpha_0)^2}\right)$. $\qquad\square$

**Lemma I.2.** *For DCMMSB-type models, we have* $(\mathbf{Y}_P\mathbf{Y}_P^T)^{-1}\mathbf{1} \geq \frac{(\min_i \gamma_i)^2}{\lambda_1(\mathbf{\Theta}^T\mathbf{\Gamma}^2\mathbf{\Theta})}\mathbf{\Theta}^T\mathbf{1}$. *Furthermore, if* $\boldsymbol{\theta}_i \sim \mathrm{Dirichlet}(\boldsymbol{\alpha})$*, with probability larger than* $1 - 1/n^3 - K\exp\left(-\frac{n}{36\nu^2(1+\alpha_0)^2}\right)$*,*
$(\mathbf{Y}_P\mathbf{Y}_P^T)^{-1}\mathbf{1} \geq \frac{(\min_i \gamma_i)^2}{2\lambda_1(\mathbf{\Theta}^T\mathbf{\Gamma}^2\mathbf{\Theta})}\frac{n}{\nu}\mathbf{1} \geq \frac{\gamma_{\min}^2}{3\gamma_{\max}^2}\frac{1}{\nu}\mathbf{1}$*, where* $\nu = \frac{\sum \alpha_i}{\min \alpha_i}$*.*

*Proof.* First note that, for diagonal matrices $\mathbf{D} \in \mathbb{R}_{\geq 0}^{m\times m}$ and $\mathbf{\Gamma} \in \mathbb{R}_{\geq 0}^{n\times n}$ that have strictly positive elements on the diagonal, and some matrices $\mathbf{G} \in \mathbb{R}_{\geq 0}^{m\times m}$ and $\mathbf{H}_1 \in \mathbb{R}_{\geq 0}^{n\times n}$, $\mathbf{H}_2 \in \mathbb{R}_{\geq 0}^{n\times m}$ we have

$$\mathbf{D}\mathbf{G}\mathbf{D}\mathbf{1} \geq (\min_i \mathbf{D}_{ii})^2\mathbf{G}\mathbf{1}, \qquad (11)$$

$$\mathbf{H}_1^T\mathbf{\Gamma}\mathbf{H}_2\mathbf{1} \geq \min_i \mathbf{\Gamma}_{ii}\mathbf{H}_1^T\mathbf{H}_2\mathbf{1}. \qquad (12)$$

Eq. (11) is true because
$$\mathbf{D}\mathbf{G}\mathbf{D}\mathbf{1} - (\min_i \mathbf{D}_{ii})^2\mathbf{G}\mathbf{1} = \mathbf{D}\mathbf{G}\mathbf{D}\mathbf{1} - \min_i \mathbf{D}_{ii}\mathbf{G}\mathbf{D}\mathbf{1} + \min_i \mathbf{D}_{ii}\mathbf{G}\mathbf{D}\mathbf{1} - (\min_i \mathbf{D}_{ii})^2\mathbf{G}\mathbf{1}$$
$$= (\mathbf{D} - \min_i \mathbf{D}_{ii}\mathbf{I})\mathbf{G}\mathbf{D}\mathbf{1} + \min_i \mathbf{D}_{ii}\mathbf{G}(\mathbf{D} - \min_i \mathbf{D}_{ii}\mathbf{I})\mathbf{1} \geq \mathbf{0},$$

where last step follows that $\mathbf{D}$, $\mathbf{G}$ and $(\mathbf{D} - \min_i \mathbf{D}_{ii}\mathbf{I})$ are all non-negative. Eq. (12) can be proved in a similar way. Now use these on Eq. (9), we have

$$(\mathbf{Y}_P\mathbf{Y}_P^T)^{-1}\mathbf{1} = \mathbf{N}_P^{-1}\mathbf{\Gamma}_P^{-1}\mathbf{\Theta}^T\mathbf{\Gamma}^2\mathbf{\Theta}\mathbf{\Gamma}_P^{-1}\mathbf{N}_P^{-1}\mathbf{1} \geq \left(\min_i \frac{\|\mathbf{v}_{I(i)}\|}{\gamma_{I(i)}}\right)^2\mathbf{\Theta}^T\mathbf{\Gamma}^2\mathbf{\Theta}\mathbf{1}$$

$$\geq \left(\min_i \frac{\|\mathbf{v}_{I(i)}\|}{\gamma_{I(i)}}\right)^2(\min_i \gamma_i)^2\mathbf{\Theta}^T\mathbf{1} \geq \frac{(\min_i \gamma_i)^2}{\lambda_1(\mathbf{\Theta}^T\mathbf{\Gamma}^2\mathbf{\Theta})}\mathbf{\Theta}^T\mathbf{1},$$

where the last step follows Lemma H.1. By Lemma C.1. of [4], we know if rows of $\mathbf{\Theta}$ are from Dirichlet distribution with parameter $\boldsymbol{\alpha} = (\alpha_0, \alpha_2, \cdots, \alpha_K)$, $\alpha_0 = \sum_i \alpha_i$, $\nu = \alpha_0/\min_i \alpha_i$,

$$\mathbf{\Theta}^T\mathbf{1} \geq \frac{n}{\nu}\left(1 - O_P\left(\sqrt{\frac{\nu\log n}{n}}\right)\right)\mathbf{1}$$

with probability larger than $1 - 1/n^3$. Now by Lemma I.1, we have, with probability larger than $1 - 1/n^3 - K\exp\left(-\frac{n}{36\nu^2(1+\alpha_0)^2}\right)$,

$$(\mathbf{Y}_P\mathbf{Y}_P^T)^{-1}\mathbf{1} \geq \frac{(\min_i \gamma_i)^2}{\lambda_1(\mathbf{\Theta}^T\mathbf{\Gamma}^2\mathbf{\Theta})}\mathbf{\Theta}^T\mathbf{1} \geq \frac{(\min_i \gamma_i)^2}{\lambda_1(\mathbf{\Theta}^T\mathbf{\Gamma}^2\mathbf{\Theta})}\frac{n}{\nu}\left(1 - O_P\left(\sqrt{\frac{\nu\log n}{n}}\right)\right)\mathbf{1}$$

$$\geq \frac{2\gamma_{\min}^2\alpha_0(1+\alpha_0)}{3\gamma_{\max}^2 n\,(\alpha_{\max}+\|\boldsymbol{\alpha}\|^2)}\frac{n}{\nu}\frac{1}{2}\mathbf{1} = \frac{\gamma_{\min}^2\alpha_{\min}(1+\alpha_0)}{3\gamma_{\max}^2\,(\alpha_{\max}+\|\boldsymbol{\alpha}\|^2)}\mathbf{1} \geq \frac{\gamma_{\min}^2}{3\gamma_{\max}^2}\frac{1}{\nu}\mathbf{1}.$$
$\qquad\square$

We use a crucial result from [5] that shows row-wise eigenspace concentration for general low rank matrix.

**Theorem I.3** (Row-wise eigenspace concentration [5]). *Suppose* $\mathbf{P}$ *has rank* $K$*,* $\max_{i,j}\mathbf{P}_{ij} \leq \rho$*. Let* $\mathbf{A}_{ij} = \mathbf{A}_{ji} \sim \mathrm{Ber}(\mathbf{P}_{ij})$*,* $\mathbf{V}$ *and* $\hat{\mathbf{V}}$ *are* $\mathbf{P}$ *and* $\mathbf{A}$*'s top-$K$ eigenvectors respectively. If* $P(\max_i \|\mathbf{V}_{:,i}\|_\infty > \sqrt{\rho}) \leq \delta_1$*, and for some constant* $\xi > 1$*,* $\rho n = \Omega((\log n)^{2\xi})$ *and* $P(\lambda^*(\mathbf{P}) < 4\sqrt{n\rho}(\log n)^\xi) < \delta_2$*, then for a fixed* $i \in [n]$*, with probability at least* $1 - \delta_1 - \delta_2 - O(Kn^{-3})$*,*

$$\|\mathbf{e}_i^T(\hat{\mathbf{V}}\hat{\mathbf{V}}^T - \mathbf{V}\mathbf{V}^T)\| = O\left(\frac{\min\{K,\kappa(\mathbf{P})\}\sqrt{Kn\rho}}{\lambda^*(\mathbf{P})}\right)\left((\min\{K,\kappa(\mathbf{P})\} + (\log n)^\xi)\max_i \|\mathbf{V}_{:,i}\|_\infty + (K+1)n^{-2\xi}\right).$$

*Proof of Theorem 3.1.* First by Lemma H.3,

$$\|\mathbf{y}_i - \hat{\mathbf{y}}_i\| \leq \frac{2\epsilon_0}{\|\mathbf{v}_i\|} \leq \frac{2\epsilon_0\sqrt{\lambda_1(\mathbf{\Theta}^T\mathbf{\Gamma}^2\mathbf{\Theta})}}{\gamma_i}.$$

Also using Lemma H.1,

$$\max_j \|\mathbf{V}_{:,j}\|_\infty \leq \max_i \|\mathbf{v}_i\| \leq \max_i \gamma_i/\sqrt{\lambda_K(\mathbf{\Theta}^T\mathbf{\Gamma}^2\mathbf{\Theta})},$$

By Lemma I.1, we have $\max_j \|\mathbf{V}_{:,j}\|_\infty \leq \max_i \gamma_i/\sqrt{\lambda_K(\mathbf{\Theta}^T\mathbf{\Gamma}^2\mathbf{\Theta})}$ with probability at least $1 - \delta_1$ for $\delta_1 \leq K\exp\left(-\frac{n}{36\nu^2(1+\alpha_0)^2}\right)$. Also from the condition of $\nu$, $\max_i \gamma_i/\sqrt{\lambda_K(\mathbf{\Theta}^T\mathbf{\Gamma}^2\mathbf{\Theta})} \leq \sqrt{\rho}$. Then it is easy to see $\mathrm{P}(\max_i \|\mathbf{V}_{:,i}\|_\infty > \sqrt{\rho}) \leq \delta_1$. Also, from the condition of $\lambda^*(\mathbf{B})/\nu$, we have $4\sqrt{n\rho}(\log n)^\xi \leq \frac{\gamma_{\min}^2\lambda^*(\mathbf{B})}{2\nu(1+\alpha_0)}\rho n$. Then combined with Lemma I.1, $\mathrm{P}(\lambda^*(\mathbf{P}) < 4\sqrt{n\rho}(\log n)^\xi) < \delta_2$ is satisfied with $\delta_2 \leq K\exp\left(-\frac{n}{36\nu^2(1+\alpha_0)^2}\right)$. Also we have,

$$\max_i \gamma_i/\sqrt{\lambda_K(\mathbf{\Theta}^T\mathbf{\Gamma}^2\mathbf{\Theta})} \geq \max_i \gamma_i/\sqrt{\lambda_1(\mathbf{\Theta}^T\mathbf{\Gamma}^2\mathbf{\Theta})} \geq \sqrt{2/(3n)} \gg (K+1)n^{-2\xi}$$

with high probability. Then by Theorem I.3 we have

$$\epsilon_0 = O\left(\frac{\min\{K, \kappa(\mathbf{P})\}\sqrt{Kn\rho}}{\lambda^*(\mathbf{P})}\right)\left((\min\{K, \kappa(\mathbf{P})\} + (\log n)^\xi)\max_i \|\mathbf{V}_{:,i}\|_\infty + (K+1)n^{-2\xi}\right)$$

$$= \tilde{O}\left(\frac{\min\{K^2, (\kappa(\mathbf{P}))^2\}\sqrt{Kn\rho}}{\rho\lambda^*(\mathbf{B})\lambda_K(\mathbf{\Theta}^T\mathbf{\Gamma}^2\mathbf{\Theta})}\right)\frac{\gamma_{\max}}{\sqrt{\lambda_K(\mathbf{\Theta}^T\mathbf{\Gamma}^2\mathbf{\Theta})}}.$$

with probability at least $1 - \delta_1 - \delta_2 - O(Kn^{-3}) = 1 - O(Kn^{-3})$. So,

$$\|\mathbf{y}_i - \hat{\mathbf{y}}_i\| \leq \frac{2\epsilon_0\sqrt{\lambda_1(\mathbf{\Theta}^T\mathbf{\Gamma}^2\mathbf{\Theta})}}{\gamma_i} = \tilde{O}\left(\frac{\min\{K^2, (\kappa(\mathbf{P}))^2\}\sqrt{Kn}}{\sqrt{\rho}\lambda^*(\mathbf{B})\lambda_K(\mathbf{\Theta}^T\mathbf{\Gamma}^2\mathbf{\Theta})}\right)\frac{\gamma_{\max}\sqrt{\kappa(\mathbf{\Theta}^T\mathbf{\Gamma}^2\mathbf{\Theta})}}{\gamma_i}.$$

And using Lemma I.1,

$$\epsilon = \max_i \|\mathbf{y}_i - \hat{\mathbf{y}}_i\| = \tilde{O}\left(\frac{\gamma_{\max}\min\{K^2, (\kappa(\mathbf{P}))^2\}\sqrt{\kappa(\mathbf{\Theta}^T\mathbf{\Gamma}^2\mathbf{\Theta})}\sqrt{Kn}}{\gamma_{\min}\lambda^*(\mathbf{B})\lambda_K(\mathbf{\Theta}^T\mathbf{\Gamma}^2\mathbf{\Theta})\sqrt{\rho}}\right)$$

$$= \tilde{O}\left(\frac{\gamma_{\max}\min\{K^2, (\kappa(\mathbf{P}))^2\}\sqrt{\kappa(\mathbf{\Theta}^T\mathbf{\Gamma}^2\mathbf{\Theta})}K^{0.5}\nu(1+\alpha_0)}{\gamma_{\min}^3\lambda^*(\mathbf{B})\sqrt{n\rho}}\right)$$

with probability at least $1 - O(Kn^{-2})$. $\qquad\qquad\square$

*Proof of Theorem 3.2.* Note that $\mathbf{P} = \rho\mathbf{\Gamma}\mathbf{\Theta}\mathbf{B}\mathbf{\Theta}^T\mathbf{\Gamma} = \mathbf{V}\mathbf{E}\mathbf{V}^T$, we have $\rho\mathbf{\Gamma}_P\mathbf{B}\mathbf{\Gamma}_P = \mathbf{V}_P\mathbf{E}\mathbf{V}_P^T$, then $\rho\mathbf{N}_P\mathbf{\Gamma}_P\mathbf{B}\mathbf{\Gamma}_P\mathbf{N}_P = \mathbf{N}_P\mathbf{V}_P\mathbf{E}\mathbf{V}_P^T\mathbf{N}_P = \mathbf{Y}_P\mathbf{V}\mathbf{E}\mathbf{V}^T\mathbf{Y}_P^T$. As $\mathbf{B}$ has unit diagonal, let $\mathbf{B}(i,i) = c^2$, then $c^2\rho\gamma_{I(i)}^2/\|\mathbf{v}_{I(i)}\|^2 = \rho\mathbf{e}_i^T\mathbf{N}_P\mathbf{\Gamma}_P\mathbf{B}\mathbf{\Gamma}_P\mathbf{N}_P\mathbf{e}_i = \mathbf{e}_i^T\mathbf{Y}_P\mathbf{V}\mathbf{E}\mathbf{V}^T\mathbf{Y}_P^T\mathbf{e}_i := d_i^2$. Since our estimation for $c^2\rho\gamma_{I(i)}^2/\|\mathbf{v}_{I(i)}\|^2$ is $\mathbf{e}_i^T\mathbf{\Pi}^T\hat{\mathbf{Y}}_C\hat{\mathbf{V}}\hat{\mathbf{E}}\hat{\mathbf{V}}^T\hat{\mathbf{Y}}_C^T\mathbf{\Pi}\mathbf{e}_i$, and note that $\|\mathbf{E}\| \leq \max_i \|\mathbf{e}_i^T\mathbf{P}\|_1 = O(\rho n)$, $\|\hat{\mathbf{E}}\| \leq \|\mathbf{E}\| + \|\mathbf{A} - \mathbf{P}\| = O(\rho n)$ using Weyl's inequality and Theorem 5.2 of [3], and $\|\mathbf{V}\mathbf{E}\mathbf{V}^T - \hat{\mathbf{V}}\hat{\mathbf{E}}\hat{\mathbf{V}}^T\| \leq \lambda_{K+1}(\mathbf{A}) + \|\mathbf{P} - \mathbf{A}\| \leq 2\|\mathbf{P} - \mathbf{A}\| = O(\sqrt{\rho n})$. Let $\hat{d}_i^2 = \mathbf{e}_i^T\hat{\mathbf{Y}}_C\hat{\mathbf{V}}\hat{\mathbf{E}}\hat{\mathbf{V}}^T\hat{\mathbf{Y}}_C^T\mathbf{e}_i$, then we have,

$$\begin{aligned}
|d_i^2 - \hat{d}_{\pi(i)}^2| &= \|\mathbf{e}_i^T\mathbf{Y}_P\mathbf{V}\mathbf{E}\mathbf{V}^T\mathbf{Y}_P^T\mathbf{e}_i - \mathbf{e}_i^T\mathbf{\Pi}^T\hat{\mathbf{Y}}_C\hat{\mathbf{V}}\hat{\mathbf{E}}\hat{\mathbf{V}}^T\hat{\mathbf{Y}}_C^T\mathbf{\Pi}\mathbf{e}_i\| \\
&\leq \|\mathbf{e}_i^T(\mathbf{Y}_P - \mathbf{\Pi}^T\hat{\mathbf{Y}}_C)\mathbf{V}\mathbf{E}\mathbf{V}^T\mathbf{Y}_P^T\mathbf{e}_i\| + \|\mathbf{e}_i^T\mathbf{\Pi}^T\hat{\mathbf{Y}}_C(\mathbf{V}\mathbf{E}\mathbf{V}^T - \hat{\mathbf{V}}\hat{\mathbf{E}}\hat{\mathbf{V}}^T)\mathbf{Y}_P^T\mathbf{e}_i\| \\
&\quad + \|\mathbf{e}_i^T\mathbf{\Pi}^T\hat{\mathbf{Y}}_C\hat{\mathbf{V}}\hat{\mathbf{E}}\hat{\mathbf{V}}^T(\mathbf{Y}_P^T - \hat{\mathbf{Y}}_C^T\mathbf{\Pi})\mathbf{e}_i\| \\
&\leq \|\mathbf{e}_i^T(\mathbf{Y}_P - \mathbf{\Pi}^T\hat{\mathbf{Y}}_C)\|\|\mathbf{E}\| + \|\mathbf{V}\mathbf{E}\mathbf{V}^T - \hat{\mathbf{V}}\hat{\mathbf{E}}\hat{\mathbf{V}}^T\| + \|\hat{\mathbf{E}}\|\|\mathbf{e}_i^T(\mathbf{Y}_P - \mathbf{\Pi}^T\hat{\mathbf{Y}}_C)\| \\
&\leq O(\rho n)\epsilon_4/\sqrt{K} + O(\sqrt{\rho n}).
\end{aligned}$$

Using Lemma H.1, $c\sqrt{\rho\lambda_K(\mathbf{\Theta}^T\mathbf{\Gamma}^2\mathbf{\Theta})} \leq d_i \leq c\sqrt{\rho\lambda_1(\mathbf{\Theta}^T\mathbf{\Gamma}^2\mathbf{\Theta})}$, and by Lemma I.1, $\lambda_1(\mathbf{\Theta}^T\mathbf{\Gamma}^2\mathbf{\Theta}) \leq \frac{3\gamma_{\max}^2 n(\alpha_{\max}+\|\boldsymbol{\alpha}\|^2)}{2\alpha_0(1+\alpha_0)}$, $\lambda_K(\mathbf{\Theta}^T\mathbf{\Gamma}^2\mathbf{\Theta}) \geq \frac{\gamma_{\min}^2 n}{2\nu(1+\alpha_0)}$, then we have $d_i \geq c\sqrt{\frac{\gamma_{\min}^2 \rho n}{2\nu(1+\alpha_0)}}$, and $d_i \leq c\sqrt{\frac{3\gamma_{\max}^2 \rho n(\alpha_{\max}+\|\boldsymbol{\alpha}\|^2)}{2\alpha_0(1+\alpha_0)}}$ with probability at least $1-2K\exp\left(-\frac{n}{36\nu^2(1+\alpha_0)^2}\right)$. Then, using Lemma H.2,

$$
\begin{aligned}
|d_i - \hat{d}_{\pi(i)}| &\leq \frac{O(\rho n)\epsilon_4/\sqrt{K} + O(\sqrt{\rho n})}{\min_j(d_j + \hat{d}_{\pi(j)})} \leq \frac{O(\rho n)\epsilon_4/\sqrt{K} + O(\sqrt{\rho n})}{\sqrt{\rho\lambda_K(\mathbf{\Theta}^T\mathbf{\Gamma}^2\mathbf{\Theta})}} \\
&\leq \frac{O(\rho n/\sqrt{K})\frac{c_Y K\zeta}{(\lambda_K(\mathbf{Y}_P\mathbf{Y}_P^T))^{1.5}}\epsilon + O(\sqrt{\rho n})}{\sqrt{\rho\lambda_K(\mathbf{\Theta}^T\mathbf{\Gamma}^2\mathbf{\Theta})}} = O\left(\frac{K^{0.5}(\kappa(\mathbf{\Theta}^T\mathbf{\Gamma}^2\mathbf{\Theta}))^{1.5}\zeta\sqrt{\rho}n}{\sqrt{\lambda_K(\mathbf{\Theta}^T\mathbf{\Gamma}^2\mathbf{\Theta})}}\epsilon\right) \\
&= O\left(\frac{K^{1.5}(\kappa(\mathbf{\Theta}^T\mathbf{\Gamma}^2\mathbf{\Theta}))^3\sqrt{\rho}n}{\eta\sqrt{\lambda_K(\mathbf{\Theta}^T\mathbf{\Gamma}^2\mathbf{\Theta})}}\epsilon\right).
\end{aligned}
$$

Let $\mathbf{D} = \operatorname{diag}(d_1, d_2, \cdots, d_K)$ and $\hat{\mathbf{D}} = \operatorname{diag}(\hat{d}_1, \hat{d}_2, \cdots, \hat{d}_K)$, then $\mathbf{D} = c\sqrt{\rho}(\mathbf{N}_P\mathbf{\Gamma}_P)$. Now as we estimate $c\sqrt{\rho}(\mathbf{\Gamma}\mathbf{\Theta})$ by $\hat{c}\sqrt{\hat{\rho}}\hat{\mathbf{\Gamma}}\hat{\mathbf{\Theta}} = \hat{\mathbf{M}}\hat{\mathbf{D}}$, we have

$$
\begin{aligned}
&\|\mathbf{e}_i^T(c\sqrt{\rho}\mathbf{\Gamma}\mathbf{\Theta} - \hat{c}\sqrt{\hat{\rho}}\hat{\mathbf{\Gamma}}\hat{\mathbf{\Theta}}\mathbf{\Pi})\| = \|\mathbf{e}_i^T(\mathbf{M}\mathbf{D} - \hat{\mathbf{M}}\hat{\mathbf{D}}\mathbf{\Pi})\| \leq \|\mathbf{e}_i^T(\mathbf{M} - \hat{\mathbf{M}}\mathbf{\Pi})\mathbf{D}\| + \|\mathbf{e}_i^T\hat{\mathbf{M}}\mathbf{\Pi}(\mathbf{D} - \mathbf{\Pi}^T\hat{\mathbf{D}}\mathbf{\Pi})\| \\
&\leq \|\mathbf{e}_i^T(\mathbf{M} - \hat{\mathbf{M}}\mathbf{\Pi})\|\|\mathbf{D}\| + \|\mathbf{e}_i^T\hat{\mathbf{M}}\|\|\mathbf{D} - \mathbf{\Pi}^T\hat{\mathbf{D}}\mathbf{\Pi}\| \leq \epsilon_{M,i}\max_j d_j + (\|\mathbf{e}_i^T\mathbf{M}\| + \epsilon_{M,i})\max_j|d_j - \hat{d}_{\pi(j)}| \\
&\leq \epsilon_{M,i}\max_j d_j + (\gamma_i\max_{j\in I}\|\mathbf{v}_j\|/\gamma_j + \epsilon_{M,i})\max_j|d_j - \hat{d}_{\pi(j)}| \\
&\leq c\sqrt{\rho\lambda_1(\mathbf{\Theta}^T\mathbf{\Gamma}^2\mathbf{\Theta})}\epsilon_{M,i} + \left(\frac{\gamma_i}{\sqrt{\lambda_K(\mathbf{\Theta}^T\mathbf{\Gamma}^2\mathbf{\Theta})}} + \epsilon_{M,i}\right)O\left(\frac{K^{1.5}(\kappa(\mathbf{\Theta}^T\mathbf{\Gamma}^2\mathbf{\Theta}))^3\sqrt{\rho}n}{\eta\sqrt{\lambda_K(\mathbf{\Theta}^T\mathbf{\Gamma}^2\mathbf{\Theta})}}\epsilon\right),
\end{aligned}
$$

where we use $\|\mathbf{e}_i^T\mathbf{M}\| = \|\mathbf{e}_i^T\mathbf{\Gamma}\mathbf{\Theta}\mathbf{\Gamma}_P^{-1}\mathbf{N}_P^{-1}\| \leq \gamma_i\|\boldsymbol{\theta}_i\|\max_{j\in I}\|\mathbf{v}_j\|/\gamma_j$ and $\|\boldsymbol{\theta}_i\| \leq 1$ for DCMMSB and OCCAM for the last inequality. As

$$
\begin{aligned}
\epsilon_{M,i} &= \frac{c_M\kappa(\mathbf{Y}_P\mathbf{Y}_P^T)\|\mathbf{e}_i^T\mathbf{Z}\|K\zeta}{(\lambda_K(\mathbf{Y}_P\mathbf{Y}_P^T))^{2.5}}\epsilon \leq \frac{c_M\kappa(\mathbf{Y}_P\mathbf{Y}_P^T)\|\mathbf{e}_i^T\mathbf{Z}\|K}{(\lambda_K(\mathbf{Y}_P\mathbf{Y}_P^T))^{2.5}}\frac{4K}{\eta(\lambda_K(\mathbf{Y}_P\mathbf{Y}_P^T))^{1.5}}\epsilon \\
&\leq \frac{c_1\|\mathbf{e}_i^T\mathbf{Z}\|(\kappa(\mathbf{\Theta}^T\mathbf{\Gamma}^2\mathbf{\Theta}))^6 K^2}{\eta}\epsilon \leq \frac{c_1\gamma_i(\kappa(\mathbf{\Theta}^T\mathbf{\Gamma}^2\mathbf{\Theta}))^6 K^2}{\eta\sqrt{\lambda_K(\mathbf{\Theta}^T\mathbf{\Gamma}^2\mathbf{\Theta})}}\epsilon.
\end{aligned}
$$

Then

$$
\begin{aligned}
\epsilon_5 &= \|\mathbf{e}_i^T(c\sqrt{\rho}\mathbf{\Gamma}\mathbf{\Theta} - \hat{c}\sqrt{\hat{\rho}}\hat{\mathbf{\Gamma}}\hat{\mathbf{\Theta}}\mathbf{\Pi})\| \\
&= c\sqrt{\rho\lambda_1(\mathbf{\Theta}^T\mathbf{\Gamma}^2\mathbf{\Theta})}\epsilon_{M,i} + \left(\frac{\gamma_i}{\sqrt{\lambda_K(\mathbf{\Theta}^T\mathbf{\Gamma}^2\mathbf{\Theta})}} + \epsilon_{M,i}\right)O\left(\frac{K^{1.5}(\kappa(\mathbf{\Theta}^T\mathbf{\Gamma}^2\mathbf{\Theta}))^3\sqrt{\rho}n}{\eta\sqrt{\lambda_K(\mathbf{\Theta}^T\mathbf{\Gamma}^2\mathbf{\Theta})}}\epsilon\right) \\
&= c\sqrt{\rho\lambda_1(\mathbf{\Theta}^T\mathbf{\Gamma}^2\mathbf{\Theta})}\frac{c_1\gamma_i(\kappa(\mathbf{\Theta}^T\mathbf{\Gamma}^2\mathbf{\Theta}))^6 K^2}{\eta\sqrt{\lambda_K(\mathbf{\Theta}^T\mathbf{\Gamma}^2\mathbf{\Theta})}}\epsilon + O\left(\gamma_i\frac{K^{1.5}(\kappa(\mathbf{\Theta}^T\mathbf{\Gamma}^2\mathbf{\Theta}))^3\sqrt{\rho}n}{\eta\lambda_K(\mathbf{\Theta}^T\mathbf{\Gamma}^2\mathbf{\Theta})}\epsilon\right) \\
&= O\left(\max\left\{K^{0.5}(\kappa(\mathbf{\Theta}^T\mathbf{\Gamma}^2\mathbf{\Theta}))^{3.5}, \frac{n}{\lambda_K(\mathbf{\Theta}^T\mathbf{\Gamma}^2\mathbf{\Theta})}\right\}\frac{\gamma_i K^{1.5}(\kappa(\mathbf{\Theta}^T\mathbf{\Gamma}^2\mathbf{\Theta}))^3\sqrt{\rho}}{\eta}\epsilon\right).
\end{aligned}
$$

As $|c\sqrt{\rho}\gamma_i - \hat{c}\sqrt{\hat{\rho}}\hat{\gamma}_i| = \|\mathbf{e}_i^T(c\sqrt{\rho}\mathbf{\Gamma}\mathbf{\Theta} - \hat{c}\sqrt{\hat{\rho}}\hat{\mathbf{\Gamma}}\hat{\mathbf{\Theta}}\mathbf{\Pi})\mathbf{1}\| \leq \sqrt{K}\epsilon_5$, let $\mathbf{X}_i = \mathbf{e}_i^T c\sqrt{\rho}\mathbf{\Gamma}\mathbf{\Theta}$ and $\hat{\mathbf{X}}_i = \mathbf{e}_i^T\hat{c}\sqrt{\hat{\rho}}\hat{\mathbf{\Gamma}}\hat{\mathbf{\Theta}}\mathbf{\Pi}$, then for DCMMSB, $\|\mathbf{X}_i\|_1 = c\sqrt{\rho}\gamma_i$, $\|\hat{\mathbf{X}}_i\| \leq \|\hat{\mathbf{X}}_i\|_1 = \hat{c}\sqrt{\hat{\rho}}\hat{\gamma}_i$; for OCCAM,

$\|\mathbf{X}_i\| = c\sqrt{\rho}\gamma_i \|\mathbf{e}_i^T\mathbf{\Theta}\| = c\sqrt{\rho}\gamma_i$ and $\|\hat{\mathbf{X}}_i\| = \hat{c}\sqrt{\hat{\rho}}\hat{\gamma}_i$. So we have,

$$\|\mathbf{e}_i^T(\mathbf{\Theta} - \hat{\mathbf{\Theta}}\mathbf{\Pi})\| = \left\| \frac{\mathbf{X}_i}{c\sqrt{\rho}\gamma_i} - \frac{\hat{\mathbf{X}}_i}{\hat{c}\sqrt{\hat{\rho}}\hat{\gamma}_i} \right\| \le \frac{\|\mathbf{X}_i - \hat{\mathbf{X}}_i\|}{c\sqrt{\rho}\gamma_i} + \left\| \frac{c\sqrt{\rho}\gamma_i - \hat{c}\sqrt{\hat{\rho}}\hat{\gamma}_i}{c\sqrt{\rho}\gamma_i \hat{c}\sqrt{\hat{\rho}}\hat{\gamma}_i} \right\| \|\hat{\mathbf{X}}_i\|$$

$$\le \frac{\epsilon_5}{c\sqrt{\rho}\gamma_i} + \frac{\sqrt{K}}{c\sqrt{\rho}\gamma_i}\epsilon_5 = O\left( \frac{\sqrt{K}}{\gamma_i\sqrt{\rho}}\epsilon_5 \right)$$

$$= O\left( \max\left\{ K^{0.5}(\kappa(\mathbf{\Theta}^T\mathbf{\Gamma}^2\mathbf{\Theta}))^{3.5}, \frac{n}{\lambda_K(\mathbf{\Theta}^T\mathbf{\Gamma}^2\mathbf{\Theta})} \right\} \frac{K^2(\kappa(\mathbf{\Theta}^T\mathbf{\Gamma}^2\mathbf{\Theta}))^3}{\eta}\epsilon \right)$$

$$= \tilde{O}\left( \max\left\{ K^{0.5}(\kappa(\mathbf{\Theta}^T\mathbf{\Gamma}^2\mathbf{\Theta}))^{3.5}, \frac{n}{\lambda_K(\mathbf{\Theta}^T\mathbf{\Gamma}^2\mathbf{\Theta})} \right\} \frac{\gamma_{\max}K^{2.5}\min\{K^2, (\kappa(\mathbf{P}))^2\}(\kappa(\mathbf{\Theta}^T\mathbf{\Gamma}^2\mathbf{\Theta}))^{3.5}\sqrt{n}}{\gamma_{\min}\eta\lambda^*(\mathbf{B})\lambda_K(\mathbf{\Theta}^T\mathbf{\Gamma}^2\mathbf{\Theta})\sqrt{\rho}} \right)$$

$$= \tilde{O}\left( \frac{\gamma_{\max}K^{2.5}\min\{K^2, (\kappa(\mathbf{P}))^2\}n^{3/2}}{\gamma_{\min}\eta\lambda^*(\mathbf{B})\lambda_K^2(\mathbf{\Theta}^T\mathbf{\Gamma}^2\mathbf{\Theta})\sqrt{\rho}} \right). \qquad \text{(when } \kappa(\mathbf{\Theta}^T\mathbf{\Gamma}^2\mathbf{\Theta}) = \Theta(1))$$

Note that this bound works for both DCMMSB and OCCAM, and $\lambda_K(\mathbf{\Theta}^T\mathbf{\Gamma}^2\mathbf{\Theta}) = \Omega(n)$, so the bound is about $\tilde{O}\left(1/\sqrt{\rho n}\right)$. specifically, for DCMMSB,

$$\|\mathbf{e}_i^T(\mathbf{\Theta} - \hat{\mathbf{\Theta}}\mathbf{\Pi})\| = \tilde{O}\left( \frac{\gamma_{\max}K^{2.5}\min\{K^2, (\kappa(\mathbf{P}))^2\}n^{3/2}}{\gamma_{\min}\eta\lambda^*(\mathbf{B})\lambda_K^2(\mathbf{\Theta}^T\mathbf{\Gamma}^2\mathbf{\Theta})\sqrt{\rho}} \right)$$

$$= \tilde{O}\left( \frac{\gamma_{\max}K^{2.5}\min\{K^2, (\kappa(\mathbf{P}))^2\}\nu^2(1 + \alpha_0)^2}{\gamma_{\min}^5\eta\lambda^*(\mathbf{B})\sqrt{\rho n}} \right).$$

$\square$

## J  Topic model error bounds

### J.1  Eigenspcae concentration for topic models

Consider the following setup similar to [1].

$$\mathbf{A}_{ij} \overset{iid}{\sim} \text{Binomial}(N, \mathcal{A}_{ij}) \qquad \text{For } i \in [V], j \in [D] \tag{13}$$

Here $\mathcal{A}$ is the probability matrix for words appearing in documents. Furthermore, we have $\mathcal{A} = \mathbf{TH}$, where $\mathbf{T}$ is the word to topic probabilities with columns summing to 1 and $\mathbf{H}$ is the topic to document matrix with columns summing to 1. Also note that, $\sum_i \|\mathbf{e}_i^T\mathcal{A}\mathcal{A}^T\|_1 = D$, since the columns of $\mathcal{A}$ sum to one. We will construct a matrix $\mathbf{A}_1\mathbf{A}_2^T$, where $\mathbf{A}_1$ and $\mathbf{A}_2$ are obtained by dividing the words in each document uniformly randomly in two equal parts. For simplicity denote $N_1 = N/2$. Consider the matrix $\mathbf{U} = \frac{\mathbf{A}_1\mathbf{A}_2^T}{N_1^2}$. We have $\mathrm{E}[\mathbf{U}] = \mathcal{A}\mathcal{A}^T$.

**Lemma J.1.** *For topic models, we have* $(\mathbf{Y}_P\mathbf{Y}_P^T)^{-1}\mathbf{1} \ge \frac{\min_i \|\mathbf{e}_i^T\mathbf{T}\|_1}{\lambda_1(\mathbf{T}^T\mathbf{T})}\mathbf{1} \ge \frac{\min_i \|\mathbf{e}_i^T\mathbf{T}\|_1}{K}\mathbf{1}$, *where* $\mathbf{T}$ *is the word-topic probability matrix.*

*Proof.* Noting that $\mathbf{T} = \mathbf{\Gamma}\mathbf{\Theta}$ for topic models, where $\gamma_i = \mathbf{\Gamma}_{ii} = \|\mathbf{e}_i^T\mathbf{T}\|_1$. Following the steps of Lemma I.2, we find

$$(\mathbf{Y}_P\mathbf{Y}_P^T)^{-1}\mathbf{1} \ge \frac{\min_i \gamma_i}{\lambda_1(\mathbf{\Theta}^T\mathbf{\Gamma}^2\mathbf{\Theta})}(\mathbf{\Gamma}\mathbf{\Theta})^T\mathbf{1} = \frac{\min_i \gamma_i}{\lambda_1(\mathbf{T}^T\mathbf{T})}\mathbf{T}^T\mathbf{1} = \frac{\min_i \gamma_i}{\lambda_1(\mathbf{T}^T\mathbf{T})}\mathbf{1} \ge \frac{\min_i \|\mathbf{e}_i^T\mathbf{T}\|_1}{K}\mathbf{1},$$

where the last step is true because $\lambda_1(\mathbf{T}^T\mathbf{T}) \le \text{trace}(\mathbf{T}^T\mathbf{T}) = \sum_i \|\mathbf{Te}_i\|^2 \le K$.

So $\eta \ge \frac{\min_i \|\mathbf{e}_i^T\mathbf{T}\|_1}{\lambda_1(\mathbf{T}^T\mathbf{T})} \ge \frac{\min_i \|\mathbf{e}_i^T\mathbf{T}\|_1}{K}$. $\square$

**Lemma J.2.** *Using Eq (13), we see that under Assumption 3.1,*

$$\mathrm{P}\left( \|\mathbf{U} - \mathcal{A}\mathcal{A}^T\|_F \ge \sqrt{\frac{50D\log\max(V, D)}{N_1}} \right) \le \frac{2}{(\max(V, D))^3}.$$

*Proof.* Recall that from Assumption 3.1, $g_{ik} = \mathbf{e}_i^T \boldsymbol{\mathcal{A}}\boldsymbol{\mathcal{A}}^T e_k$. Let $\mathbf{R} := \mathbf{U} - \boldsymbol{\mathcal{A}}\boldsymbol{\mathcal{A}}^T$.

$$\mathbf{R}_{ik} = \frac{\sum_{j=1}^{D} \mathbf{A}_1(ij)\mathbf{A}_2(kj)}{N_1^2} - g_{ik}$$

Note that $\mathrm{E}[\mathbf{R}_{ij}] = 0$, and $\mathbf{A}_1(ij)\mathbf{A}_2(kj)/N_1^2$ is bounded by 1. Also $\mathbf{A}_1(i,j)$ and $\mathbf{A}_2(i,j)$ are independent. For independent $X := \mathbf{A}_1(ij)/N_1$, $Y := \mathbf{A}_1(kj)/N_1$,

$$\mathrm{var}(XY) = \mathrm{var}(X)\mathrm{var}(Y) + \mathrm{var}(X)\mathrm{E}[X]^2 + \mathrm{var}(Y)\mathrm{E}[Y]^2 \leq \frac{3\boldsymbol{\mathcal{A}}_1(ij)\boldsymbol{\mathcal{A}}_2(kj)}{N_1}$$

$$\mathrm{var}(\mathbf{R}_{ik}) \leq 3g_{ik}/N_1$$

When $g_{ik} = 0$, $\mathbf{U}_{ik} = 0$. When $g_{ik} > 0$, using Bernstein's inequality, we have:

$$\mathrm{P}\left(|\mathbf{R}_{ik}| \geq t_{ik}\right) \leq 2\exp\left(-\frac{t_{ik}^2}{2(3g_{ik}/N_1 + t_{ik}/3)}\right),$$

Setting, $t_{ik} = \sqrt{50\log\max(V,D)g_{ik}/N_1}$, we see that,

$$\sum_{i,k} t_{ik}^2 = 50\log\max(V,D)\sum_{ik} g_{ik}/N_1 = 50\log\max(V,D)D/N_1$$

Then,

$$\mathrm{P}\left(\|\mathbf{R}\|_F^2 \geq \sum_{i,k} t_{ik}^2\right) \leq V^2 \max_{i,k} \mathrm{P}\left(|\mathbf{R}_{ik}| \geq t_{ik}\right) \leq 2V^2/(\max(V,D))^5 \leq 2/(\max(V,D))^3.$$

This yields the result. $\qquad\qquad\qquad\qquad\qquad\qquad\qquad\qquad\qquad\qquad\qquad\qquad\square$

**Lemma J.3.** *Using Eq (13), we see that, under Assumption 3.1, there exists constants $C, r$ such that,*

$$\mathrm{P}\left(\|\mathbf{U} - \boldsymbol{\mathcal{A}}\boldsymbol{\mathcal{A}}^T\| \geq C_r\sqrt{\frac{D\log\max(V,D)}{N}}\right) \leq \frac{2}{(\max(V,D))^r}.$$

*Proof.* We use the Matrix Bernstein bound in [8]. Let $\mathbf{S}_k := \frac{\mathbf{A}_{1k}\mathbf{A}_{2k}^T}{N_1^2} - \boldsymbol{\mathcal{A}}_k\boldsymbol{\mathcal{A}}_k^T$, where $\mathbf{M}_k$ is the

$k^{th}$ column of matrix $\mathbf{M}$. Note that $\mathrm{E}[\mathbf{S}_k]$ is the $V \times V$ zeros matrix. We also see that by symmetry of the random splitting, $\mathrm{E}[\mathbf{S}_k\mathbf{S}_k^T] = \mathrm{E}[\mathbf{S}_k^T\mathbf{S}_k]$.

We will now note some theoretical properties of the $\mathbf{S}_k$ matrices. Let $\mathbf{X}$ be a vector of size $V$, such that, $\mathbf{X}_i \sim \mathrm{Binomial}(N_1, a_i)$.

$$\frac{\mathrm{E}[\mathbf{X}^T\mathbf{X}]}{N_1^2} = \sum_{i=1}^{V} \frac{\mathrm{E}[\mathbf{X}_i^2]}{N_1^2} = \sum_{i=1}^{V} \frac{\mathrm{E}[\mathbf{X}_i]^2 + \mathrm{var}(\mathbf{X}_i)}{N_1^2}$$

$$= \sum_{i=1}^{V} \frac{N_1^2 a_i^2 + N_1 a_i(1 - a_i)}{N_1^2} = \left(1 - \frac{1}{N_1}\right)\|a\|^2 + \frac{1}{N_1} \qquad (14)$$

Furthermore, let

$$\mathrm{Cov}(\mathbf{X}) = \boldsymbol{\Sigma}, \qquad \boldsymbol{\Sigma}_{ij} = N_1 a_i(1 - a_i)\mathbb{1}(i = j) \qquad (15)$$

Then,

$$\mathrm{E}[\mathbf{S}_k\mathbf{S}_k^T] = \mathrm{E}\left[\frac{\mathbf{A}_{1k}\mathbf{A}_{2k}^T\mathbf{A}_{2k}\mathbf{A}_{1k}^T}{N_1^4} - \boldsymbol{\mathcal{A}}_k\boldsymbol{\mathcal{A}}_k^T\boldsymbol{\mathcal{A}}_k\boldsymbol{\mathcal{A}}_k^T\right]$$

$$\text{(By independence)} \qquad = \frac{\mathrm{E}[\mathbf{A}_{2k}^T\mathbf{A}_{2k}]\mathrm{E}[\mathbf{A}_{1k}\mathbf{A}_{1k}^T]}{N_1^4} - \|\boldsymbol{\mathcal{A}}_k\|^2\boldsymbol{\mathcal{A}}_k\boldsymbol{\mathcal{A}}_k^T$$

$$\text{(By Eq (14) and (15))} \qquad = \left(\frac{1}{N_1} + \|\boldsymbol{\mathcal{A}}_k\|^2(1 - \frac{1}{N_1})\right)\left(\frac{\boldsymbol{\Sigma}_k}{N_1^2} + \boldsymbol{\mathcal{A}}_k\boldsymbol{\mathcal{A}}_k^T\right) - \|\boldsymbol{\mathcal{A}}_k\|^2\boldsymbol{\mathcal{A}}_k\boldsymbol{\mathcal{A}}_k^T$$

$$= \|\boldsymbol{\mathcal{A}}_k\|^2\frac{\boldsymbol{\Sigma}_k}{N_1^2} + \frac{1 - \|\boldsymbol{\mathcal{A}}_k\|^2}{N_1}\left(\frac{\boldsymbol{\Sigma}_k}{N_1^2} + \boldsymbol{\mathcal{A}}_k\boldsymbol{\mathcal{A}}_k^T\right)$$

Since $\|\mathbf{\Sigma}_k\| \le N_1\|\mathcal{A}_k\|_1 = N_1, \|\mathcal{A}\|_F^2 \le D$,

$$v(\mathbf{S}) = \left\|\sum_k \mathrm{E}[\mathbf{S}_k\mathbf{S}_k^T]\right\| \le 2\frac{\|\mathcal{A}\|_F^2}{N_1} + \frac{D}{N_1^2} \le \frac{D}{N_1}\left(2 + \frac{1}{N_1}\right).$$

Furthermore,

$$\|\mathbf{S}_k\| \le \|\mathcal{A}_k\|^2 + \frac{\|\mathbf{A}_{1k}\|\|\mathbf{A}_{2k}\|}{N_1^2} \le 2 =: L$$

So the Matrix Bernstein bound gives us:

$$\mathrm{P}\left(\|\sum_k \mathbf{S}_k\| \ge t\right) \le 2V\exp\left(-\frac{t^2/2}{v(\mathbf{S}) + Lt/3}\right) = 2V\exp\left(-\frac{t^2/2}{3D/N_1 + 2t/3}\right)$$

Using $t = C_r\sqrt{D\log\max(V, D)/N}$, and using the condition in Assumption 3.1, we get the bound. $\square$

*Proof of Lemma 3.3.* First note the proof is under Assumption 3.1. Let $\mathbf{R} = \mathbf{U} - \mathcal{A}\mathcal{A}^T$. Using the Davis-Kahan Theorem [9], we see that there exists an orthogonal matrix $\mathbf{O}$:

$$\|\hat{\mathbf{V}}\mathbf{O} - \mathbf{V}\|_F \le \frac{\sqrt{8}(2\lambda_1(\mathcal{A}\mathcal{A}^T) + \|\mathbf{R}\|_2)\min(\sqrt{K}\|\mathbf{R}\|_2, \|\mathbf{R}\|_F)}{\lambda_K^2(\mathcal{A}\mathcal{A}^T)},$$

where $\lambda_1$ and $\lambda_K$ are the largest and $K^{th}$ largest singular values (and also eigenvalue) of $\mathcal{A}\mathcal{A}^T$ respectively. Thus,

$$\begin{aligned}
\|\hat{\mathbf{V}}\mathbf{O} - \mathbf{V}\|_F &\le \frac{\sqrt{8}(2\lambda_1(\mathcal{A}\mathcal{A}^T) + \|\mathbf{R}\|_2)\min(\sqrt{K}\|\mathbf{R}\|_2, \|\mathbf{R}\|_F)}{\lambda_K^2(\mathcal{A}\mathcal{A}^T)} \\
&\le \sqrt{8}\frac{2\lambda_1(\mathcal{A}\mathcal{A}^T) + C_r\sqrt{D\log\max(V,D)/N}}{\lambda_K(\mathcal{A}\mathcal{A}^T)^2}\sqrt{\frac{D\log\max(V,D)}{N}}\max\left(C_r\sqrt{K}, \sqrt{C}\right) \\
&\le \frac{\lambda_1(\mathbf{H}\mathbf{H}^T)\lambda_1(\mathbf{T}^T\mathbf{T})}{\lambda_K^2(\mathbf{H}\mathbf{H}^T)\lambda_K^2(\mathbf{T}^T\mathbf{T})}O_P\left(\sqrt{\frac{KD\log\max(V,D)}{N}}\right) \\
&= \frac{\kappa(\mathbf{H}\mathbf{H}^T)\kappa(\mathbf{T}^T\mathbf{T})}{\lambda_K(\mathbf{T}^T\mathbf{T})}O_P\left(\sqrt{\frac{K\log\max(V,D)}{DN}}\right),
\end{aligned}$$

where the third inequality follows Lemma H.4 with $\mathbf{P} = \mathcal{A}\mathcal{A}^T, \mathbf{\Gamma}\mathbf{\Theta} = \mathbf{T}, \mathbf{B} = \mathbf{H}\mathbf{H}^T$ and $\rho = 1$. Now we bound $\epsilon_0 = \max_i\|\mathbf{z}_i - \hat{\mathbf{z}}_i\| = \|\mathbf{e}_i^T(\hat{\mathbf{V}}\hat{\mathbf{V}}^T - \mathbf{V}\mathbf{V}^T)\|$ as:

$$\begin{aligned}
\|\mathbf{e}_i^T(\hat{\mathbf{V}}\hat{\mathbf{V}}^T - \mathbf{V}\mathbf{V}^T)\| &\le \|\hat{\mathbf{V}}\hat{\mathbf{V}}^T - \mathbf{V}\mathbf{V}^T\|_2 \le \|(\hat{\mathbf{V}}\mathbf{O} - \mathbf{V})\mathbf{O}^T\hat{\mathbf{V}}^T + \mathbf{V}(\hat{\mathbf{V}}\mathbf{O} - \mathbf{V})^T\| \\
&= \frac{\kappa(\mathbf{H}\mathbf{H}^T)\kappa(\mathbf{T}^T\mathbf{T})}{\lambda_K(\mathbf{T}^T\mathbf{T})}O_P\left(\sqrt{\frac{K\log\max(V,D)}{DN}}\right).
\end{aligned}$$

By Lemma H.3, $\|\mathbf{y}_i - \hat{\mathbf{y}}_i\| \le \frac{2\epsilon_0}{\|\mathbf{v}_i\|} \le \frac{2\epsilon_0\sqrt{\lambda_1(\mathbf{T}^T\mathbf{T})}}{\|\mathbf{e}_i^T\mathbf{T}\|_1}$. So,

$$\epsilon = \max_i\|\mathbf{y}_i - \hat{\mathbf{y}}_i\| = \frac{\kappa(\mathbf{H}\mathbf{H}^T)(\kappa(\mathbf{T}^T\mathbf{T}))^{1.5}}{\min_j\|\mathbf{e}_j^T\mathbf{T}\|_1\sqrt{\lambda_K(\mathbf{T}^T\mathbf{T})}}O_P\left(\sqrt{\frac{K\log\max(V,D)}{DN}}\right).$$

$\square$

## J.2 Parameter estimation for topic models

*Proof of Theorem 3.4.* For topic models, $\mathbf{M} = \mathbf{T}\mathbf{D}$, where $\mathbf{T} = \boldsymbol{\Gamma}\boldsymbol{\Theta}$, $\mathbf{D} = (\mathbf{N}_P \boldsymbol{\Gamma}_P)^{-1}$, $\gamma_i = \boldsymbol{\Gamma}_{ii} = \|\mathbf{e}_i^T \mathbf{T}\|_1$. For empirical estimation we have $\hat{\mathbf{M}} = \hat{\mathbf{T}}\hat{\mathbf{D}}$, where $\hat{\mathbf{D}}(i,i) = \|\hat{\mathbf{M}}(:,i)\|_1$. First we have $\forall i \in K$, $\|\mathbf{T}(:,i)\|_1 = 1$, then $\|\mathbf{M}(:,i)\|_1 = \mathbf{D}(i,i) = \|\mathbf{v}_{I(i)}\|/\gamma_{I(i)}$. Let $\pi$ be the permutation function for permutation matrix $\boldsymbol{\Pi}$ in Theorem 2.8, then,

$$|\mathbf{D}(i,i) - \hat{\mathbf{D}}(\pi(i),\pi(i))| = |\|\mathbf{M}(:,i)\|_1 - \|\hat{\mathbf{M}}(:,\pi(i))\|_1| \le \|\mathbf{M}(:,i) - \hat{\mathbf{M}}(:,\pi(i))\|_1$$

$$= \sum_{j=1}^{V} |\mathbf{M}(j,i) - \hat{\mathbf{M}}(j,\pi(i))| \le \sum_{j=1}^{V} \|\mathbf{M}(j,:) - \hat{\mathbf{M}}(j,:)\boldsymbol{\Pi}\|_1$$

$$= \sum_{j=1}^{V} \|\mathbf{e}_j^T \mathbf{T}\|_1 \frac{\|\mathbf{M}(j,:) - \hat{\mathbf{M}}(j,:)\boldsymbol{\Pi}\|_1}{\|\mathbf{e}_j^T \mathbf{T}\|_1}$$

$$\le K \max_j \frac{\|\mathbf{M}(j,:) - \hat{\mathbf{M}}(j,:)\boldsymbol{\Pi}\|_1}{\|\mathbf{e}_j^T \mathbf{T}\|_1} \le K^{1.5} \max_j \frac{\|\mathbf{M}(j,:) - \hat{\mathbf{M}}(j,:)\boldsymbol{\Pi}\|}{\|\mathbf{e}_j^T \mathbf{T}\|_1}$$

$$\le \frac{K^{1.5} \max_j \epsilon_{M,j}}{\min_j \|\mathbf{e}_j^T \mathbf{T}\|_1} := \epsilon_D$$

Note that $\mathbf{T}^T \mathbf{T} = \boldsymbol{\Theta}^T \boldsymbol{\Gamma}^2 \boldsymbol{\Theta}$, and from Lemma H.1, we know

$$1/\sqrt{\lambda_1(\mathbf{T}^T \mathbf{T})} \le \|\mathbf{v}_i\|/\gamma_i \le 1/\sqrt{\lambda_K(\mathbf{T}^T \mathbf{T})}, \ \forall i \in [n]$$

Using Lemma H.2, we have $\lambda_1(\mathbf{Y}_P \mathbf{Y}_P^T) \le \kappa(\boldsymbol{\Theta}\boldsymbol{\Gamma}^2\boldsymbol{\Theta}) = \kappa(\mathbf{T}^T \mathbf{T})$, $\lambda_K(\mathbf{Y}_P \mathbf{Y}_P^T) \ge 1/\kappa(\boldsymbol{\Theta}\boldsymbol{\Gamma}^2\boldsymbol{\Theta}) = 1/\kappa(\mathbf{T}^T \mathbf{T})$, and $\kappa(\mathbf{Y}_P \mathbf{Y}_P^T) \le (\kappa(\boldsymbol{\Theta}\boldsymbol{\Gamma}^2\boldsymbol{\Theta})) = (\kappa(\mathbf{T}^T \mathbf{T}))^2$.

Then the error for each row of $\mathbf{T}$ is

$$\|\mathbf{e}_i^T(\hat{\mathbf{T}} - \mathbf{T}\boldsymbol{\Pi}^T)\| = \|\mathbf{e}_i^T(\hat{\mathbf{M}}\hat{\mathbf{D}}^{-1} - \mathbf{M}\mathbf{D}^{-1}\boldsymbol{\Pi}^T)\|$$

$$\le \|\mathbf{e}_i^T(\hat{\mathbf{M}} - \mathbf{M}\boldsymbol{\Pi}^T)\hat{\mathbf{D}}^{-1}\| + \|\mathbf{e}_i^T \mathbf{M}\boldsymbol{\Pi}^T(\hat{\mathbf{D}}^{-1} - \boldsymbol{\Pi}\mathbf{D}^{-1}\boldsymbol{\Pi}^T)\|$$

$$\le \|\mathbf{e}_i^T(\hat{\mathbf{M}} - \mathbf{M}\boldsymbol{\Pi}^T)\| \max_j 1/\hat{\mathbf{D}}(j,j) + \|\mathbf{e}_i^T \mathbf{M}\| \max_j \left\| \frac{\mathbf{D}(j,j) - \hat{\mathbf{D}}(\pi(j),\pi(j))}{\mathbf{D}(j,j)\hat{\mathbf{D}}(\pi(j),\pi(j))} \right\|$$

$$\le \frac{2\epsilon_{M,i}}{\min_j \mathbf{D}(j,j)} + \frac{2\epsilon_D}{(\min_j \mathbf{D}(j,j))^2} \|\mathbf{e}_i^T \mathbf{M}\|$$

$$\le \frac{2\epsilon_{M,i}}{\min_j \|\mathbf{v}_{I(j)}\|/\gamma_{I(j)}} + \frac{2\epsilon_D}{\min_j \|\mathbf{v}_{I(j)}\|/\gamma_{I(j)}} \frac{\max_j \mathbf{D}(j,j)\|\mathbf{e}_i^T \mathbf{T}\|}{\min_j \mathbf{D}(j,j)}$$

$$\le 2\sqrt{\lambda_1(\mathbf{T}^T \mathbf{T})}\epsilon_{M,i} + 2\sqrt{\lambda_1(\mathbf{T}^T \mathbf{T})} \frac{\sqrt{\lambda_1(\mathbf{T}^T \mathbf{T})}}{\sqrt{\lambda_K(\mathbf{T}^T \mathbf{T})}} \|\mathbf{e}_i^T \mathbf{T}\| \frac{K^{1.5} \max_j \epsilon_{M,j}}{\min_j \|\mathbf{e}_j^T \mathbf{T}\|_1}$$

$$\le 4\sqrt{\lambda_1(\mathbf{T}^T \mathbf{T})}\sqrt{\kappa(\mathbf{T}^T \mathbf{T})} \frac{\|\mathbf{e}_i^T \mathbf{T}\|}{\min_j \|\mathbf{e}_j^T \mathbf{T}\|_1} K^{1.5} \frac{c_M \kappa(\mathbf{Y}_P \mathbf{Y}_P^T) \max_j \|\mathbf{e}_j^T \mathbf{Z}\| K\zeta}{(\lambda_K(\mathbf{Y}_P \mathbf{Y}_P^T))^{2.5}} \epsilon$$

$$\le c_1 \sqrt{\lambda_1(\mathbf{T}^T \mathbf{T})}(\kappa(\mathbf{T}^T \mathbf{T}))^{5.5} K^{2.5} \frac{\gamma_{\max}\|\mathbf{e}_i^T \mathbf{T}\|}{\min_j \|\mathbf{e}_j^T \mathbf{T}\|_1} \zeta\epsilon$$

$$\le \frac{c_2 \sqrt{\lambda_1(\mathbf{T}^T \mathbf{T})}(\kappa(\mathbf{T}^T \mathbf{T}))^7 K^{3.5}}{\eta} \frac{\max_j \|\mathbf{e}_j^T \mathbf{T}\|_1}{\min_j \|\mathbf{e}_j^T \mathbf{T}\|_1} \|\mathbf{e}_i^T \mathbf{T}\|\epsilon,$$

$$\text{(using } \zeta \le \tfrac{4K}{\eta(\lambda_K(\mathbf{Y}_P \mathbf{Y}_P^T))^{1.5}})$$

where we use $\epsilon_D \le \mathbf{D}(j,j)/2$ for relaxation in the 3rd inequality and $c_1$ and $c_2$ are some constants. Under Assumption 3.1, by Lemma 3.3, we have

$$\epsilon = \max_i \|\mathbf{y}_i - \hat{\mathbf{y}}_i\| = \frac{\kappa(\mathbf{H}\mathbf{H}^T)(\kappa(\mathbf{T}^T \mathbf{T}))^{1.5}}{\min_j \|\mathbf{e}_j^T \mathbf{T}\|_1 \sqrt{\lambda_K(\mathbf{T}^T \mathbf{T})}} O_P \left( \sqrt{\frac{K \log \max(V,D)}{DN}} \right).$$

Then,

$$\frac{\|\mathbf{e}_i^T(\hat{\mathbf{T}} - \mathbf{T}\mathbf{\Pi}^T)\|}{\|\mathbf{e}_i^T\mathbf{T}\|} \leq \frac{c_2\sqrt{\lambda_1(\mathbf{T}^T\mathbf{T})}(\kappa(\mathbf{T}^T\mathbf{T}))^7 K^{3.5}}{\eta}\frac{\max_j\|\mathbf{e}_j^T\mathbf{T}\|_1}{\min_j\|\mathbf{e}_j^T\mathbf{T}\|_1}\epsilon$$

$$=\frac{\sqrt{\lambda_1(\mathbf{T}^T\mathbf{T})}(\kappa(\mathbf{T}^T\mathbf{T}))^7 K^{3.5}}{\eta}\frac{\max_j\|\mathbf{e}_j^T\mathbf{T}\|_1}{\min_j\|\mathbf{e}_j^T\mathbf{T}\|_1}\frac{\kappa(\mathbf{H}\mathbf{H}^T)(\kappa(\mathbf{T}^T\mathbf{T}))^{1.5}}{\min_j\|\mathbf{e}_j^T\mathbf{T}\|_1\sqrt{\lambda_K(\mathbf{T}^T\mathbf{T})}}O_P\left(\sqrt{\frac{K\log\max(V,D)}{DN}}\right)$$

$$=\frac{\max_j\|\mathbf{e}_j^T\mathbf{T}\|_1}{(\min_j\|\mathbf{e}_j^T\mathbf{T}\|_1)^2}\frac{\kappa(\mathbf{H}\mathbf{H}^T)(\kappa(\mathbf{T}^T\mathbf{T}))^9}{\eta}O_P\left(K^4\sqrt{\frac{\log\max(V,D)}{DN}}\right)$$

$$=O_P\left(\frac{K^4\max_j\|\mathbf{e}_j^T\mathbf{T}\|_1}{\eta(\min_j\|\mathbf{e}_j^T\mathbf{T}\|_1)^2}\sqrt{\frac{\log\max(V,D)}{DN}}\right) \qquad \text{(if } \kappa(\mathbf{T}^T\mathbf{T}) = \Theta(1) \text{ and } \kappa(\mathbf{H}\mathbf{H}^T) = \Theta(1))$$

$$\square$$

## K  Converting SBMO to DCMMSB

Since for stochastic blockmodel with overlaps (SBMO) [2], $\mathbf{P} = \rho\mathbf{Z}\mathbf{B}\mathbf{Z}^T$, where rows of $\mathbf{Z}$ are binary assignments to different communities, we have $\mathbf{P} = \rho\mathbf{Z}\mathbf{B}\mathbf{Z}^T = \rho'\mathbf{\Gamma}\mathbf{\Theta}\mathbf{B}\mathbf{\Theta}^T\mathbf{\Gamma}$, where $\gamma_i' = \|\mathbf{e}_i^T\mathbf{Z}\|_1 \in [K]$, $\boldsymbol{\theta}_i = \mathbf{e}_i^T\mathbf{Z}/\|\mathbf{e}_i^T\mathbf{Z}\|_1$, $\gamma_i$ is normalized from $\gamma_i'$ to sum to $n$ for identifiability and $\rho' = \rho(\sum\gamma_i'/n)^2$. We can see each SBMO model is corresponding to an identifiable DCMMSB model, thus we can use SVM-cone to recover SBMO model. The way to get binary assignment can be easily done by setting threshold as $1/K$ for each element in $\hat{\mathbf{\Theta}}$.

## L  Closed form rate for known special cases

For a Stochastic Blockmodel (SBM) with $K = 2$ classes of equal size and standard parameters ($\rho = p$, $\mathbf{B}_{11} = \mathbf{B}_{22} = 1, \mathbf{B}_{12} = \mathbf{B}_{21} = q/p$), our result suggests that as long as $(p - q)/\sqrt{p} = \tilde{\Omega}(1/\sqrt{n})$, SVM-cone will consistently estimate the label of each node uniformly with probability tending to one. This is similar to separation conditions in existing literature for consistent estimation in SBMs, up-to a log factor.

# M   Network statistics for DBLP datasets

Table M.1: Network statistics

(a) DBLP coauthorship networks.

| Dataset | DBLP1 | DBLP2 | DBLP3 | DBLP4 | DBLP5 |
|---|---|---|---|---|---|
| # nodes $n$ | 30,566 | 16,817 | 13,315 | 25,481 | 42,351 |
| # communities $K$ | 6 | 3 | 3 | 3 | 4 |
| Average Degree | 8.9 | 7.6 | 8.5 | 5.2 | 6.8 |
| Overlap % | 18.2 | 14.9 | 21.1 | 14.4 | 18.5 |

(b) DBLP bipartite author-paper networks.

| Dataset | DBLP1 | DBLP2 | DBLP3 | DBLP4 | DBLP5 |
|---|---|---|---|---|---|
| # nodes $n$ | 103,660 | 50,699 | 42,288 | 53,369 | 81,245 |
| # communities $K$ | 12 | 6 | 6 | 6 | 8 |
| Average Degree | 3.4 | 3.4 | 3.6 | 2.6 | 3.0 |
| Overlap % | 6.3 | 5.6 | 5.7 | 6.9 | 9.7 |

# N   Wall-clock time on the DBLP bipartite author-paper networks

Figure N.1: The wall-clock time of the competing methods respectively on the biparite author-paper DBLP network. BSNMF was out of memory for DBLP1 and DBLP5.

## O   Statistics of topic modeling datasets

Table O.1: Statistics of topic modeling datasets

| Corpus | Vocabulary size $V$ | Number of documents $D$ | Total number of words |
|---|---|---|---|
| NIPS[1] | 5002 | 1,491 | 1,589,280 |
| NYTimes[1] | 5004 | 296,784 | 68,876,786 |
| PubMed[1] | 5001 | 7,829,043 | 485,719,597 |
| 20NG[2] | 5000 | 9,540 | 886,043 |
| Enron[1] | 5003 | 29,823 | 4,963,162 |
| KOS[1] | 5001 | 3,412 | 405,190 |

## P   Topics in Real Data

Table P.1: Top-10 word of 5 topics for different topic modeling datasets

| Corpus | Top-10 words |
|---|---|
| NIPS | algorithm data problem method parameter point vector distribution error space |
| | neuron output pattern signal circuit visual synaptic unit layer current |
| | data unit training output image information object recognition pattern point |
| | unit hidden output layer weight object pattern visual representation connection |
| | error algorithm training weight data parameter method problem vector classifier |
| NYT | con son solo era mayor zzz_mexico director sin fax sector |
| | zzz_bush government school campaign show american member country zzz_united_states law |
| | company companies market stock business billion plan money analyst government |
| | team game season play player games run coach win won |
| | file sport zzz_los_angeles notebook internet zzz_calif read output web computer |
| PubMed | receptor expression gene binding system function region genes dna mechanism |
| | concentration strain gene dna system expression region genes test function |
| | tumor gene expression disease genes lesion mutation region dna clinical |
| | rat concentration plasma day serum animal liver drug response administration |
| | children disease clinical year test therapy women system diagnosis drug |
| 20NG | key government car chip state including information cs number long |
| | god jesus bible question things life christian world christ true |
| | year michael game team cs games win play including car |
| | drive mb scsi windows card hard disk dos computer drives |
| | windows window dos file files program card fax run win |
| Enron | report status changed payment approved approval amount paid due expense |
| | database error operation perform hourahead data file process start message |
| | power california customer gas order deal list office forward comment |
| | message contract corp receive offer free send list received click |
| | hourahead final file hour data price process error detected variances |
| KOS | iraq administration military iraqi president american troops bushs officials soldiers |
| | voting vote senate polls governor electoral voter media voters primary |
| | percent senate race elections republican party state voters campaign polls |
| | senate polls governor electoral primary vote ground races voter contact |
| | dean edwards primary clark gephardt lieberman iowa results polls kucinich |

## Footnotes

[1]`https://archive.ics.uci.edu/ml/datasets/Bag+of+Words`

[2]`http://qwone.com/~jason/20Newsgroups/`