[Reviews · NeurIPS 2018]

Reviewer 1



The paper studies the problem of overlapping clustering in relational data under a model that encompasses some other popular models for community detection and topic modeling. In this model, the samples are represented in a latent space that represents the clustering memberships in which the coordinates are convex combinations of the corners of a cone. The authors propose a method to find these corners using one-class SVM, which is shown to exactly recover the corners in the population matrix, and vectors that are close to the corners in an empirical matrix. The authors establish consistency results for clustering recovery in network and topic models, and evaluate the performance of the method in synthetic and real data, comparing with other state-of-the-art methods. The problem of overlapping clustering has received increasing attention and is of great interest in different communities. The authors presented a method that can be used in several contexts. The geometric aspects of overlapping clustering models have been studied previously, but the approach followed by the authors is very original and theoretically justified. The method is simple and efficient. I found the method especially relevant for overlapping clustering in network models, as the current approaches in the literature usually require stronger assumptions or make use of combinatorial algorithms. I am not very familiar with the literature in topic modeling, but the method requires to have anchor words on each topic, which seems to be a strong assumption (see Huang et al. "Anchor-Free Correlated Topic Modeling: Identifiability and Algorithm" NIPS 2016). Additionally, there seems to be other work that study the geometrical aspects of topic models, so it would be worth if the authors make more comments in the literature review or numerical comparisons (see for example Yurochkin and Nguyen "Geometric Dirichlet Means algorithm for topic inference", NIPS 2016). The paper is in general clearly written, and Section 2 is well explained and motivated. However, I found the conditions and results in Section 3 hard to interpret since the authors do not introduce identifiability constraints on the parameters. Maybe this issue can be clarified if the authors are able to obtain closed-forms of the rates for special cases, and compare with other rates in the literature. The authors make reference to other models which I think should be formally defined (at least in the appendix) for completeness. UPDATE: Thanks to the authors for clarifying my questions in the rebuttal letter

Reviewer 2



This paper employs one-class SVM to generalize several overlapping clustering models. I like such unifying framework very much, which builds connection among the existing mature knowledge. Here are my detailed comments. 1. The authors might want to change the title so that this paper can be easily searched. 2. The sizes of Figure 1 are not inconsistent. 3. The presentation should be polished. (1) The second and third paragraphs in the introduction part are disconnected. (2) Some notations are not clearly illustrated, such as E in Lemma 2.1 (3) The authors should summarize several key points for the generalization. And a table along these points to generalize network and topic model should be provided 4. Do you try fuzzy c-means? 5. The experimental part is relatively week. (1) Why choose RC as the evaluation metric? Some widely used ones, such as NMI, Rn are encouraged. (2) The authors might want to demonstrate the performance difference of their proposed SVM-core and the traditional model. (3) The time complexity and execution time of SVM-core should be reported.

Reviewer 3



Summary: The authors propose and analyze an algorithm based on 1-class svm for solving the mixed membership clustering problem. Both theoretical and empirical results are presented. 1. The paper is easy to read and technically sound. The presentation of the main idea is clear and convincing. 2. It would be helpful if the authors could provide some remarks on how to best interpret the results such as Theorem 3.1 and 3.2. For instance, which property dominates the error bound? 3. Since the recovery guarantee relies on row-wise error (in the non-ideal case), I wonder how outliers affect the performance. 4. I wonder how the algorithm performs when most members are almost pure, as in the non-overlapping case. 5. Some of the terms are not defined when they first appear in the paper. For example, kappa and lambda, also DCMMSB and OCCAM. Fixing these would improve the readability.